# Fosl2 facilitates chromatin accessibility to determine developmental events during follicular maturation

Hongyong Zhang[1,8] ✉, Zechen Li[2], Yanmei Zhu[1], Wencong Lyu[3], Wenlu Wei[1], Haochen Wang[3], Shuangjie Tian[4], Wei Yue[5], Jiajing Zhong[6], Qing-Yuan Sun [7] & Yiting Guan [1,8] ✉

Granulosa cells (GCs) are the most dynamically responsive cell lineage to encourage continuous folliculogenesis; however, developmental dynamics and interplay with downstream transcription circuitry remain unclear. Here, we unravel the redistribution of genome-wide chromatin areas that drive broad developmental-related transcriptomic alterations during follicular maturation in murine and porcine GCs. Distinct GC-activated accessibility regions (GAAs) at the ovulatory phase are responsible for augmenting flanking GC-involved developmental gene (GDG) expression, which are essential for transcriptional responses to developmental cues. Mechanistically, the transcription factor Fosl2 is strongly recruited to GAAs, facilitating chromatin accessibility state transition. Elevated GAA signals driven by Fosl2 loading induce a significant upregulation of adjacent GDG expression. Additionally, GC-specific Fosl2 deletion in mice perturbs GC cellularity, leading to subfertility related to reproductive aging. Together, we highlight a dynamic chromatin accessibility landscape during follicular maturation, revealing the indispensable Fosl2 function not only controls transcriptional activation via a reconfigured chromatin state, but also orchestrates intricate signaling pathways that are fundamental for ovulation and reproduction.

Millions of women worldwide struggle with female infertility. Any malfunction caused mainly by meiotic anomalies extensively imperils reproductive potential and fertility rates[1–3]. At the heart of the reproductive process lies the ovarian follicle, a structure that plays a pivotal role in female fertility. Within the ovary, the follicle establishes a complex bidirectional communication network between oocytes and supporting cells, which is crucial for synchronized oocyte maturation and follicular growth, thereby ensuring precise developmental gene expression patterns[4]. Therefore, clarifying the regulatory mechanisms that govern ovulation and folliculogenesis is a critical research imperative for combating organ dysfunction and its associated impact on female fertility.

[1]Zhanjiang Institute of Clinical Medicine, Central People's Hospital of Zhanjiang, Guangdong Medical University, Zhanjiang, PR China. [2]Precision Clinical Laboratory, Central People's Hospital of Zhanjiang, Guangdong Medical University, Zhanjiang, PR China. [3]The MOE Key Laboratory of Cell Proliferation and Differentiation, School of Life Sciences, Peking University, Beijing, PR China. [4]State Key Laboratory of Stem Cell and Reproductive Biology, Institute of Zoology, Chinese Academy of Sciences, Beijing, PR China. [5]College of Life Science, Shenyang Normal University, Shenyang, PR China. [6]Department of Reproductive Health and Infertility, Central People's Hospital of Zhanjiang, Guangdong Medical University, Zhanjiang, PR China. [7]Guangzhou Key Laboratory of Metabolic Diseases and Reproductive Health, Guangdong-Hong Kong Metabolism & Reproduction Joint Laboratory, Reproductive Medicine Center, Guangdong Second Provincial General Hospital, Guangzhou, PR China. [8]These authors jointly supervised this work: Hongyong Zhang, Yiting Guan. ✉e-mail: zhanghongyong13@mails.ucas.ac.cn; ytguan@pku.edu.cn

Oocyte maturation is a complex dynamic process, tightly controlled by autocrine and paracrine regulatory factors[5]. In mammals, oocyte maturation is highly dependent on the neighboring granulosa cells (GCs) support. As the most abundant cell in the ovary, proliferating GCs are progressively released from the ooplasm and undergo cumulus expansion as the follicles mature from the antral phase to the ovulatory phase. They possess stem-like properties that protect the oocytes from the detrimental effects of the microenvironment, allowing for nutrient and signal exchange, which in turn promotes oocyte growth[6]. Functioning as signaling hubs, GCs facilitate dialogue with oocytes throughout developmental phases and simultaneously receive directives from the pituitary gland. During the transition from antral to ovulatory follicles, GCs communicate with oocytes, directly affecting gene expression and protein synthesis, ultimately promoting meiotic maturation of the oocyte[7]. Alternatively, GC apoptosis has been recognized as a key factor triggering follicular atresia[8–10]. In developing GCs, the follicle-stimulating hormone receptor (FSHR) highly expressed in GCs is the most important mediator of estradiol synthesis, serving as the prerequisite for the acquisition of oocyte maturation competence[11]. Given that oocyte competence is exclusively developed through its bidirectional interaction with GC, the closely surrounding GCs offer a valuable biological resource for conducting molecular analyses designed to assess the developmental potential of the oocyte. Indeed, disruptions in GCs during follicular maturation can cause ovarian dysfunction and various diseases[12,13]. Thus, elucidating the regulatory mechanism of GC during folliculogenesis in diverse physiological contexts remains fundamental for understanding follicle growth control and reproductive diseases pathogenesis.

The functional architecture of the genome exists in a state of dynamic equilibrium, with alterations in epigenetic landscapes known to contribute to extensive gene regulatory networks in folliculogenesis[14–16]. In terms of epigenetic control, the cooperation between chromatin accessibility and putative regulatory elements, including H3K4me3-decorated promoters and H3K27ac-decorated enhancers, fundamentally changes the chromatin state, thereby determining downstream gene expression patterns and cellular fates[17–19]. As a major genetic process affecting biological behaviors at specific loci, accessible chromatin recruits master transcription factors (TFs) to target closed chromatin and urge these regions to open. Importantly, the advent of transposase-accessible chromatin with sequencing (ATAC-seq) has greatly expanded our knowledge regarding the mechanisms governing epigenetic alteration and the control of gene expression during fundamental biological processes[20,21]. To date, a wealth of outstanding research has highlighted the chromatin remodeling events that have happened from terminally differentiated gametes to the highly totipotent blastomeres during the activation of the embryonic genome across different species[22–24]. In this context, several crucial pluripotency TFs have been uncovered as important, early development and transcriptional regulators in mammalian preimplantation development[25–28]. GC proliferation during follicular maturation is a prerequisite for oocyte maturation and subsequent embryogenesis. However, despite the recognized importance of these TFs, the precise mechanisms by which they interact with regulatory elements to control the transcriptional circuitry are yet to be fully understood. Characterizing chromatin accessibility changes in the control of gene expression in synchronized oocytes and GCs during follicle maturation is crucial for deciphering the mechanisms that govern signal transduction and follicular development, potentially offering significant insights for enhancing fertility.

In this study, we characterized the reorganization of chromatin accessibility that precipitates extensive transcriptomic alterations associated with GCs across different follicular maturation phases in murine and porcine models. A pronounced increase in accessibility sites, designated as GC-activated accessibility regions (GAAs), was identified in ovulatory follicles, which enhanced the expression of neighboring GC-involved developmental genes (GDGs). Meanwhile, the transcription factor Fosl2 was found to be significantly enriched within GAAs, acting as an essential initiator for opening chromatin. The suppression of Fosl2 led to a reduction in GAA activity and consequently perturbed the downstream expression pattern of GDGs. In female mice, specific Fosl2 ablation in GCs caused subfertility related to reproductive aging, attributed to compositional and functional alterations in GC subclusters within ovarian cellularity. Collectively, our findings elucidate the critical role of Fosl2 in orchestrating chromatin accessibility dynamics, thereby advancing our comprehension of the regulatory mechanisms governing GCs during follicular maturation.

## Results

### GCs undergo globally distinct chromatin landscape remodeling

Due to their similar compartmentalization and folliculogenesis patterns to human ovaries, porcine ovaries are frequently used in reproductive medicine[29,30]. We derived porcine cumulus-oocyte complexes (pCOCs) from porcine ovaries and matured them in vitro to generate porcine granulosa cells (pGCs) and oocytes at different phases (Fig. 1a and Supplementary Fig. 1a). Notably, pGCs showed compact clusters around oocytes in the antral follicles; however, these clusters dispersed as the ovulatory phase was reached, whereas oocytes themselves exhibited limited morphological alterations following their transition from the germinal vesicle (GV) stage to the metaphase II (MII) stage (Fig. 1b). Meanwhile, pGC transcriptomic dynamics displayed more pronounced fluctuations when compared to the oocyte (Supplementary Fig. 1b). In pGCs, several biomarkers including *Cyp11a1*, *Star*, *Smad1*, along with genes related to growth and differentiation, were substantially elevated throughout follicular maturation (Fig. 1c and Supplementary Fig. 1c)[31,32]. In contrast, the expression of silent genes upon GC proliferation, including *Nr5a1* and *Cdc20*, was decreased at the ovulatory phase, while oocyte-specific markers, such as *Lhx8*, *Sycp3*, and *Sox30*, were not observed in pGCs spanning follicular maturation phases (Fig. 1c and Supplementary Fig. 1c)[33–37]. We also noticed that MII oocytes devoid of pGCs support displayed extensive transcriptional changes when compared to normal MII oocytes, involving many fundamental genes during oocyte maturation, and emphasizing indispensable GC physical support and a microenvironment for oocyte growth (Fig. 1d)[38–40]. Moreover, the differentially expressed genes (DEGs) of GCs were always enriched in multiple signaling processes, which are required for follicular maturation (Supplementary Fig. 1d, e). In particular, those overexpressed genes at the ovulatory phase were predominantly engaged in folliculogenesis pathways, reinforcing the notion that the rewiring of a developmental gene subset occurred during follicular maturation (Supplementary Fig. 1f, g).

Chromatin accessibility reflects the transcriptome profiles that determine cell fate[41]. Given the unique alterations of developmental pathways and gene changes that have appeared during follicular maturation, we used ATAC-seq to examine accessible chromatin in pGCs and oocytes across different follicular maturation phases. As reported previously, poor ATAC-seq enrichment for oocytes was obtained at neither the GV nor the MII stage, despite several attempts[42]; therefore, we focused our analysis on genome-wide chromatin accessible pGC data both at the antral and ovulatory phases (Supplementary Fig. 2a, b). Unsurprisingly, highly reproducible ATAC-seq data confirmed the typical nucleosome phasing patterns in insert size distributions (Supplementary Fig. 2c, d). The identified chromatin accessibility was predominantly situated in intergenic regions containing abundant *cis*-regulatory elements, with genome annotations

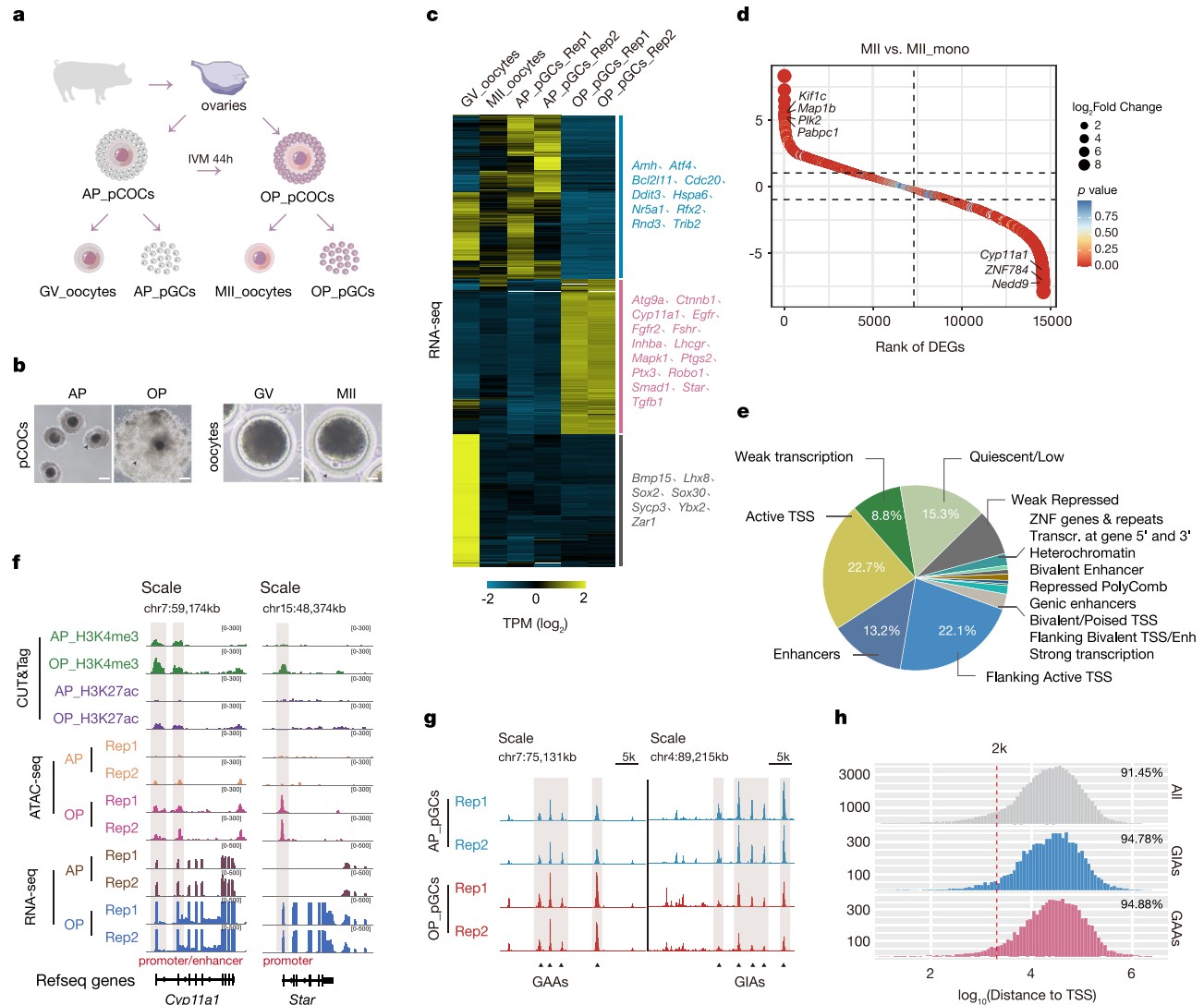

**Fig. 1 | Open chromatin remodeling and transcriptional networks occur during follicular maturation.** **a** Schematic showing the acquisition of porcine granulosa cells (pGCs) and oocytes. pCOCs porcine cumulus-oocyte complexes, IVM in vitro maturation, GV germinal vesicle, MII metaphase II, AP antral phase, OP ovulatory phase. **b** Bright field images showing the feature of pGCs and oocytes under different follicular maturation phases. Results are validated through at least three biological replicates. Scale bar, left, 100 μm; right, 20 μm. Results shown are representative of n = 3 biologically independent experiments with similar results. **c** Heatmaps showing the expression (TPM) of RNA-seq in pGCs and oocytes across different maturation phases (left), with example genes listed (right). **d** Rank of the differentially expressed genes (DEGs) between the oocytes at the MII stage (MII) and MII oocytes without pGCs support (MII_mono). Representative genes are labeled. The circle size represents the fold change of genes, and the color

represents the *p* value. The *p* value was generated from a two-sided Wald test. **e** Annotations of all open chromatin regions presenting the chromatin states trained using public data from the ENCODE project. TSS transcription start site. **f** Integrative Genomics Viewer (IGV) snapshot displaying the CUT&Tag signal of H3K4me3, H3K27ac, and the ATAC-seq and RNA-seq signals in pGCs during follicular maturation at the representative gene *Cyp11a1* and *Star* loci. Vertical gray boxes indicate the potential promoter and enhancer peaks predicted with CUT&Tag. **g** Insertion tracks of GAAs and GIAs at chromosome 7 (left) and chromosome 4 (right) loci. Differentially open regions are labeled with grey boxes. GAAs GC-activated accessibility regions, GIAs GC-inactivated accessibility regions. **h** Distance distribution between ATAC-seq peaks (all peaks, GAAs, and GIAs) and TSSs. Proportions >2k are labeled.

also showing that a large fraction of these peaks overlapped with promoter and enhancer signals, consistent with previous findings (Fig. 1e and Supplementary Fig. 2e, f)[37]. Considering that *cis*-regulatory elements in accessible chromatin regions potentially control gene regulatory networks, we examined the promoter and enhancer occupancy across follicular maturation phases using H3K4me3 and H3K27ac-driven CUT&Tag methods. An increased portion of our ATAC-seq peaks overlapped with promoter signals (Supplementary Fig. 2g, h); the expression levels of prominent biomarkers, for instance, *Cyp11a1* and *Star*, were significantly elevated at the ovulatory phase, with strong H3K4me3 signals observed at their promoters, but not at H3K27ac-anchored enhancer regions, suggesting considerable

chromatin accessibility promoter occupancy in follicular maturation (Fig. 1f).

### GAAs at promoter regions control neighboring GDG expression

Chromatin accessibility and gene expression at various developmental stages exhibit shared features, yet the differentially accessible regions (DARs) specific to pGCs are particularly essential for folliculogenesis. From analyses, >10,000 DARs were arranged using quantitative peak signals, including 8646 GC-activated accessibility regions (GAAs) and 7954 GC-inactivated accessibility regions (GIAs) during follicular maturation (Fig. 1g and Supplementary Fig. 3a). Consistently, DARs were localized to intergenic regions at approximately 2 kb from

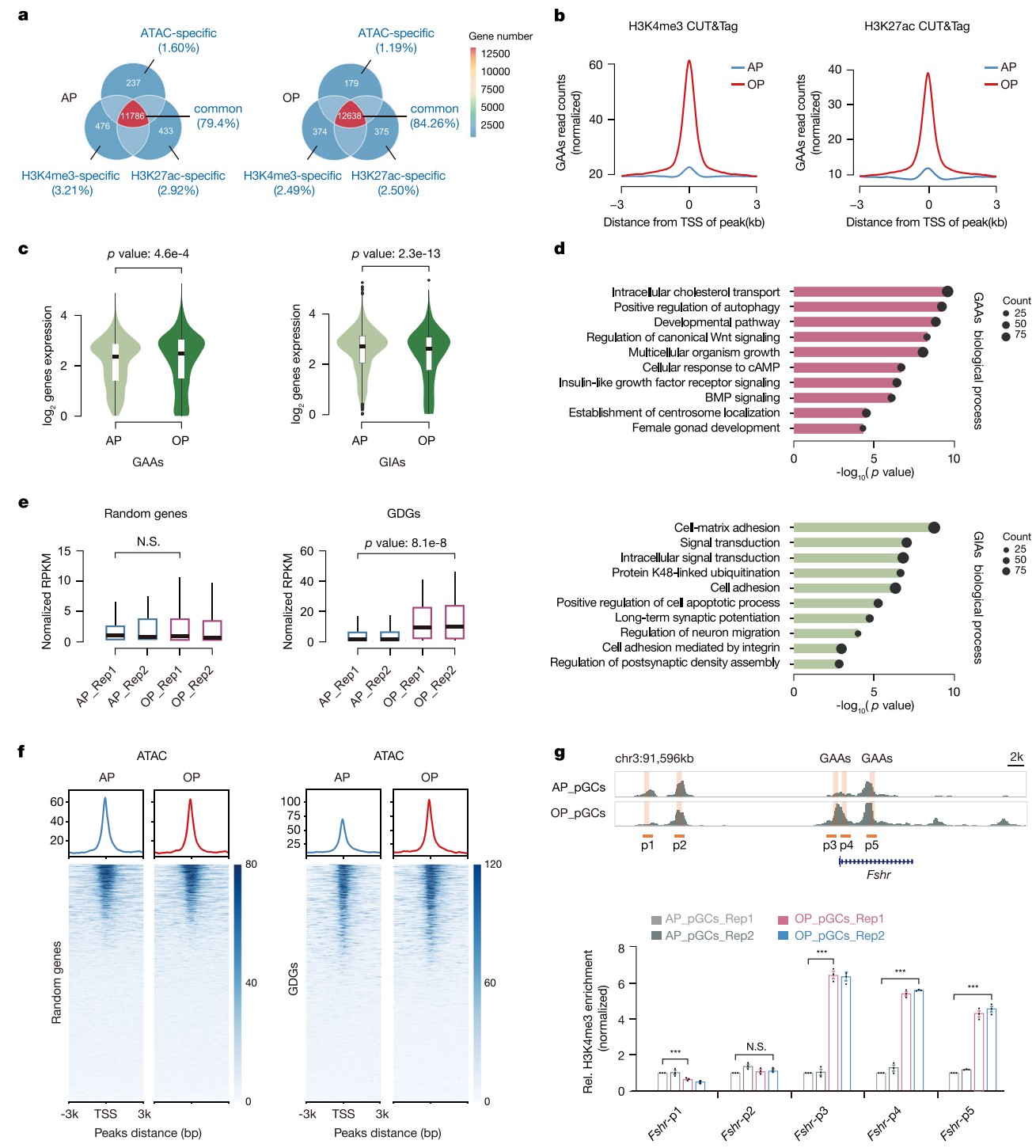

transcription start sites (TSSs), echoing promoter occupancy in ATAC-seq peaks (Fig. 1h). Although the majority of these peaks shared common regions with *cis*-regulatory elements, no significant differences in promoter or enhancer signals were observed between antral and ovulatory phases across the genome (Fig. 2a and Supplementary Fig. 3b). However, as both *cis*-regulatory elements signals were examined in dynamic DARs, we noted that genes flanking GAAs showed increased H3K4me3 and H3K27ac signals at the ovulatory phase, whereas genes flanked by GIAs exhibited reduced levels of these histone marks (Fig. 2b and Supplementary Fig. 3c). It seemed that dynamic DARs residing in *cis*-regulatory elements appeared to contribute to specific chromatin opening that facilitated neighboring gene activation. To verify this, we evaluated whether *cis*-elements,

particularly those anchored to promoters, could influence adjacent gene expression. Indeed, GAA-neighboring genes were upregulated while those adjacent to GIAs were downregulated, indicating that regulated gene expression was invariably correlated with chromatin accessibility at promoter regions (Fig. 2c).

Subsequent Gene ontology (GO) enrichment analysis was performed to explore the biological functions of genes adjacent to DARs. Of note, genes neighboring GAAs were always implicated in developmental regulation, including nutrient transport, growth signal secretion, and other developmental pathways required for follicular maturation (Fig. 2d)[43–45]. In agreement with GO enrichment analyses, a subset of GC-involved developmental genes (GDGs), which were significantly upregulated at the ovulatory phase, were found in proximity

**Fig. 2 | GAAs located in promoter regions modulate adjacent GDG expression.**
**a** Venn diagram showing ATAC-seq signals overlap with H3K4me3 and H3K27ac CUT&Tag signals at the antral and ovulatory phases. Gene numbers are indicated with colors. **b** Enrichment plots displaying normalized GAA read densities for H3K4me3 and H3K27ac CUT&Tag signals. Tracks are centered at the TSS and extend ±3 kb. **c** Violin plots displaying the distributions of expression changes in GAA- (left, $n = 2722$) and GIA-adjacent (right, $n = 2717$) genes. The $p$ value was generated from a two-sided Student's $t$ test. Violin plots display the full distribution: center line: median; upper/lower hinges: 75th and 25th percentiles, upper and lower whiskers represent the data extending from the hinge to at most 1.5 times the interquartile range. **d** Gene Ontology (GO) enrichments of GAAs (upper) and GIAs (lower) using Genomic Regions Enrichment of Annotations Tool (GREAT) analysis. Bar length represents the enriched $p$ value for biological processes. The circle size represents the gene counts involved in the indicated pathway. The $p$ value was generated from a one-sided Hypergeometric test. **e** Box plots showing the average

expression levels (RPKM) of randomly selected genes (left, $n = 2068$) and GDGs (right, $n = 2068$) under different maturation phases. The $p$ value was generated from a two-sided Wilcoxon rank-sum test. N.S. not significant. Boxplot summary statistics are: center line: median; upper/lower hinges: 75th and 25th percentiles, upper and lower whiskers represent the data extending from the hinge to at most 1.5 times the interquartile range. **f** Heatmaps and enrichment plots displaying normalized read densities of ATAC-seq signals for randomly selected genes and GDGs under different maturation phases in pGCs. Tracks are centered at the TSS and extend ±3 kb. **g** Location diagram of H3K4me3 ChIP-qPCR primers within the *Fshr* locus (upper). ChIP-qPCR is used to measure the relative H3K4me3 levels for GAAs within the corresponding *Fshr* gene at the antral and ovulatory phases (lower). IgG is used as the negative control. The enrichment is normalized to a 1:10 dilution of the input. Error bars indicate the mean ± S.E.M. ($n = 3$ biological replicates). The $p$ value was generated from a two-sided Student's $t$ test. N.S. not significant, ***$p < 0.001$. Source data are provided as a Source Data file.

to GAAs, including the well-recognized GC biomarkers (Supplementary Fig. 3d). We measured the fluctuation of GDG expression using a defined list ($n = 2068$, Methods), and observed that their expression was prominently upregulated, while a randomly selected set of genes displayed unchanged expression (Fig. 2e). Conversely, the pathways enriched in genes adjacent to GIAs, such as signal transduction and regulation of neuron migration, indicated that the pGCs at the ovulatory stage lost growth capacity and connections between themselves, which aligns with previous studies (Fig. 2d)[14,46]. As genes neighboring GAAs were always upregulated, and many were well-known GDGs, we speculated if these GDGs had regulatory roles in chromatin accessibility dynamics. Indeed, peaks at the ovulatory phase led to substantial enrichment of GDGs compared with the signal at the antral phase, whereas randomly selected genes showed no signal changes (Fig. 2f). Key GC proliferation regulators, e.g., *Fshr* and *Atg9a* belonging to GDGs, were flanked by GAAs that exhibited significant elevations in chromatin accessibility as determined by H3K4me3-decorated ChIP-qPCR in pGCs during follicular maturation (Fig. 2g and Supplementary Fig. 3e–g)[47,48]. Correspondingly, the transcriptome expression levels of these two genes both increased across follicular maturation phases, indicating the potential GAA regulatory roles in harboring adjacent GDG expression. Taken together, the results above indicate that GAA-harboring GDGs, accompanied by multiple developmental pathways, are influenced by cellular signaling; moreover, these H3K4me3-decorated GAAs may serve as contributing factors or prerequisites to enhance adjacent GDG expression.

### Fosl2 recruitment to GAAs is required for shaping downstream transcriptome profiles

To determine the potential chromatin remodeling driver that may influence downstream transcriptome profiles, we conducted a regulatory elements motif enrichment within GAAs using the HOMER algorithms. Two of the most highly-ranked motifs shared considerable homology with consensus binding sites for activator protein 1 (AP-1) and zinc finger (ZF) (Fig. 3a). Interestingly, the AP-1 motif was specifically enriched in GAAs, while the ZF motif was enriched both in GAAs and GIAs (Fig. 3b and Supplementary Fig. 4a). To identify TFs that potentially regulating GAAs, we screened for AP-1 TF expression changes in pGCs during follicular maturation, and identified Fosl2, which showed considerable mRNA and protein expression increases from the antral to the ovulatory phase (Supplementary Fig. 4b–d). Accordingly, increased fluorescence intensity of Fosl2 was observed at the ovulatory phase (Fig. 3c and Supplementary Fig. 4e).

Considering the potential for Fosl2 to engage in long-lasting interactions with DNA prior to follicular maturation, we conducted CUT&Tag analysis in pGCs across different maturation phases to assess genome-wide Fosl2 occupancy and determine its involvement in chromatin remodeling at GAAs. Overall, Fosl2 displayed a significant

increase in genome-wide signal (Supplementary Fig. 4f). After quantifying and comparing motif scores, differentially occupied peaks had lower motif scores on average than constitutively occupied peaks, aligning with the notation that increased Fosl2 expression allows binding to lower affinity sites (Fig. 3d). Meanwhile, the differential chromatin accessibility peaks and Fosl2 occupancy peaks were relatively correlated, suggesting that a high fraction of differential accessibility is attributable to differential Fosl2 occupancy (Fig. 3e). To further verify this, we categorized differential Fosl2 occupancy levels into three clusters, and analyzed accessibility characteristics. We discovered that Fosl2-gained peaks (Fig. 3f, ovulatory-specific), rather than shared or loss peaks (Fig. 3f, antral-ovulatory shared and antral-specific), were specifically associated with the newly opened chromatin, as well as the active regulatory regions. Moreover, Fosl2-gained peaks tended to be enriched near genes involved in multiple development-related pathways, in parallel with the observation in GAAs (Fig. 3f). Indeed, Fosl2 enrichment signals on GAAs were elevated, suggesting potential Fosl2 roles in harboring GAA activity in terms of follicular maturation (Fig. 3g). To further verify that GAA dynamics were precise targets for Fosl2 occupancy, we calculated overlaps between differential Fosl2 occupancy peaks and GAAs, and found that approximately 75% of GAAs overlapped with Fosl2-gained peaks, with minimal overlap with Fosl2-loss peaks (Fig. 3h). Meanwhile, Fosl2 binding was selectively enriched at upregulated gene promoter regions, but not downregulated genes. Distal Fosl2 binding peaks were also predominantly associated with upregulated genes, revealing a principal role for Fosl2 as a transcriptional activator modulating GAA remodeling (Fig. 3i). Altogether, the results above suggest that Fosl2 is indispensable for altering the chromatin state transitions during follicular maturation, particularly in facilitating the opening of promoter-anchored GAA regions.

### Fosl2 alters GAA-harboring chromatin accessibility during follicular maturation

To examine if sustained Fosl2 expression was crucial for maintaining open chromatin states at GAAs, we performed a stable knockdown of Fosl2 in pGCs (Supplementary Fig. 5a). ATAC-seq analyses of Fosl2 knockdown samples identified substantially reduced chromatin accessibility (Fig. 4a). Meanwhile, the collapse predominantly occurred at intergenic regions enriched with Fosl2 motif sites (Fig. 4b and Supplementary Fig. 5b). However, in contrast to elevated GAA peaks found in follicular maturation, the Fosl2 knockdown displayed considerably decreased GAA activity, showing Fosl2-driven perturbations in global open chromatin and a particular impact on the signal intensities of GAAs (Fig. 4c). After evaluating correlations between differential chromatin accessibility peaks in different maturation phases and chromatin accessibility in Fosl2 knockdown cells, we found that closed chromatin regions in Fosl2 knockdown cells were inversely correlated

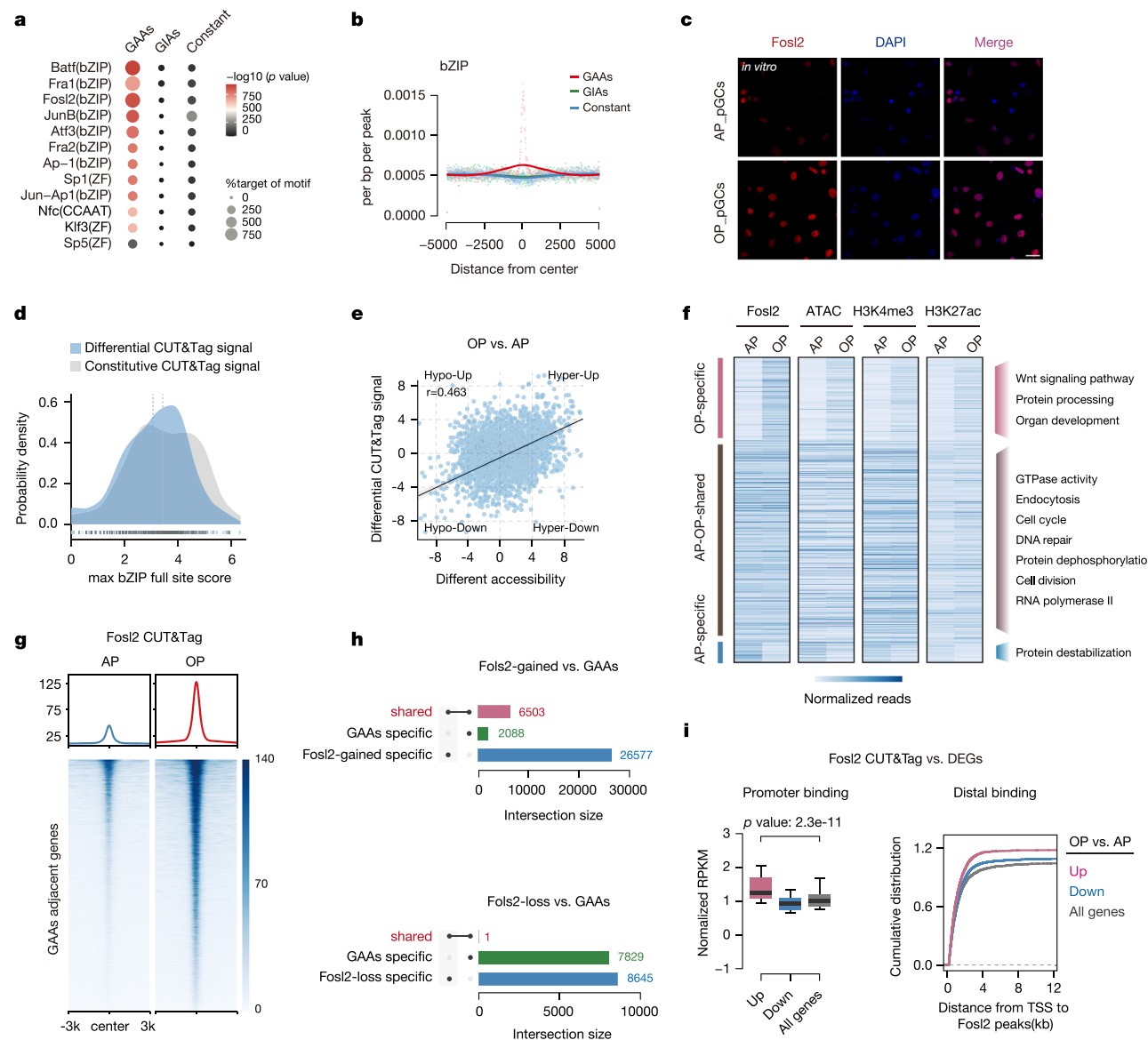

**Fig. 3 | The engagement of Fosl2 with GAAs is essential for the expression of downstream genes. a** Top-ranked enriched motifs among GAAs and GIAs are listed, as determined using HOMER algorithms. The circle size represents the percentage of motifs in the target regions, and the color represents the *p* value. The *p* value was generated from a one-sided Fisher's exact test. **b** Position-weight matrices displaying the observed accessibility densities of GAAs, GIAs, and constant regions in bZIP motif family. **c** Immunofluorescence staining of Fosl2 both at the antral and ovulatory phases in pGC. The position of the nucleolus is indicated by DAPI staining. Results shown are representative of *n* = 3 biologically independent experiments with similar results. Scale bar, 20 μm. **d** Distribution of motif scores of sites within bZIP CUT&Tag signals, either those that gave more signal at the ovulatory than antral phase (differential) or those that are not significantly different (constitutive). The maximum scoring bZIP full site within each CUT&Tag peak is used. The constitutive peaks have a higher mean motif score than the differential peaks (*p* = 2.3e−4). The *p* value was generated from a two-sided Wilcoxon rank-sum test. **e** Spearman's correlation of differential Fosl2 CUT&Tag signals in accessible regions and differential accessibility between different maturation phases

with GAA accessibility changes (*r* = 0.463, *p* = 6.1e−4). The *p* value was generated from a two-sided test. **f** Heatmaps showing the different clusters of Fosl2 binding signals and their mapping in ATAC-seq, H3K4me3 and H3K27ac CUT&Tag signals (left). The GO enrichment terms are also shown for different clusters (right). **g** Heatmaps and enrichment plots showing normalized GAA read densities of Fosl2 CUT&Tag signals both at the antral and ovulatory phases. Tracks are centered at the peaks and extend ±3 kb. **h** Bar plots displaying the distribution of differential Fosl2 binding signals with GAAs, Intersection numbers are labeled. **i** Box plots showing the average enrichment of Fosl2 binding signals at the promoters (TSS ± 2.5 kb) of upregulated (*n* = 2068), down-regulated (*n* = 2549), and all genes (*n* = 15,397) during follicular maturation (left). The cumulative distributions of genes of the above three categories with defined distances (*x*-axis) between their TSS and nearest distal Fosl2 binding peaks are shown (right). The *p* value was generated from a two-sided Wilcoxon rank-sum test. Boxplot summary statistics are: center line: median; upper/lower hinges: 75th and 25th percentiles, upper and lower whiskers represent the data extending from the hinge to at most 1.5 times the interquartile range.

with GAA accessibility changes (Supplementary Fig. 5c). These observations indicate that Fosl2 acts as a transcriptional activator in restructuring chromatin accessibility.

Attenuation of GAA intensity had a substantial impact on downstream TFs binding following Fosl2 perturbation. For example, using

TOBIAS, we examined TF footprints in differentially accessible sites and unidentified pluripotency TF motifs, including Tead4 and Nrf2, showed decreased ATAC-seq signals upon Fosl2 knockdown, while the Fosl2 motif displayed significantly decreased densities as a validation following Fosl2 silencing (Fig. 4d)[49–51]. To further investigate whether

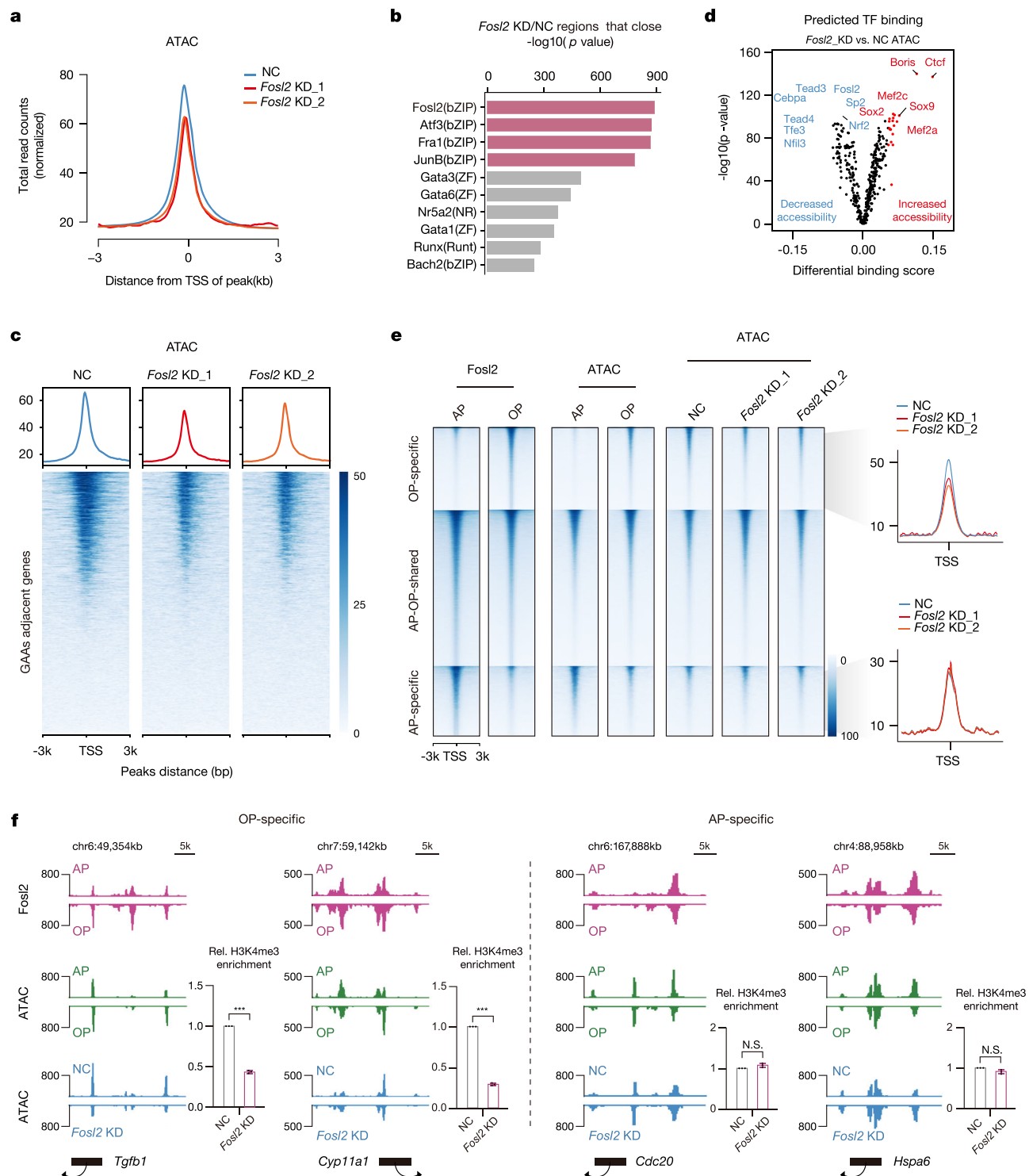

Fosl2 participated in regulating chromatin accessibility at putative promoter-anchored GAA regions, the aforementioned differential Fosl2 occupancy peaks identified with Fosl2 CUT&Tag were assigned to accessibility profiles during follicular maturation and following Fosl2 knockdown. Of note, the strength of ovulatory- and antral-specific peaks in Fosl2 occupancy consistently agreed with chromatin accessibility dynamics in the process of follicular maturation; yet, only the ovulatory-specific peaks exhibited extensive decreases in accessibility upon Fosl2 silencing, contrasting with the stable accessibility of antral-specific peaks (Fig. 4e). For instance, the well-characterized GDGs, *Tgfb1* and *Cyp11a1*, which belonged to the ovulatory-specific cluster, showed robust Fosl2 occupancy and subsequent GAA activity

collapse upon Fosl2 suppression (Fig. 4f, left)[52]. In contrast, genes at antral-specific regions, such as *Cdc20* and *Hspa6*, demonstrated significant Fosl2 occupancy but unchanged accessibility density in cells after Fosl2 knockdown (Fig. 4f, right). These findings suggest that genes targeted by Fosl2 are preferentially downregulated in its absence, revealing the critical role of Fosl2 in sustaining GAA chromatin states.

**Disrupted chromatin accessibility following Fosl2 deficiency leads to impaired GDG expression**

Chromatin accessibility influences transcription by modulating physical TF access to DNA. Building on the putative roles of Fosl2 in GAA

**Fig. 4 | Fosl2 control chromatin accessibility within GAAs during follicular maturation. a** Enrichment of ATAC-seq peaks of the total normalized read densities in normal control (NC) and Fosl2 knockdown (Fosl2 KD) in pGCs. Tracks are centered at the TSS and extend ±3 kb. **b** Motif enrichment of the regions that close after Fosl2 knockdown using GREAT analysis. The highly ranked motifs for bZIP factors are highlighted in red. Bar length represents the enriched *p* value for biological processes. The *p* value was generated from a one-sided Fisher's exact test. **c** Heatmaps and enrichment plots showing normalized GAA read densities of ATAC-seq signals after Fosl2 silencing. Tracks are centered at the peaks and extend ±3 kb. **d** Volcano plot showing the TF footprints in differentially accessible regions in ATAC-seq upon Fosl2 suppression. Example footprints in regions with increasing and decreasing accessibilities are colored in red and blue, respectively. *p* values for their enrichment are also shown on the *y*-axis. The *p* value was generated from a one-sided Fisher's exact test. **e** Heatmaps showing the different clusters of Fosl2 binding signals with CUT&Tag and their mapping in ATAC-seq peaks at the different maturation phases and peaks following Fosl2 suppression (left). The enrichment of normalized ATAC-seq peaks after Fosl2 knockdown is also shown both in antral- and ovulatory-specific clusters (right). **f** IGV views displaying the Fosl2 CUT&Tag, ATAC-seq signals under different maturation phases and signals in Fosl2-silenced cells in representative *Tgfb1*, *Cyp11a1*, *Cdc20* and *Hspa6* from ovulatory-specific (left) and antral-specific (right) clusters. ChIP-qPCR is also used to measure the relative H3K4me3 levels after Fosl2 knockdown within these genes. IgG is used as the negative control. The enrichment is normalized to a 1:10 dilution of the input. Error bars indicate the mean ± S.E.M. (*n* = 3 biological replicates). The *p* value was generated from a two-sided Student's *t* test. N.S. not significant, ***p* < 0.001. Source data are provided as a Source Data file.

remodeling, we explored if Fosl2 directly governed the transcriptional landscape in the process of follicular maturation. In Fosl2 knockdown cells, Fosl2 binding was selectively enriched at the promoters of downregulated genes, but not upregulated ones. In parallel, distal Fosl2 binding peaks preferentially occurred near downregulated genes, indicating its role as a transcriptional activator for downstream genes (Fig. 5a). Similarly, genes with Fosl2 binding sites and a higher number of Fosl2 motifs at their promoters were significantly downregulated following Fosl2 silencing (Fig. 5b).

Given that GAAs are responsible for strong downstream GDGs, we comprehensively analyzed GDG dynamics in response to Fosl2 disruption. At global levels, numerous developmental signaling pathways, including the G2M checkpoint pathway, were significantly disrupted upon Fosl2 suppression (Supplementary Fig. 5d). A plethora of GDGs were significantly downregulated after Fosl2 knockdown, including *Cyp11a1*, *Star*, *Fshr* and other well-recognized growth and differentiation-associated GC genes that are proximal to GAAs (Supplementary Fig. 5e); however, this downregulation was in stark contrast to the stable expression of a randomly selected control gene set (Fig. 5c). Moreover, after examining Fosl2 enrichment in both gene sets, we found GDGs displayed pronounced Fosl2 enrichments at their TSSs (Fig. 5d). Consistent with this observation, in distal genomic regions, we demonstrated substantially stronger binding signals in proximity to GDGs when compared to the random gene set, highlighting its essential role on harboring the wave of GDGs both at their promoters and within distal regions (Fig. 5e). To further elucidate Fosl2-driven GAA functions in shaping GDG transcriptomic repertoires, we examined accessibility densities in these gene sets in Fosl2 knockdown cells. We found that GDGs exhibited a substantial reduction in accessibility densities compared with randomly selected genes, aligning with the observed downregulation of GDG expression following Fosl2 suppression (Fig. 5f). Two representative GDGs, *Fshr* and *Mapk1*, which were proximal to GAAs, displaying elevated gene expression and augmented Fosl2 binding at the ovulatory phase[47,53]. Nevertheless, the transcription program and accessibility density of these two GDGs were robustly decreased after Fosl2 knockdown. Consistently, fluorescence intensities of these two GDGs were also correspondingly attenuated after Fosl2 knockdown (Fig. 5g, left). In comparison, the selected *Fto* and *Gapdh*, which belong to a random gene set displayed unchanged expression both during follicular maturation and Fosl2 silencing scenarios (Fig. 5g, right). Collectively, our data substantiate a model wherein Fosl2 binding to GAAs consolidates an open chromatin conformation, thereby orchestrating a broad transcriptional program of GDGs, which is fundamental to multiple developmental processes.

### Chromatin accessibility governed by Fosl2 is maintained across mammalian species

To further highlight the regulatory Fosl2 axis in vivo, we examined Fosl2-harboring GAA patterns in murine GCs (mGCs). Despite culture variations, mGCs exhibited >10,000 DARs from quantitative peak

signals, and 14072 GAAs in mGCs were identified (Fig. 6a and Supplementary Fig. 6a). Similarly, when assessing the impact chromatin accessibility peaks on GDG regulation, we found that peaks at the ovulatory phase displayed a robust signal for GDGs compared to those at the antral phase, whereas a control set of randomly selected genes showed unchanged signals (Fig. 6b). For instance, a representative folliculogenesis biomarker, *Mapk1*, which formed part of the GDGs set in pGCs, displayed a prominent GAA feature in accessibility density features and increased transcriptional activity signals in mGCs, distinct from the adjacent *Ppm1f* gene pattern (Supplementary Fig. 6b)[53]. Following the examination of enriched regulatory element motifs, AP-1 TFs, including Fosl2, were significantly enriched at GAAs defined in mGCs when compared to the nonspecific binding of NR and Runt families (Fig. 6c and Supplementary Fig. 6c). In mGCs, Fosl2 exhibited intense fluorescence at the ovulatory phase, consistent with findings in pGCs (Fig. 6d).

Throughout most of the course of folliculogenesis, oocyte-GC physical interactions such as gap junctions and microvilli facilitate transfer of molecules between the oocyte and GCs for signaling communication[54,55]. Our studies revealed significant Fosl2 fluorescence in oocytes of pre-antral phase follicles, but this fluorescence diminished in later follicular phases, contrasting with GC observations (Fig. 6e). Moreover, Fosl2 expression in oocytes was strong at the GV stage both in vivo and in vitro, in stark contrast to the muted expression at the MII stage (Fig. 6f and Supplementary Fig. 6d, e). It seemed that Fosl2-driven accessibility remodeling patterns depend on the transmission of Fosl2 shuttles from oocytes to GCs. To verify this, we performed different co-culture strategies to examine Fosl2 expression in pGCs (Fig. 6g). Of note, pGCs co-cultured with oocytes exhibited elevated Fosl2 expression compared to levels in the antral phase, and intriguingly, pGCs cultured apart from oocytes also showed robust Fosl2 expression at the ovulatory phase (Fig. 6h). Altogether, these findings indicate the mechanisms of Fosl2 in orchestrating follicular maturation are conserved across mammalian species and operate independently of direct intercellular connections with oocytes.

### GC-specific Fosl2 deletion causes reproductive impairment in female mice

Fosl2-mediated redistribution of the transcriptome profiling provides an important context for understanding folliculogenesis. To further analyze Fosl2 effects on reproductive capacity, we conditionally depleted Fosl2 in mGCs using *Fosl2^{Flox/Flox}* (*Fosl2^{F/F}*) mice, where LoxP sequences flanked Fosl2 coding sequences. *Cyp19a1-Cre*; *Fosl2^{Flox/Flox}* (*Fosl2^{cKO}*) mice were then generated by crossing to GC-specific *Cyp19a1-Cre* mice (Fig. 7a). Upon attaining sexual maturity, *Fosl2^{cKO}* mice and their littermate controls were mated to age-matched males to evaluate reproductive capacity. Intriguingly, although Fosl2 deficiency did not induce infertility, pups from *Fosl2^{cKO}* females were gradually reduced with time (Fig. 7b and Supplementary Fig. 7a). Meanwhile, after examining the reproduction cycles, the average duration of the reproductive cycle was largely extended in Fosl2-

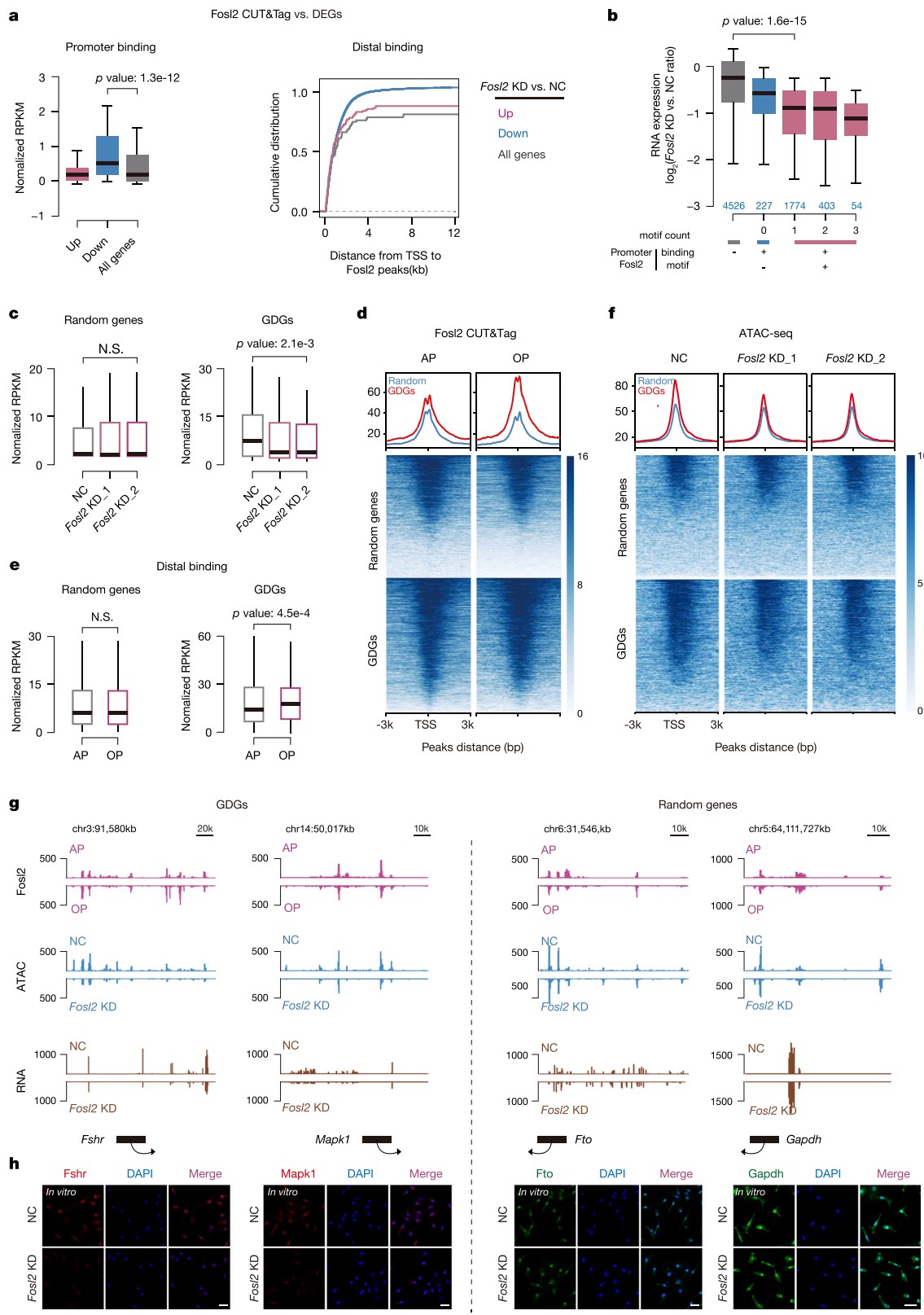

deleted mice when compared to controls (Fig. 7c). It seemed that the absence of Fosl2 had a profound effect on the reproductive capabilities. Indeed, mice deficient in Fosl2 demonstrated a substantial reduction in their sensitivity to gonadotropins, which are hormones crucial for the regulation of follicular maturation, and this decreased responsiveness significantly impacted the efficiency of superovulation treatments (Supplementary Fig. 7b). During the superovulation

process, Fosl2-deficient mice not only had reduced ovulation rates but also experienced a pronounced decline in the quality of ovulation, leading to a substantial decrease in oocyte retrieval (Fig. 7d). The count of post-superovulation corpora lutea (CL), which are essential structures in the ovaries that form after ovulation and are indicative of a successful ovulatory cycle, was markedly lower in Fosl2-deficient mice as compared to the control group (Fig. 7e). These results indicate that

**Fig. 5 | Fosl2 deficiency disrupts chromatin accessibility and leads to the GDG expression suppression. a** Box plots showing the average enrichment of Fosl2 binding signals at the promoters (TSS ±2.5 kb) of upregulated ($n = 178$), down-regulated ($n = 231$), and all genes ($n = 14,491$) following Fosl2 suppression (left). The cumulative distributions of genes of the above three categories with defined distances ($x$-axis) between their TSS and nearest distal Fosl2 binding peaks are shown (right). **b** Box plots displaying the fold changes of gene expression after Fosl2 knockdown for all expressed genes based on the Fosl2 binding states and the numbers of motifs at promoters, with $p$ values indicated (motif count: all: $n = 4526$; 0: $n = 227$; 1: $n = 1774$; 2: $n = 403$; 3: $n = 54$). **c** Box plots showing the average expression levels (RPKM) of randomly selected genes (left, $n = 2068$) and GDGs (right, $n = 2068$) following Fosl2 suppression. **d** Heatmaps and enrichment plots showing normalized read densities of Fosl2 CUT&Tag signals for randomly selected genes and GDGs at the different maturation phases. Tracks are centered at the TSS and extend ±3 kb. **e** Box plots displaying the average expression levels (RPKM) of randomly selected genes (left, $n = 2068$) and GDGs (right, $n = 2068$) in distal genomic regions under different maturation phases. In **a**–**c** and **e**, the $p$ value was generated from a two-sided Wilcoxon rank-sum test. Boxplot summary statistics are: center line: median; upper/lower hinges: 75th and 25th percentiles, upper and lower whiskers represent the data extending from the hinge to at most 1.5 times the interquartile range. **f** Heatmaps and enrichment plots showing normalized read densities of ATAC-seq signals for randomly selected genes and GDGs after Fosl2 knockdown. Tracks are centered at the TSS and extend ±3 kb. **g** IGV views displaying the Fosl2 CUT&Tag under different maturation phases, ATAC-seq and RNA-seq signals following Fosl2 suppression in representative *Fshr*, *Mapk1*, *Fto* and *Gapdh* from GDGs (left) and randomly selected genes (right) subsets. **h** Immunofluorescence staining of the aforementioned genes after Fosl2 knockdown in pGC. The position of the nucleolus is indicated by DAPI staining. Results shown are representative of $n = 3$ biologically independent experiments with similar results. Scale bar, 40 μm.

the absence of Fosl2 can lead to a significant impairment in the reproductive potential of female mice.

In order to further investigate the mechanism of Fosl2 on fostering follicular maturation, we isolated ovaries from sexually mature *Fosl2^cKO* and their corresponding female controls ($n = 2$/group), and performed single-cell RNA sequencing (scRNA-seq) to examine the transcriptional landscape after Fosl2 loss (Supplementary Fig. 7c). Following dimensionality reduction and clustering using the Seurat algorithm, we identified multiple clusters from previous ovary studies, which were combined to represent major cell categories in ovaries (Supplementary Fig. 7d, e)[56–59]. Focusing on the specific deletion of Fosl2 in GCs, we explored the cellular heterogeneity within Fosl2 impact. GCs were categorized into six subclusters, including the main GC subclusters, mural and cumulus GCs (Fig. 7f)[60]. Distinct gene expression programs were identified in the GCs subclusters, as visualized in the heatmap (Supplementary Fig. 8a, b). Although the total captured cell numbers were consistent at the beginning, the number of GC subclusters significantly decreased after Fosl2 loss (Supplementary Fig. 8c). In particular, the mural GC subpopulation, dedicated to endocrine function, showed a notable decrease in proportion after Fosl2 deletion, whereas the count of atretic GCs rose significantly; concurrently, the proportion of cumulus GCs, responsible for providing nutritional support to the follicles, also slightly increased, confirming the subcluster dynamics of GCs were disrupted following the deletion of Fosl2 (Fig. 7g)[61]. This disordered GC diversity directly resulted in a sharp decrease in oocyte number during superovulation, consistent with the ovulatory anomalies in oocytes observed in the scRNA-seq (Supplementary Fig. 8d). We also employed *Inhba* and *Ghr*, as specific markers for mural GCs and atretic GCs, respectively, to visually demonstrate the distribution and expression in ovaries (Fig. 7h)[62–64]. After identifying the GC subclusters, we analyzed the differential gene expression in mural GCs and atretic GCs post-Fosl2 depletion using Ingenuity Pathway Analysis (IPA) software to deduce biological functions. Notably, mural and atretic GCs both exhibited age-dependent upregulated proinflammatory stress pathways, contrasting with a concurrent downregulated developmental pathway (Fig. 7i). These results shed light on Fosl2's potential role in ovarian aging. Indeed, in 7-month-old female mice lacking Fosl2, the ovaries exhibited severe organ disorder, including vacuolation and reduced size, in stark contrast to the normal morphology of ovaries in the control group. indicating a potential complete loss of ovarian function due to Fosl2 deficiency (Fig. 7j). Moreover, the number of CL in ovaries of 7-month-old female mice significantly decreased following the deletion of Fosl2, whereas there was no significant difference in younger ovaries (Fig. 7k and Supplementary Fig. 8e). In summary, our findings reveal the essential role of Fosl2 in maintaining balanced GC subclusters; its loss causes ovulatory anomalies and heightened the risks of ovarian aging, with a more severe impact in older ovaries compared to younger ones.

## Discussion

GC transcriptomic diversification represents a shift from proliferation to differentiation status, thus maintaining the hormonal balance and follicular maturation, which are required for reproductive health; however, dysfunctional programming can trigger reproductive issues. Despite the intricate interplay between epigenetic mechanisms and transcriptomic programs during follicular maturation, our knowledge of these processes remains limited. In this study, we investigated the significant chromatin remodeling that occurs at different follicular maturation phases in mammals. Newly activated GAAs were closely linked to adjacent GDGs, and importantly, Fosl2 bound to these GAAs, prompting a shift from closed to open chromatin, which in turn facilitated downstream signal transduction. Aberrant Fosl2 expression induced a collapse of the open chromatin state, and moreover, targeted deletion of Fosl2 in GCs disrupted their normal developmental diversity, resulting in reproductive disorders, including abnormal ovulation and ovarian aging (Fig. 8). These findings suggest that Fosl2-maintained chromatin state alterations are a key mechanism in follicular maturation. We also highlight the effectiveness of combining genome-wide epigenetic analyses with molecular studies to uncover previously unrecognized variations in folliculogenesis.

GCs are fundamental to the reproductive processes, serving not only as nutritional support cells but also as endocrine function cells. It is crucial to elucidate the underlying mechanisms that govern signal transduction and follicular maturation in GCs; however, the upstream TF regulators during follicular maturation remain poorly characterized. The high-resolution genome-wide profiling of chromatin accessibility and nucleosome positioning has significantly advanced our comprehension of linear gene function and identified regulatory element candidates across the genome[41]. Importantly, open chromatin spatial arrangements at enhancers and promoters have highlighted key factors governing physical interactions in the genome. AP-1 binds to specific target-gene promoters, transforming extracellular signals into gene expression changes, e.g., our Fosl2 CUT&Tag analyses strongly supported previously identified regulatory elements, particularly promoter regions marked by ATAC-seq and histone methylation, as they interacted with Fosl2, while nearby GDGs were preferentially downregulated after Fosl2 suppression (Figs. 3f and 5d)[65]. The sustained correlations between chromatin accessibility and gene expression suggest an underlying regulatory paradigm whereby dynamic chromatin reconfiguration dictates specific expression patterns. Therefore, understanding chromatin-reshaping mediated by Fosl2 activation, which contributes to GDG expression, can help us unravel regulatory pathways that govern physiological and behavioral aspects in ovulation and folliculogenesis.

c-Fos and c-Jun TFs constitute the majority of mammalian AP-1 TFs; however, most research has concentrated on highly expressed TFs with a minimal focus on poorly expressed TFs, such as Fosl2[66–68]. During embryonic development, Fosl2 expression occurs at late

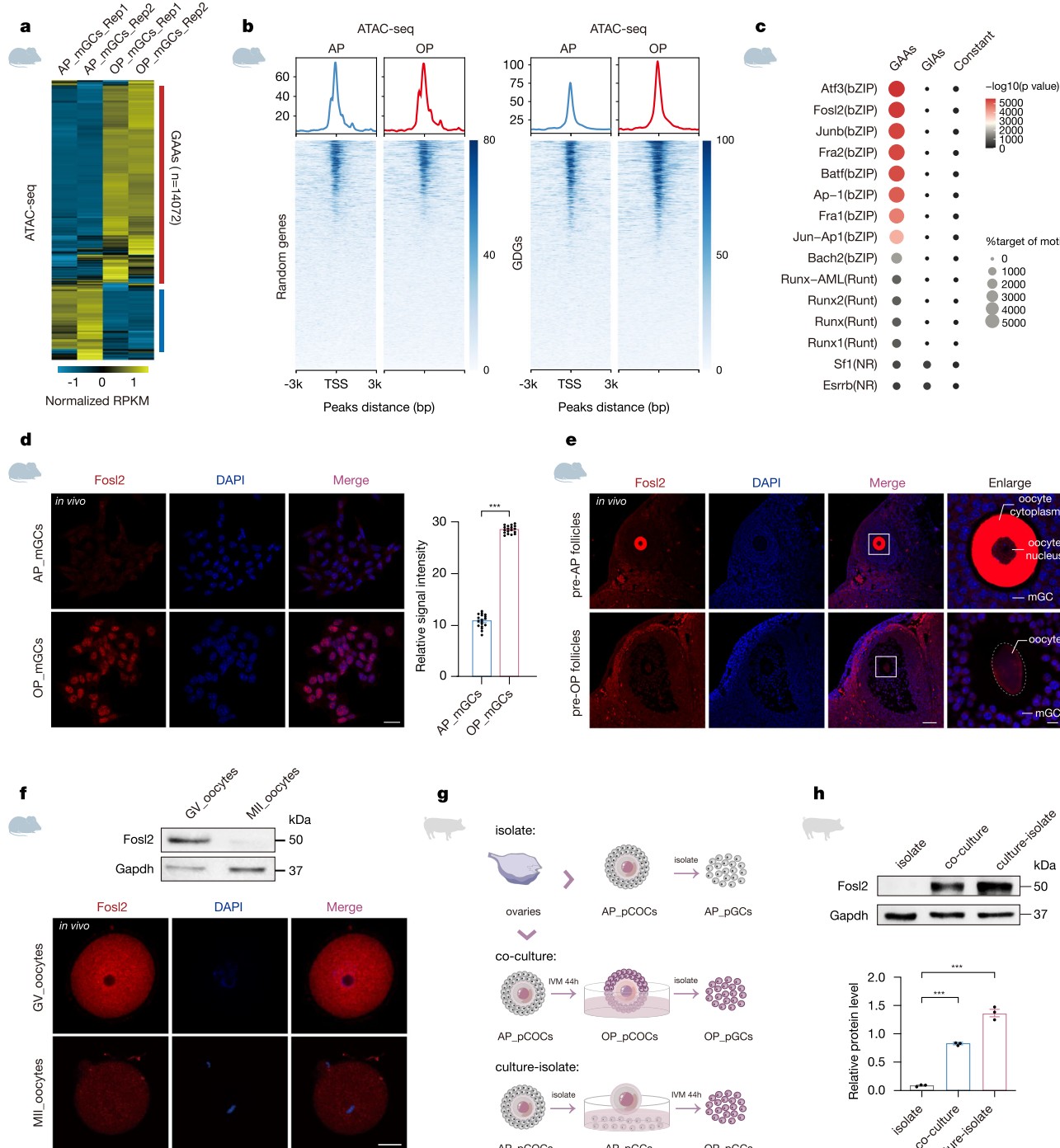

**Fig. 6 | Fosl2-driven chromatin accessibility is conserved in mammals.**
**a** Heatmap of ATAC-seq signals in murine GCs (mGCs) showing the accessibility remodeling of GAAs and GIAs. Numbers of differential accessibility regions are labeled. **b** Heatmaps and enrichment plots showing normalized read densities of ATAC-seq signals for randomly selected genes and GDGs under different maturation phases in mGCs. Tracks are centered at the TSS and extend ±3 kb. **c** Top-ranked enriched motifs among GAAs and GIAs defined in mGCs are listed, as determined using HOMER algorithms. The circle size represents the percentage of motifs in the target regions, and the color represents the *p* value. The *p* value was generated from a one-sided Fisher's exact test. **d** Immunofluorescence staining and quantification of Fosl2 in mGCs both at the antral and ovulatory phases. The position of the nucleolus is indicated by DAPI staining. Scale bar, 25 μm. Error bars indicate the mean ± S.E.M. (*n* = 3 biological replicates). The *p* value was generated from a two-sided Student's *t* test. ****p* < 0.001. Source data are provided as a Source Data file.

**e** Immunofluorescence staining of Fosl2 during murine pre-antral and pre-ovulatory phases of follicles. The enlarged views showing the staining oocytes and mGCs. The position of the nucleolus is indicated by DAPI staining. Results shown are representative of *n* = 3 biologically independent experiments with similar results. Scale bar, 60 μm and 10 μm, respectively. **f** Immunoblotting analysis (upper) and immunofluorescence staining (lower) of Fosl2 expression in murine oocytes under different maturation phases. Gapdh served as the loading control. The position of the nucleolus is indicated by DAPI staining. Results shown are representative of *n* = 3 biologically independent experiments with similar results. Scale bar, 20 μm. **g** Schematic diagram showing the three culture strategies of pGCs. IVM in vitro maturation. **h** Immunoblotting analysis and quantification of pGCs under different culture strategies. Gapdh served as the loading control. Error bars indicate the mean ± S.E.M. (*n* = 3 biological replicates). The *p* value was generated from a two-sided Student's *t* test. ****p* < 0.001.

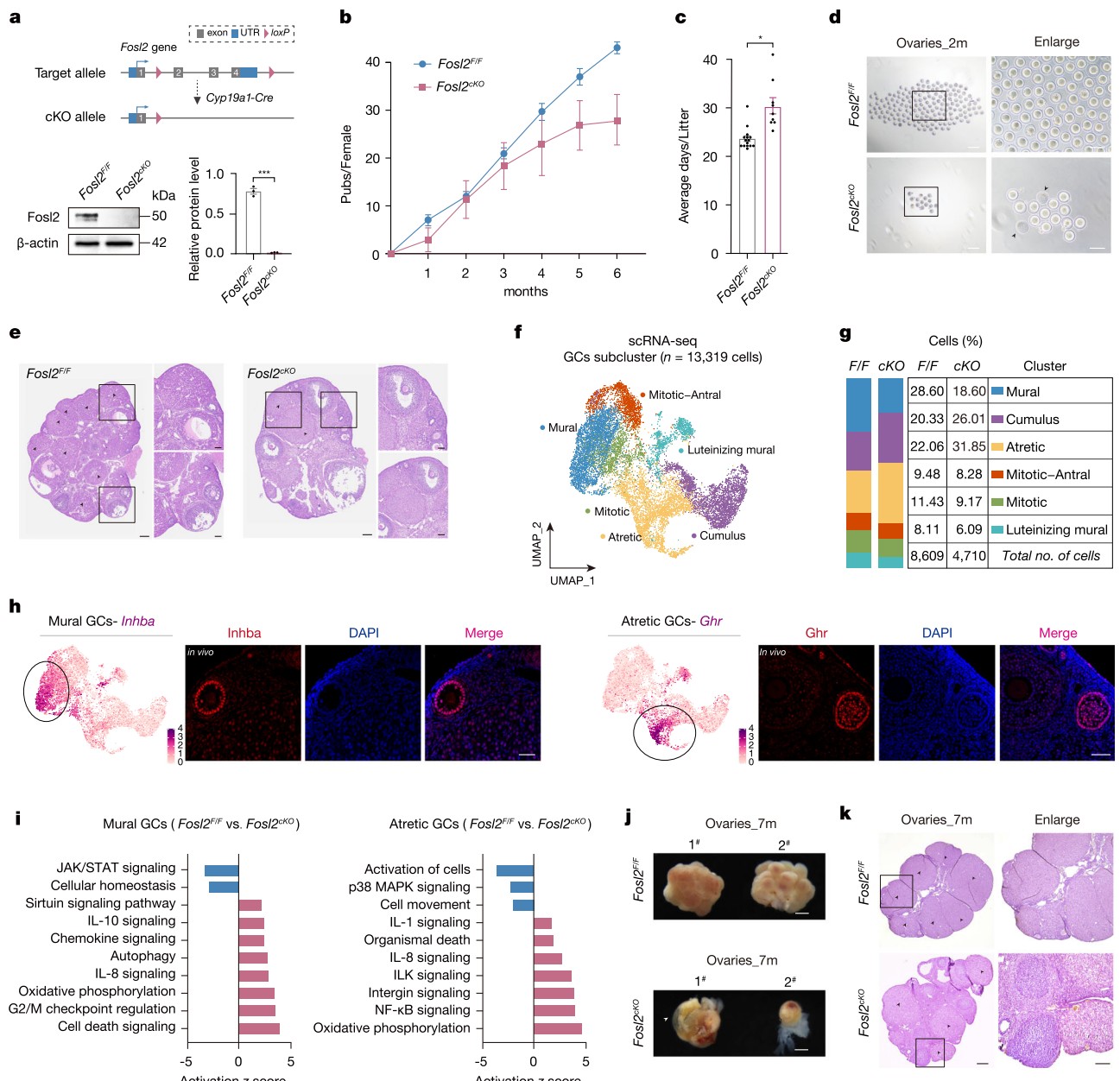

**Fig. 7 | Altered ovarian cellularity in *Fosl2ᶜᵏᵒ* GCs results in reproductive dysfunction. a** Schematic diagram showing the construction of *Fosl2ᶠˡᵒˣ/ᶠˡᵒˣ* (*Fosl2ᶠ/ᶠ*) and generation of *Cyp19a1-Cre*; *Fosl2ᶠˡᵒˣ/ᶠˡᵒˣ* (*Fosl2ᶜᵏᵒ*) mice (upper). LoxP sites (pink triangles) flank exons 2 and 4. Immunoblotting analysis and quantification showing Fosl2 protein depletion in mGCs of *Fosl2ᶜᵏᵒ* mice (lower). β-actin served as the loading control. Error bars indicate the mean ± S.E.M. (*n* = 3 biological replicates). The *p* value was generated from a two-sided Student's *t* test. ***$p < 0.001$. Source data are provided as a Source Data file. **b** Line plots showing the cumulative number of pups per female for 6 consecutive cohabitation months in breeding assays. Error bars indicate the mean ± S.E.M. (*n* = 3 biological replicates). **c** Box plots showing the average days per litter. Error bars indicate the mean ± S.E.M. (*Fosl2ᶠ/ᶠ*, *n* = 15 biological replicates, *Fosl2ᶜᵏᵒ*, *n* = 9 biological replicates). The *p* value was generated from a two-sided Student's *t* test. *$p < 0.05$. **d** Bright field images showing the number of oocytes after superovulation in 2-month-old *Fosl2ᶠ/ᶠ* and *Fosl2ᶜᵏᵒ* female mice. Arrowheads in enlarged views indicate the dead oocytes. Scale bar, left, 200 μm; right, 100 μm. **e** Histological images of ovaries of 2-month-old *Fosl2ᶠ/ᶠ* and *Fosl2ᶜᵏᵒ* female mice after superovulation. The enlarged views show higher magnification of follicles. Arrowheads indicate existing corpora lutea (CL) after

ovulation. Results shown are representative of *n* = 3 biologically independent experiments with similar results. Scale bar, left, 100 μm; right, 40 μm. **f** Uniform manifold approximation and projection (UMAP) plot featuring different cell subclusters belonging to the GC clusters (specific subcluster circled in each UMAP) in 2-month-old female mice. Clustering analysis revealing 6 distinct GC populations. **g** Numbers and percentages of GC subclusters in *Fosl2ᶠ/ᶠ* and *Fosl2ᶜᵏᵒ* mice. **h** Validation of the identity of GC subclusters, mural GCs and atretic GCs, by UMAP plots and corresponding immunofluorescence staining in *Fosl2ᶠ/ᶠ* and *Fosl2ᶜᵏᵒ* mice. Results shown are representative of *n* = 3 biologically independent experiments with similar results. Scale bar, 50 μm. **i** IPA canonical pathways indicating activation or inhibition of specific pathways in mural GCs (left) and atretic GCs (right). **j** Representative images of the ovaries from 7-month-old *Fosl2ᶠ/ᶠ* and *Fosl2ᶜᵏᵒ* female mice, Scale bar, 500 μm. Arrowheads indicate ovarian vacuolation. **k** Histological images of ovaries of 7-month-old *Fosl2ᶠ/ᶠ* and *Fosl2ᶜᵏᵒ* female mice. The enlarged views showing higher magnification of CL. Arrowheads indicate existing CL. Results shown are representative of *n* = 3 biologically independent experiments with similar results. Scale bar, left, 250 μm; right, 100 μm.

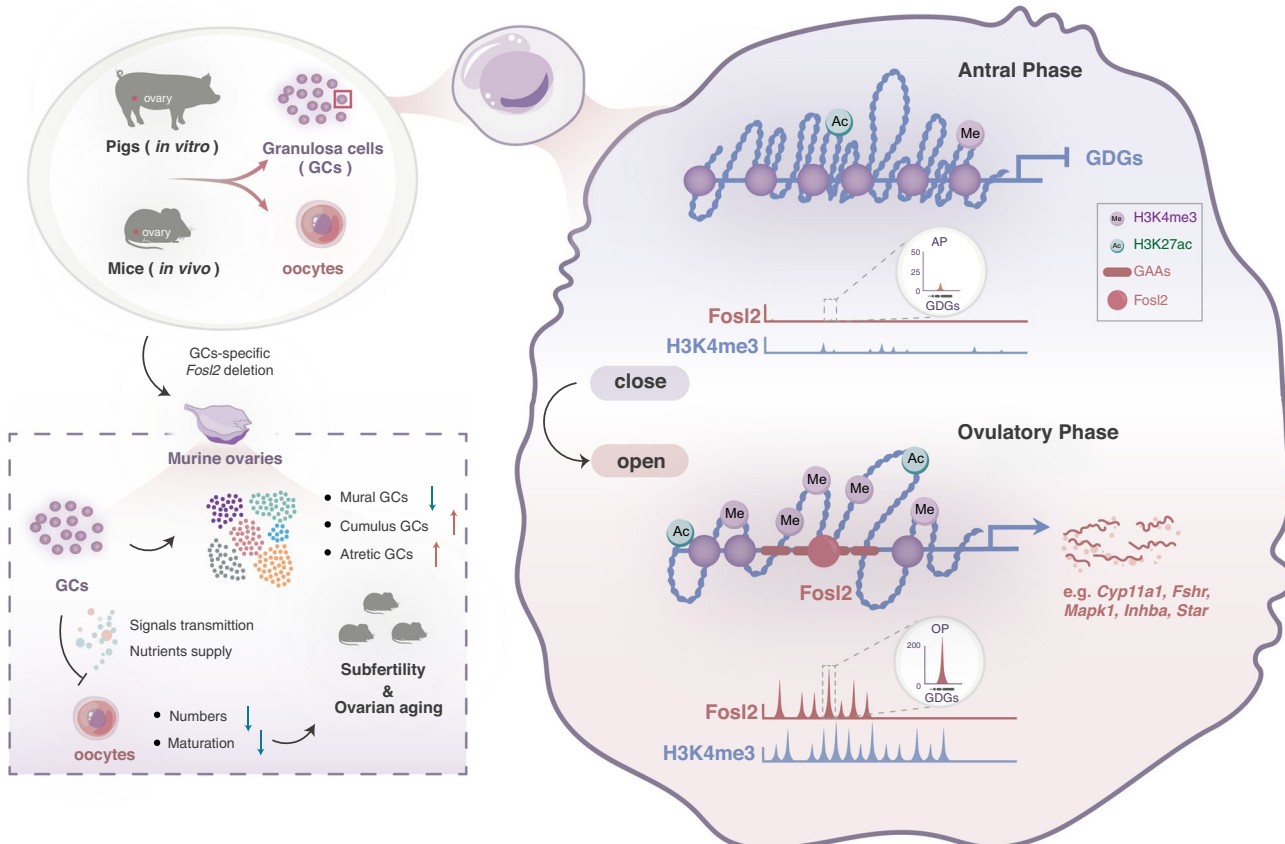

**Fig. 8 | Fosl2 maintains chromatin accessibility to settle the GC lineage commitment in follicular maturation.** During follicular maturation, newly GC-activated accessibility regions (GAAs) defined at the ovulatory phase, located within H3K4me3-decorated promoter regions, strongly induce expression of adjacent GC-involved developmental genes (GDGs). In particular, Fosl2 is recruited to these GAAs, amplifying their strength and ensuring the GDG expression, as well as the downstream developmental process during follicular maturation. Moreover,

the GC-specific knockout approach for Fosl2 leads to an abnormal composition of GC subclusters, which results in the inability to provide essential support to the oocyte. These reproductive disorders stemming from the loss of Fosl2 become particularly pronounced as time progresses. Therefore, Fosl2-driven chromatin accessibility is a pivotal epigenetic mechanism that underpins the intricate process of follicular maturation, ensuring the continuity and health of the reproductive system.

organogenesis stages, including the ovaries[69,70]. Even so, research has illustrated that mice lacking Fosl2 typically do not survive beyond 1 week after birth[71]. Nevertheless, there are only a few studies that have used mice with tissue-specific expression of Fosl2. We specifically knocked out Fosl2 in ovarian GCs, and observed that Fosl2 significantly impacted reproduction, primarily due to ovarian dysfunction (Fig. 7b, c). This was mainly attributed to altered GC subclusters, as Fosl2 is crucial for cell growth and differentiation and balancing signals both inside and outside cells[72]. Intriguingly, it is precisely the composition of mural GCs responsible for communication with external signals that undergoes an obstruction in Fosl2 deletion mice (Fig. 7g). Meanwhile, although the overall GCs collapse after Fosl2 deletion, the proportion of atretic and cumulus GCs has been elevated, which led to a reduction in the nutritional support to oocytes (Fig. 7g and Supplementary Fig. 8c). Bidirectional communications between an oocyte and GCs are necessary for the acquisition of maturation and early embryonic developmental competence following fertilization. Understanding Fosl2-driven function during follicular maturation may be helpful for predicting oocyte quality and subsequent embryonic development competence, as well as pregnancy outcomes in the field of infertility treatment.

Of note, the expression of c-Fos TFs in the ovary is required for ovulation and folliculogenesis[73]. Human GC studies have reported that c-Fos is critical for prostaglandin biosynthesis, which is crucial for ovulation[73,74]. In c-Fos-deficient mice, follicular development was impeded, while the ovaries lacked antral follicles and CL, consistent

with the impaired ovaries in our conditional knockout model (Fig. 7e)[75]. In polycystic ovary syndrome, diminished c-Fos activity has been linked to the etiology of hyperandrogenism[76,77]. c-Fos was also implicated in exacerbating ovarian dysfunction in premature ovarian insufficient mice, and exhibited decreased expression during ovarian aging, mirroring our data in GC-specific Fosl2 null ovaries, and highlighting significant c-Fos function during ovarian development (Fig. 7j, k)[78,79]. Furthermore, the potential compensatory roles of c-Fos family members that share structural homology with Fosl2 (e.g., Fosl1) during follicular maturation merit consideration. For example, the absence of severe fertility defects in GC-specific Fosl2 knockout mice may reflect functional redundancy through dosage compensation by Fosl1, given their conserved DNA-binding domains. Nevertheless, deletion of Fosl2 in mice results in lethality and affects the development of the heart, bones, and intestines, highlighting its critical and non-redundant regulatory role in maintaining overall embryonic and postnatal health[71]. This suggests that while compensatory mechanisms may mitigate acute fertility loss, Fosl2 exerts unique temporal or context-dependent control over pathways critical for ovarian homeostasis, underscoring its indispensable pathophysiological relevance. From another perspective, these TFs participate in chromatin remodeling, suggesting the potential functional interplay and combinatorial effects across different developmental contexts, aligning with the enrichment of AP-1 TFs at GAAs in our study (Fig. 3a)[80–83]. Our findings highlight the dynamic compensatory mechanisms modulating nucleosome occupancy, thereby defining developmental outcomes in the ovaries, but

more importantly, nucleosomal occupancy patterns are profoundly influenced by temporal factors, as evidenced by Fosl2 influences on aged ovaries, which were significantly different from younger ovaries (Fig. 7k and Supplementary Fig. 8e). Despite having excluded other AP-1 TFs in shaping the GAA landscape in terms of follicular maturation, it remains unclear if these TFs, particularly c-Fos, act as drivers by binding in GAAs or synergizing with other factors to reshape other GDG subsets. The functions of these TFs are only poorly understood in terms of follicular maturation, and will be an exciting topic for future studies.

Finally, our findings reveal that several immune response pathways mediated by the senescence-associated secretory phenotype (SASP) are enriched within altered GC subclusters. This suggests a reconfiguration of specific SASP genes and a link to ovarian aging following Fosl2 silencing in GCs (Fig. 7i). Ovarian aging manifests as reproductive decline, and subfertility is one of the earliest clinical signs in the cascade of events associated with reproductive aging. Our data from Fosl2-null ovaries corroborate the existing criteria for ovarian aging markers, covering both physiological and histological dimensions[84]. Therefore, future research needs to characterize cellular changes, particularly SASP-related cytokine profiles in accordance with SenNet guidelines, to fully elucidate the Fosl2 impact on ovarian aging[85].

## Methods

### Ethics statement
This study was meticulously conducted in strict compliance with the ethical protocols approved by the Institutional Animal Care and Use Committee (IACUC) of Central People's Hospital of Zhanjiang, as authorized under Permission No. ZJDY2023-08.

### Mice
Wild-type C57BL/6J mice and *Cyp19a1-Cre* mice on a C57BL/6J genetic background were purchased from the Model Animal Research Center of Nanjing University (Nanjing, China). Floxed *Fosl2* mice, *Fosl2^{Flox/Flox}* (*Fosl2^{F/F}*), were purchased from GemPharmatech Corporation (Jiangsu, China). *Cyp19a1-Cre* males were initially mated with *Fosl2^{F/F}* females to generate males heterozygous for the Fosl2 floxed allele *Cyp19a1-Cre; Fosl2^{+/Flox}*. Subsequently, these males were bred with *Fosl2^{F/F}* females to produce female offspring with the *Cyp19a1-Cre; Fosl2^{Flox/Flox}* (*Fosl2^{cKO}*) genotype, and designated as *Fosl2* conditional knockout mice. Genomic DNA isolation and genotype PCR were conducted using tail biopsies. The specific primers employed for PCR are detailed in Supplementary Data 1. All mice were maintained at the Central People's Hospital of Zhanjiang SPF Laboratory Animal Center. Mice were housed in a SPF animal facility with controlled environmental conditions (20–25 °C, 50–70% relative humidity, 12-h light-dark cycle) and free access to food and water.

### The acquisition of granulosa cells (GCs) and oocytes
**Collection of porcine cumulus-oocyte complexes (pCOCs).** Fresh ovaries from the slaughterhouse were transported to the laboratory within a 2-h window, preserved in a vacuum flask maintained at 30–35 °C. Ovaries with smooth surfaces, indicative of health and full development, were devoid of corpora lutea and possessed a substantial number of evenly distributed, fully matured luminal follicles. Any adjacent adipose tissue, oviduct, and bursa were meticulously excised. Follicles within the diameter range from 3 to 6 mm were carefully aspirated using an 18-gauge needle attached to a 10-ml syringe. Oocytes enclosed by multiple layers of tightly packed GCs and exhibiting homogeneously dark cytoplasm were chosen for maturation. pCOCs were then subjected to three rinses in HEPES-buffered Tyrode's medium, supplemented with 0.01% PVA within an incubator at 37 °C. Subsequently, the pCOCs were transferred into 0.1% hyaluronidase solution for 5 min. This treatment facilitated the dispersion of GCs, allowing for the isolation of GCs and oocytes at the antral phase.

**Maturation of porcine GCs (pGCs) and oocytes.** To generate pGCs and oocytes at the ovulatory phase, the pCOCs were washed three times in the in vitro maturation (IVM) medium (Gibco, 3110-035) supplemented with 0.1% polyvinylalcohol (Sigma, P8136), 0.026 M sodium bicarbonate (Sigma, S5761), 3.05 mM D-glucose (Sigma, G7021), 0.91 mM sodium pyruvate (Sigma, P4562), 1 mg/mL gentamicin (Sigma, G1264), 10% porcine follicular fluid, 0.57 mM cysteine (Sigma, C7352), 0.5 mg/mL luteinizing hormone (Sigma, L5269), 0.5 mg/mL follicle-stimulating hormone (Sigma, F2293), and 10 ng/mL epidermal growth factor (Sigma, S4127). Approximately 70–80 pCOCs were then placed into individual wells (NEST, 702001), each well containing 500 µL of IVM medium overlaid with 400 µL of mineral oil. The plates were cultured for 42–44 h at 39 °C and 5% $CO_2$ to facilitate maturation. Ovulatory-phase pGCs were removed from pCOCs by a brief 5-min exposure to 0.1% hyaluronidase solution. The resulting MII oocytes, characterized by the presence of an extruded first polar body, were individually inspected under stereoscopic microscopy (Nikon, Japan) for further use.

**The acquisition of murine GCs (mGCs) and oocytes.** Eight-week-old female mice post-sexual maturity were chosen for their fully developed reproductive systems and stable hormone levels. They were subjected to an intraperitoneal injection of 10 units of PMSG (Sigma-Aldrich, 9002-70-4) to induce superovulation. After 40–42 h, the mice were euthanized, and their ovaries were aseptically harvested and immediately transferred into M2 medium enriched with 200 µM IBMX to maintain the prophase I arrest of oocytes. The large follicles were delicately punctured using a fine-gauge syringe needle, facilitating the extrusion of murine COCs (mCOCs). The mCOCs were then meticulously transferred to a fresh drop of M2 medium containing 1 mg/mL hyaluronidase, an enzyme that effectively digests the hyaluronic acid-rich matrix of the mGCs. Once the mGCs were dispersed, the mGCs at the antral phase and oocytes at the GV stage were isolated and collected for further study.

For the acquisition of mGCs at the ovulatory phase and oocytes at the MII stage, mice were injected intraperitoneally with PMSG. After 46–48 h, the mice received a subsequent intraperitoneal injection of hCG (Sigma-Aldrich, 9002-61-3) to trigger ovulation. Fourteen hours post-hCG administration, superovulated mice were euthanized, and mCOCs were retrieved from the ampullae of the oviducts, collected in M2 medium containing 1 mg/mL hyaluronidase. Once the mCOCs were digested, the mGCs and oocytes were carefully isolated and collected for further study.

### Histological analysis and immunofluorescence staining
Eight-week-old *Fosl2^{F/F}* and *Fosl2^{cKO}* female mice from the same litter were euthanized to collect ovarian tissues. The harvested ovaries were promptly fixed in 4% PFA at 4 °C overnight. Following fixation, tissues were stored in 70% ethanol for embedding in paraffin for histological processing. Next, 5 µm tissue sections were prepared using a microtome and mounted onto glass slides. The sections underwent a standard deparaffinization and rehydration protocol.

For immunofluorescence staining, the slides were incubated with primary antibody (Fosl2, CST, 19967; Fshr, Proteintech, 22665; Mapk1, Proteintech,51068-1-AP; Fto, Proteintech, 27226-1-AP; Inhba, Bioss, bs-1774R; Ghr, Bioss, bs-0654R) at 4 °C overnight. The following day, the slides were washed and incubated with the secondary antibody for 1 h. The nuclei were counterstained with DAPI for 10 min to visualize nuclear morphology. Images were captured using a laser scanning confocal microscope (Nikon, Japan) and were processed using ImageJ (Adobe, USA).

## Western blotting

Proteins were extracted by using RIPA buffer (Thermo Fisher, 89901). The extracted proteins were subjected to denaturation in a 5×SDS loading buffer (Beyotime, P0015L) at 100 °C for 10 min. The denatured proteins were resolved by 10% SDS polyacrylamide gel electrophoresis, and the resolved proteins were transferred onto PVDF membranes (Millipore, IPVH00010) for immunoblotting analysis. To minimize non-specific antibody binding, the PVDF membranes were blocked in a solution containing 5% skimmed milk for 1 h. The membranes were incubated with primary antibodies (Fosl2, CST, 19967; Gapdh, Proteintech, 10494-1-AP; β-actin, Proteintech, 20536-1-AP) at 4 °C overnight. After extensive washing, the membranes were incubated with appropriate HRP-conjugated secondary antibodies and were visualized using an iBright™ instrument (Thermo Fisher, USA).

## RNA interference (RNAi)

For the knockdown experiment, two small interfering RNA (siRNA) against Fosl2 from GenePharma were transfected into pGCs. Briefly, pGCs were seeded into 24-well plates in DMEM-F12 medium supplemented with 10% FBS, 1 µg/mL of β-estradiol, 0.22 mM sodium pyruvate, 10 IU/mL of PMSG, 10 IU/mL hCG and 10 ng/mL EGF. *Fosl2* siRNA (*Fosl2* siRNA-1: 5′-CCAGUCAUCAGACUCCUUGAA-3′ *Fosl2* siRNA-2: 5′-CGAACCUCGUCUUCACCUA(dT)(dT)-3′) transfections were performed using lipo3000 (Invitrogen, L3000015) following the manufacturer's instructions. After 48 h, the cells were collected for further study.

## Estrus cycle examining

The estrus stage of mice was determined by vaginal smears. In brief, around 20 µL of normal saline was used for vaginal lavage. At the vaginal canal opening, a pipette tip was employed to repeatedly flush and aspirate. The final saline was placed on a coverglass, smeared, and then stained with crystal violet according to the manufacturer's instructions (Beyotime, C0121). Proestrus was identified by the presence of numerous nucleated cells, and estrus by cornified epithelial cells. Diestrus and metestrus were distinguished by the presence of nucleated epithelial cells in diestrus, along with leukocytes.

## Bulk RNA assay

Total RNA was extracted from either 50 oocytes or $5 \times 10^4$ GCs using TRIzol reagent (Invitrogen, 15596018), following the manufacturer's protocol. The concentration, purity and integrity were determined using NanoDrop 2000 (Thermo Fisher, USA) and Agilent Bioanalyzer 2100 systems (Agilent, USA), respectively. For RNA-qPCR assay, 1 µg total RNA was reverse-transcribed into cDNA with random hexamer DN6 and M-MLV reverse transcriptase. A 10 µL reaction system comprising 2 µL of the diluted reverse-transcribed product (1:50) and 8 µL of SYBR Green I Master Mix (Roche, 04707516001) was evaluated using a LightCycler 480 instrument (Roche, Switzerland). The specific primers employed for RNA-qPCR are detailed in Supplementary Data 1. For RNA-seq analysis, after mRNA isolation, fragmentation and priming, cDNAs were subjected to RNA library prep kit for Illumina (NEB, E7770). All libraries were subjected to paired-end sequencing on the Novaseq 6000 platform (Novogene, China) to a depth of at least $1.5 \times 10^7$ reads.

## ChIP-qPCR assay

In brief, $1 \times 10^6$ GCs were fixed by crosslinking with 1% formaldehyde solution, followed by quenching with glycine to terminate the fixation. Chromatin shearing was achieved using Bioruptor (Diagenode, Belgium), resulting in fragments ranging from 300 to 500 bp. A total of 3 µg of primary antibody (H3K4me3, Abcam, ab8580; H3K27ac, Abcam, ab4729; IgG, Proteintech, 30000-0-AP) was incubated with a beads-DNA mixture at 4 °C overnight. The following day, antibody-protein-DNA complexes were immunoprecipitated using pre-treated

Protein A Sepharose bead slurry (Sigma-Aldrich, GE17-5280). The eluted complexes underwent extensive washing and de-crosslinking, and the ChIP-enriched DNA was purified using phenol-chloroform extraction for subsequent qPCR analysis. The specific primers employed in ChIP-qPCR assay are listed in Supplementary Data 1.

## scRNA-seq library preparation

Eight-week-old *Fosl2^{F/F}* and *Fosl2^{cKO}* female mice from the same litter were euthanized to collect ovarian tissues, and all samples were processed in parallel by a single operator within a 24-h period to eliminate technical variability. The single-cell library was meticulously prepared using the Chromium Single Cell 3′ Library and Gel Bead Kit v3 (10×Genomics, 120237), strictly following the manufacturer's protocol. Briefly, cell viability was confirmed to be >70% in ovarian tissues from Fosl2-deficient and control mice using trypan blue exclusion. Cell suspensions were filtered using a 70 µm strainer, and dissociated cells were pelleted and re-suspended with 0.04% BSA in PBS. Single-cell suspensions were loaded on the Chromium Controller (10×Genomics, USA) to generate Gel Bead-In-Emulsions (GEMs). Adhering to the manufacturer's protocol, scRNA-seq libraries were constructed to paired-end sequencing by the Novaseq 6000 platform to a depth of at least $3.5 \times 10^8$ reads accordingly.

## ATAC-seq library generation

The ATAC-seq procedure was performed as previously described with minor modifications[86]. In brief, 50 zona pellucida-free oocytes were lysed in a lysis buffer (10 mM Tris-HCl (pH 7.4), 10 mM NaCl, 3 mM MgCl₂, 0.1% Igepal CA-630, 0.1% Tween-20, 0.01% Digitonin, 1% Cytochalasin B, 0.1% collagenase I) for 10 min on ice to generate the nuclei. Meanwhile, GC nuclei were isolated from $5 \times 10^4$ cells using the efficient nuclear separation kit (SHBIO, 52009). The nuclei were then centrifuged at $500 \times g$ for 5 min to pellet and remove the supernatant. Tn5 transposome and tagmentation reactions were carried out at 37 °C for 30 min (Vazyme, TD501). After the tagmentation, DNA was purified using $2 \times$ AMPure beads (Beckman Coulter, A63881). Library amplification was performed for 16 cycles for oocytes and 14 cycles for GCs using the following PCR conditions: 72 °C for 3 min, 98 °C for 30 s, and thermocycling at 98 °C for 15 s, 60 °C for 30 s and 72 °C for 3 min; concluding with a final extension at 72 °C for 5 min. The amplified DNA was size-selected for 200–500 bp fragments using AMPure beads. All libraries were subjected to paired-end sequencing on the Novaseq 6000 platform to a depth of at least $4.0 \times 10^7$ reads.

## CUT&Tag library generation

The CUT&Tag was conducted using CUT&Tag Assay Kit for Illumina according to the manufacturer's protocol with minor modifications (Vazyme, TD903). Briefly, nuclei were isolated from $1 \times 10^4$ GCs using the efficient nuclear separation kit and incubated with ConA beads for at least 10 min to achieve a comprehensive combination. The antibody-protein-nuclei complexes were then incubated with primary antibodies (H3K4me3, Abcam, ab8580; H3K27ac, Abcam, ab4729; Fosl2, CST, 19967) at 4 °C overnight, with an IgG antibody (IgG, Proteintech, 30000-0-AP) serving as a negative control. The following day, the samples underwent incubation with a secondary antibody, followed by the application of a diluted pA-Tn5 transposase to cleave the genome and append a unique adapter sequence, thereby constructing the library. All sequencing libraries were prepared after 12 cycles of PCR amplification. The selected DNA fragments were subjected to paired-end sequencing on the Novaseq 6000 platform to a depth of at least $2.5 \times 10^7$ reads.

## Data analysis

**Bulk RNA-seq data processing.** Quality control was performed using FastQC to assess read quality and detect potential contaminants. Subsequent trimming of low-quality reads (Phred score <20) and

adapter sequences was carried out with Trimmomatic. Cleaned reads were then aligned to mm10 or Sscrofa11.1 genome using the STAR aligner, producing aligned BAM files[87]. Gene expression was assembled and quantified using package HTseq-count, which was processed to obtain FPKM and TPM values[88]. In addition, to control for technical variability, the RUVseq package (v1.34.0) was employed to estimate and remove unwanted variation from the data, while the Ensembl Sscrofa v11.1 annotation served as the reference for defining genes used in the RUVseq analysis[89]. Principal component analysis (PCA) was conducted to assess the overall structure of the data and identify potential batch effects or biological variation[90]. Using the R package DESeq2, differentially expressed genes (DEGs) were identified based on the criteria of log2 (fold-change) ≥1 and adjusted $p$ value < 0.05[91]. The list of DEGs is provided in Supplementary Data 2 (The $p$ value was generated from a two-sided test, with adjustment for multiple testing via the Benjamini–Hochberg (BH) method). Gene set enrichment analysis was performed using the R package fgse and was visualized using the R package clusterProfiler[92,93]. The gene set terms used for this analysis were downloaded from the MSigDB database (https://www. gsea-msigdb.org/gsea/msigdb).

**Identification of GC-involved developmental genes (GDGs).** As for the identification of GDGs, genes were obtained based on RNA-seq data using staged pGCs cultured in vitro. Genes that were not expressed or lowly expressed at the antral phase, but become upregulated (at least two-fold upregulation) at the ovulatory phase were included in the list. An equal number of randomly selected genes were chosen from protein-coding genes with an FPKM > 1 in the RNA-seq data to serve as GDG controls. The list of GDGs is provided in Supplementary Data 3.

**ATAC-seq data processing.** FastQC was used to evaluate the quality of the sequencing data, and Trimmomatic was explored to remove the adapter sequences and obtain the clean data. At the same time, Q20, Q30 and GC content of the clean data were calculated. All the downstream analyses were based on the high-quality clean data. The clean reads were then aligned to the UCSC mm10 or Sscrofa11.1 reference genome using the bwa program. Using the ENCODE blacklist regions for the mm10 genomic data allows for effective filtering out of potential technical artifacts. The bam file generated by the unique mapped reads was used as an input file, using MACS2 software for callpeak with cutoff $q$ value < 0.05[94]. Data quality metrics for FRiP, NRF were calculated using deepTools (v3.5.1) and are detailed in Supplementary Data 4. Genomic features of peaks were annotated using ChIPseeker R package. To identify DEGs between differently passaged GCs, the ATAC-seq peaks of each sample were merged and the number of peaks among this set was counted using BEDTools (v2.29.0)[95]. GAAs and GIAs were identified using DESeq2, with thresholds log2 (fold-change) ≥1 and adjusted $p$ value < 0.05. The locations and descriptions of GAA regions are accessible in Supplementary Data 5. To find the sequence motifs enriched in GAAs and GIAs, findMotifsGenome.pl from the HOMER program was used[96]. Genome-wide normalized signal coverage tracks were created by bamCoverage in deepTools and were visualized in the Integrative Genomics Viewer. Promoters were defined as ±2.5 kb around the TSS. Peaks at least 2.5 kb away from TSSs were defined as distal peaks. The functional enrichment for genes that are near stage-specific distal peaks was analyzed using the GREAT tool by default settings[97]. Hierarchical clustering was performed in R by hclust () function with ChIP-seq RPKM values via Pearson correlation coefficients.

**CUT&Tag data processing.** CUT&Tag data were aligned to Sscrofa11.1 genome sequences by Bowtie with similar parameters as ATAC-seq data. Similarly, peaks were called using MACS2 peak caller with $q$ value < 0.05. Genome-wide normalized signal coverage tracks were

visualized in the IGV. Peaks that were two-fold greater at a false discovery rate <0.1 were considered differential peaks, while peaks that did not have a log2 fold change significantly different from zero were termed constitutive peaks.

**scRNA-seq data processing.** The raw sequencing data were processed with CellRanger from 10×Genomics (v3.1.0) with default parameters. Alignment was performed against the mouse genome reference sequence GRCm38. Uniquely mapped reads were retained for UMI counting, and gene expression levels were quantified for each detected barcode. CellBender was employed to process the raw feature-barcode matrices with default parameters to eliminate ambient RNA molecules from each sample[98]. The resulting clean corrected expression matrices were then analyzed using the Seurat package (v5.0.2)[99]. Quality control was initially conducted independently on each library to determine appropriate filtering thresholds, and cells were excluded if they failed to meet the criteria of having a minimum of 500 detected genes, over 1000 detected UMIs, mitochondrial UMI counts constituting less than 10% of the total, and hemoglobin gene expression levels below 0.1%. Subsequently, we used the DoubletFinder R package to identify and remove potential doublets. Data integration was performed using SCTransform with regression on mitochondrial gene percentage[100]. The clustree package (v0.5.0) was employed to visualize hierarchical relationships between clusters across resolutions. The full lists for scRNA-seq makers in each ovary cluster and GC subcluster are accessible in Supplementary Data 6 and 7, respectively (The $p$ value was generated from a two-sided Wilcoxon rank-sum test, with adjustment for multiple testing via the Benjamini-Hochberg (BH) method). Gene lists were imported into IPA_01.12 (Qiagen Bioinformatics) software and filtered on false discovery rate (FDR) < 0.1 and log(fold change) > 0.25 for pathway analyses to assess pathway/biological function enrichment analysis.

### Fertility test
For fertility testing, *Fosl2*[F/F] (control; $n = 7$) and *Fosl2*[cKO] female ($n = 7$) mice with sexual maturity were continuously mated to wild-type males (C57BL/6J, 2-month-old) with known fertility for >6 months. Litter sizes were checked daily for counting the number of pups.

### Statistics and reproducibility
Data were analyzed using Prism 8.0 (GraphPad) or the R programming environment. The comparison of the mean between two groups was conducted using Student's $t$ test, and differences between two independent groups were assessed using the Wilcoxon rank-sum test. Fisher's exact test was performed using BEDTools. The correlation between two continuous variables was calculated using Spearman's correlation and $p < 0.05$ was considered significant. Results were validated through at least three biological replicates and/or independent experimental repetitions

### Reporting summary
Further information on research design is available in the Nature Portfolio Reporting Summary linked to this article.

## Data availability
The raw sequencing and processed data generated in this study were deposited in the Gene Expression Omnibus (GEO) database, accessible using GSE267974. GEO accession codes for study data are RNA-seq, GSE267849; ATAC-seq, GSE267850; CUT&Tag, GSE267851; and scRNA-seq, GSE281100. Source data are provided with this paper.

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

## Acknowledgements

We thank Prof. Wei Tao (Peking University) and Prof. Jianguo Zhao (Institute of Zoology, Chinese Academy of Sciences) for their kind suggestions. We thank Prof. Chao Zhang (Kunming Institute of Zoology, Chinese Academy of Sciences) and Dr. Lumeng Jia (Dana-Farber Cancer Institute) for technical assistance and support. For data acquisition and processing, we thank the staff of the Cell Biology Laboratory, SPF Laboratory Animal Center, and Biomedical Imaging Platform in Zhanjiang Institute of Clinical Medicine. This work was supported by the National Natural Science Foundation of China 82301863 (H.Y.Z.) and 82201715 (Y.T.G.), the Guangdong Basic and Applied Basic Research Foundation 2024A1515012916 (Y.T.G.) and 2022A1515111164 (H.Y.Z.).

## Author contributions

H.Y.Z. and Y.T.G. conceived the project and obtained financial support. H.Y.Z. and Z.C.L. participated in sample collection. H.Y.Z. and Y.T.G. prepared the bench experiments with help from Y.M.Z. W.C.L. and H.C.W. performed the bioinformatics analysis. S.J.T. and W.Y. interpreted the statistical data. J.J.Z. and Q.Y.S. supervised the mouse work. Y.T.G. prepared the original manuscripts. H.Y.Z., Y.T.G., and W.C.L. conducted the revised bench experiments. Y.T.G. wrote the revised manuscripts. All authors approved the final version.

## Competing interests

The authors declare no competing interests.
