## [Transparent Peer Review file · Nature Communications]

Fosl2 facilitates chromatin accessibility to determine developmental events during follicular maturation

Corresponding Author: Professor Yiting Guan

Version 0:

Reviewer comments:

Reviewer #1

(Remarks to the Author)

Zhang et al. investigated the role of Fosl2 in porcine and mouse models, with a primary focus on the antral and ovulatory phases. Their study generated valuable transcriptomic and epigenetic data to assess the involvement of Fosl2. They identified accessible regions activated in granulosa cells, linking these regions to genes associated with granulosa cell development and the regulation of Fosl2. While the results appear promising, the study has significant flaws concerning the models used for the "ovulation phase" and lacks transparency in providing adequate supplementary information and standardized analytical approaches for both transcriptomic and epigenetic data. The following concerns require clarification from the authors.

Models used:

1. Maturation of porcine granulosa cells and oocytes to recapitulate the ovulatory phase

- In lines 92-95, the authors mention: "We derived porcine cumulus-oocyte complexes (pCOCs) from porcine ovaries and matured them in vitro to generate porcine granulosa cells (pGCs) and oocytes at different phases (Fig. 1a). One of the questions that needs to be addressed is how the authors optimized the conditions to achieve the highest similarity with the follicle maturation process in porcine ovaries (i.e., achieving optimal conditions for IVM).

-1.b. Even considering that they have applied the best possible conditions, there is always the challenge of achieving a transcriptome and epigenetic regulation that closely resembles natural conditions in various biological contexts. Therefore, it would be beneficial for the authors to demonstrate the extent to which variability in IVM impacts the transcriptomic and epigenetic methods used in their study.

2. Mouse superovulation model used in Figure 5.

The authors used mature, cycling mice (8 weeks old) for superovulation and ovulatory phase studies, but this model presents several challenges compared to prepubertal models. One of the primary issues is follicular asynchrony, as sexually mature mice naturally undergo estrous cycles, resulting in a heterogeneous population of follicles at various developmental stages. This variability impacts the response to PMSG and hCG stimulation, as some follicles may already be at the antral stage while others remain preantral, leading to less uniform follicular growth compared to prepubertal mice, where follicles are more synchronized. Consequently, the efficiency of superovulation in mature mice tends to be lower, as studies have reported a reduced number of ovulated oocytes due to follicular heterogeneity.

3. scRNA-seq Data from Fosl2F/F and Fosl2cKO.

For this experiment, the authors used two 8-week-old female mice for each genotype (Fosl2F/F and Fosl2cKO) from the same litter. Were these mice monitored to confirm that they were at the same stage of the estrous cycle? When mature cycling mice (8 weeks old) are used as a model for transcriptomic and epigenetic studies aimed at investigating changes during the ovulatory phase, several critical factors must be considered to ensure data reliability and biological relevance. One significant challenge is the variability in baseline gene expression, as follicular development is not synchronized. This lack of synchronization leads to transcriptomic profiles that reflect a mixture of follicular stages rather than a distinct ovulatory phase signature. Such high biological variability complicates the isolation of ovulation-specific transcriptional changes.

Additionally, heterogeneous chromatin states across follicles at different developmental stages may confound epigenetic analyses, as changes in chromatin accessibility will not be uniform, potentially obscuring ovulation-driven chromatin remodeling events. Furthermore, batch effects and sample normalization are crucial considerations, as the high variability among animals necessitates larger sample sizes and stringent normalization strategies. These factors suggest that the immature mouse model may provide a more controlled and predictable system for studying the ovulatory phase. The absence of this information, combined with the poor distinction of cell population marker expression in Figure S9, raises several questions regarding the comparative analyses used to demonstrate the effects of Fosl2 knockout (KO).

Other Comments:

- In Figure 1c, the authors present a figure comparing gene expression across different replicates and samples based on FPKM (Fragments Per Kilobase of transcript per Million mapped reads). It is recommended, however, to use Transcript Per Million (TPM) for comparing gene expression levels. While the overall expression pattern may not change, the expression values for each gene would differ and become more directly comparable.
- In lines 102–106 (“In contrast...”), the authors state that the expression of Lhx8, Sycp3, and Sox30, identified as oocyte-specific markers, is not detectable in porcine granulosa cells (pGCs). This raises two questions: First, why is it necessary to demonstrate that oocyte markers are not expressed in granulosa cells? Secondly, “detectable” is not an appropriate term in this context, as the heat map appears to be based on the z-scores of FPKM values. It is expected that a lack of visible color in the heat map does not necessarily indicate the absence of expression. To support their claim, it is recommended to include a separate graph (e.g., a bar plot) showing the actual FPKM values for these genes.
- Extended Data Fig. 3: It is uncommon to use IGV snapshots to demonstrate universal overlap. Moreover, the conclusion that the accessible regions are generally shared between ATAC-seq and CUT&Tag data cannot be made solely based on such snapshots. Both ATAC-seq and CUT&Tag data are typically processed to identify “peaks,” which represent regions of enrichment. To compare the peak lists from these methods and determine the degree of overlap, tools like Intervene (<https://bmcbioinformatics.biomedcentral.com/articles/10.1186/s12859-017-1708-7>), specifically the “intersect” feature, could be utilized. This approach would provide a more accurate assessment of how many regions are identified by both techniques. Additionally, using tools like GREAT to annotate open chromatin regions from the CUT&Tag and ATAC-seq data and overlapping their associated genes would be more meaningful than relying on an IGV snapshot.
- Lines 164-167: The authors mention ‘In agreement with GO...’, but this claim is not well documented. It is necessary to provide the GDGs list and include a few insertion tracks of well-recognized GC biomarkers to support the claim. Additionally, in Extended Figure 4d, it is unclear what data were used to show the expression or accessibility of the GDGs list.
- Extended Data Figure 5b and its relation to Fosl1 expression: First, in Figure 5b, it is not clear which type of expression value (e.g., RPKM, TPM, etc.) was used to show the change in the expression of the top genes (Atf3, Fosl2, Fosl1, Junb, Jun, Fos). Secondly, the Fosl1 expression in the heatmap of Figure 5b, from the antral to ovulatory phase, demonstrates an increased expression level. However, in Figure 5d, the relative mRNA expression of Fosl1 is decreased during the same phase transition as shown in Figure 5b. Does the author have any explanation for this discrepancy?
- Extended Figure 6c: This figure is introduced as “GO enrichments of the 1137 regions that close after Fosl2 knockdown using GREAT analysis. Bar length represents the enriched p-value for biological processes.” However, it appears that the figure is the result of motif enrichment analysis, not GO enrichment. Additionally, the meaning of the colors on the plot is not explained.
- Line 231-236, Figure 4c: the author shows Spearman’s correlation of differential accessibility peaks between follicular maturation phases and accessibility regions that changed after Fosl2 knockdown, with $r = -0.202$. The correlation coefficient is relatively close to zero, suggesting a weak association between the two variables. The weak correlation might not be biologically meaningful. The same weak correlation has been shown in Figure 3e ($r = 0.406$), which both suggests that Fosl2 knockdown has a limited impact on differential accessibility peaks; other factors besides Fosl2 knockdown might influence differential accessibility peaks during follicular maturation.
- Lines 247–250: The sentence starting with “To further investigate...” lacks clarity, and the explanation of the data and techniques used is not well articulated. It appears that the clustering of Fosl2 categories is based on CUT&Tag for Fosl2, which was then mapped to the ATAC-seq data of pGCs in both normal and KD models. If this assumption is correct, it remains unclear and undocumented why the authors stated that “only the ovulatory-specific peaks underwent extensive decreases in accessibility upon Fosl2 silencing (Fig. 4e).” There seems to be a considerable difference in AP-specific peaks compared to Fosl2-KD1 and Fosl2-KD2. Additionally, the figures on the right side of Fig. 4e are not explained, either in the text or in the figure legends, making it difficult to understand their relevance.
- The idea of focusing on GDGs (GC-involved development genes) related to GAAs is intriguing. However, certain aspects of the GDGs lack transparency. The article does not provide a clear definition of GDGs. Which groups of genes and genomic regions are classified as GDGs? Were these genes selected from a specific database, or are they derived from analyses conducted by the authors themselves?
- Considering the second section of this part of the results, the title is misleading considering the results provided. Only the first part briefly covers what the authors concluded from comparing pGCs to mice. In the second section, the authors shift

back to discussing pGCs without effectively connecting their results to the findings in mice. Thirdly, the authors attempt to demonstrate that Fosl2 replacement from the oocyte to pGCs occurs through gap junctions. However, based on the experiments and explanations provided, such a conclusion is not substantiated. If this replacement indeed occurs through gap junctions, the authors should have shown that disrupting or destroying the gap junctions would interrupt Fosl2 replacement. Instead, in their pGC co-culture strategies, they show elevated Fosl2 expression in both antral and ovulatory follicles, which does not conclusively support their claim.

- **scRNA-seq:** In the final section of the results, the authors focus primarily on the single-cell RNA-seq data of Fosl2F/F and Fosl2cKO. However, the analysis lacks transparency and documentation, and the analysis pipeline is not well standardized. The authors have not provided any gene signatures for each cluster. Regarding data processing, they mentioned filtering cells based on two criteria (mitochondrial content higher than 30% and fewer than 500 genes per cell). These criteria are not commonly used in single-cell RNA-seq analyses, and the authors did not provide a clear rationale for selecting such stringent cutoffs. These choices could significantly impact downstream analyses, such as the number of clusters or cell types identified in Fosl2F/F and Fosl2cKO. Thirdly, there is no explanation of whether batch effect correction was considered or applied to account for potential technical variability. For instance, it is unclear whether the Fosl2F/F and Fosl2cKO libraries were prepared under the same conditions or sequenced on different days or runs. Finally, the authors did not specify how they optimized the number of clusters in their analysis or which statistical tests they used to evaluate the stability and separation of the clusters. Additionally, it remains unclear how the initial clusters were identified and subsequently annotated as sub-clusters of granulosa cells. The poor distinction between clusters in the heatmap (Extended Data Fig. 9a) of the top five genes suggests a non-standardized scRNA-seq analysis pipeline or poor sequencing quality.

- **ATAC-seq and CUT&Tag Data:** In the analysis of these epigenetic datasets, the authors did not specify whether blacklist regions or unwanted reads, such as singletons and duplicates, were removed prior to analysis. Notably, high-signal regions, particularly those in blacklist regions, can introduce significant bias and directly impact the accuracy of peak calling when using tools such as MACS2. Furthermore, with respect to the ATAC-seq data, the authors reported that “the ATAC-seq peaks of each sample were merged to generate a consensus set of unique peaks” (Methods, line 671). This raises concerns regarding the reproducibility of the reported peaks, as no evidence was provided to confirm their consistency across samples. Moreover, the authors did not justify their choice of a p-value threshold of 0.05 for peak calling, nor did they evaluate stricter thresholds, such as 0.01 or 0.03, which would result in varying numbers of identified signals. The lack of optimization or validation to establish the most appropriate threshold leaves ambiguity about the reliability of the selected approach. Given that this analysis directly affects downstream results, including GAAs and GIAs, it is strongly recommended to identify reproducible peaks across biological replicates using the Irreproducibility Discovery Rate (IDR) method. This is a widely recognized and reliable method for objectively evaluating the reproducibility of high-throughput assays.

- **RNA-seq data:** The authors have not provided documentation to demonstrate the quality control (QC) of the RNA-seq data. Additionally, there are no plots, such as principal component analysis (PCA), to show the separation of samples from different follicular states (germinal vesicle and MII in oocytes, antral and ovulatory phases in porcine granulosa cells), which would indicate transcriptomic differences among these groups. It appears that the authors have applied a standard RNA-seq pipeline, but there are some concerns with this approach. Conventional normalization methods generally address sequencing depth but fail to correct for other technical biases, such as those introduced during library preparation. This issue can be more pronounced in tissues like the ovary, where hormonal levels fluctuate, and small hormonal changes can directly contribute to variation in gene expression. Therefore, I strongly recommend performing an additional normalization step (methods such as remove unwanted variation, RUVseq) after transcript quantification to remove unwanted biases and produce normalized counts, followed by differential expression analysis to identify DEGs in each comparison.

Minor issues:

- Line 37 missing reference: “In terms of reproduction, GCs...”
- It would be better to include the criteria used for determining the DEGs either directly on the figure or in the figure legend (Extended Data Figure 1a).
- Lines 106-109 (related to Figure 1d): Some references need to be added. Based on which references are the genes referred to as 'oocyte support genes'? Also, what do the dashed lines in the figure represent?
- Line 112: What does 'overexpressed DEGs' mean in the context of a typical transcriptomic comparison between two given conditions? What are those genes? Would it be possible to provide a table?
- Figure 3h: The terms 'shared' (upper panel), 'OP/AP up only,' and '7829' (lower panel) are unclear. Additionally, earlier in line 142, the number of GAAs regions is introduced as 8646, but in Figure 3h (upper panel), the GAAs-specific regions are listed as 26,577. It would be helpful if the author could clarify why there is such a significant discrepancy here. Thirdly, while the text (lines 217-220: 'To further...') clearly explains the overlap of GAAs with Fosl2-gained peaks, the figure is somewhat misleading, as Fosl2 is neither shown in the figure nor mentioned in the legend.
- The idea of randomly selected genes is not clear. In addition, how did the authors decide to select the random genes?
- Line 177: It should be 'Fshr' instead of 'frsh'.
- For the motif analysis, there is a discrepancy: line 189 mentions the HOMER algorithm, whereas the legend for Figure 3a

refers to the HOMER2 algorithm.

- Lines 253 and 245 require references for the mention of TOBIAS tools and “unidentified pluripotency TF motifs...”.
- Extended figure 6a. It has not been mentioned what data have been used to create the heatmap.
- Figure 6d: It is not clearly indicated whether the IF figures represent antral or ovulatory follicles.
- Line 309: “a representative folliculogenesis biomarker, Mapk1” need reference.
- Line 360: Missing reference for the statement: “using the Seurat algorithm...”
- Line 421: Missing reference for the statement: “strongly supported previously identified regulatory elements...”
- Line 431: Missing reference for the statement: “however, most research has concentrated on highly expressed TFs...”
- Line 450: Missing reference for the statement: “Of note, the expression of c-Fos TFs in the ovary is required for ovulation and folliculogenesis.”
- Line 459: Missing reference for the statement: “Although Fosl2 deletion in mice is lethal, impacting the development of the heart, bone...”
- "Lines 653 lacks a reference: 'package RSEM...'"

Reviewer #2

(Remarks to the Author)

In this article, the authors systematically characterized the reorganization of chromatin accessibility that underpins extensive transcriptomic alterations in granulosa cells (GCs) across various stages of follicular maturation in murine and porcine models, and identified the transcription factor Fosl2 as a key regulator, orchestrating the dynamics of chromatin accessibility to regulate GCs function during follicular maturation. Moreover, I appreciate the authors for their thorough efforts in elucidating the role of the transcription factor Fosl2 in granulosa cells (GCs) during follicular maturation, particularly through the use of highly reliable ‘wet lab’ data. Especially, their work is valuable in the context of the mouse models with GC-specific Fosl2 deletion, which adds significant depth to our understanding of the molecular mechanisms involved. The quality of the presentation is excellent, and the structure of the presentation is clear. Overall, the conclusions appear sound and reliable. However, there are several issues that should be addressed before publication in Nature Communications. Especially, the detailed description of the quality of relevant omics data is lacking in current manuscript, which largely reduce the confidence in the related conclusions.

Specific comments:

1. For Result-1 and Fig. 1. The authors showed GCs undergo globally distinct chromatin landscape remodeling during follicular maturation using porcine model. But there was no the description of the quality of relevant omics data, which can increase the credibility of the results, such as mapping ratio, non-redundant read fraction (NRF), fraction of reads in peaks (FRiP) and others. Given that, the authors should check the whole manuscript and revised it.
2. For Lines 170-171. The authors described “Conversely, the pathways enriched in genes adjacent to GIAs were significantly associated with dysregulated development and exhausted growth capacity (Fig. 2d)”. But Fig. 2d exhibited some pathways (e.g., ‘Signal transduction’, ‘Intracellular signal transduction’, ‘Regulation of neuron migration’, ‘Regulation of postsynaptic density assembly’, and others). How to understand their roles in dysregulated development and exhausted growth capacity?
3. For Lines 222-224. The authors described “Distal Fosl2 binding peaks were also predominantly associated with upregulated genes, revealing a principal role for Fosl2 as a transcriptional activator modulating GAA remodeling (Fig. 3i)”. As we know, the three-dimensional genome architecture (i.e., topologically associating domains [TADs]) can promote/inhibit interaction between promoters and distal enhancers. Given that, I recommended the authors can investigate the relationship among TADs, Distal Fosl2 binding peaks and the promoters of its nearest genes, based on potential public Hi-C data.
4. For Lines 366-368. The authors described “Although the total captured cell numbers were consistent at the beginning, GC subclusters significantly decreased after Fosl2 loss (Extended Data Fig. 9c)”. But Extended Data Fig. 9c just exhibited the number of GCs. So, the authors should add the comparative analysis of the cell proportions of major cell types between Fosl2F/F and Fosl2cKo.
5. For Bulk RNA-seq data processing in method section. The authors performed R package DEseq2 to obtain DEGs with $\log_2(\text{fold-change}) \geq 1$ and $p \text{ value} < 0.05$. However, the reported studies generally used adjusted p value as cutoff to minimize false positive results. So, the authors should clarify why using p value, and not adjusted p value.

Version 1:

Reviewer comments:

Reviewer #1

(Remarks to the Author)

The revised manuscript addresses several of the technical concerns raised in my initial review, and I would like to acknowledge the authors' thoughtful responses and the additional data provided to improve the clarity and transparency of the genomic analyses. These efforts are appreciated. That said, several key issues related to the robustness and reproducibility of the epigenomic analyses remain insufficiently addressed, and I believe they still warrant further consideration before the conclusions can be fully supported.

In response to Comment 1.10 and Figure 4, the revised manuscript presents heatmaps and profile plots suggesting a role for Fosl2 in chromatin accessibility. While these visualizations are informative, the central conclusion would benefit from additional quantitative analyses supported by appropriate statistical tests. This would help reinforce the claim and provide a stronger evidentiary basis for Fosl2's proposed function. The inclusion of summary statistics (such as average signal intensities with boxplots) would enhance interpretability and address potential concerns regarding signal variability.

The Spearman's correlation analyses presented offer some insight into global signal trends, but the strength of the correlation still appears modest. As such, the conclusion that Fosl2 is a key determinant of open chromatin might currently appear somewhat overstated. Presenting this as a trend, rather than a definitive conclusion, or complementing the analysis with additional quantitative comparisons, would improve the balance between interpretation and evidence.

The authors have addressed the lack of a standardized pig genome blacklist by applying various filtering strategies, including the use of mm10 blacklist homologs. While this effort is appreciated, I would respectfully suggest that a direct comparison with the most recent *Sus scrofa* blacklist could further clarify the extent to which artifact-prone regions were addressed. Even a partial or preliminary overlap analysis would enhance transparency and help contextualize the limitations and strengths of the dataset.

The methodology surrounding the IDR analysis of GAAs and GIAs remains unclear. By standard definition, IDR must be performed on raw peak calls from biological replicates, following proper filtering and quality control. It is not clear from the authors' response whether these steps were followed consistently.

Moreover, while the authors compare IDR-derived peak sets to BEDTools-merged consensus peaks, they do not provide key IDR metrics such as reproducibility scores or thresholds. Without these, it is difficult to assess the reliability of the peak reproducibility claims. The argument that BEDTools merging captures rare but meaningful signals may be valid; however, the lack of overlap with IDR-defined peaks raises legitimate concerns about biological reproducibility. Given that the GAAs and GIAs are central to the study's conclusions, the lack of rigorous reproducibility assessment limits confidence in their functional relevance.

Finally, the use of PCA as a justification for reproducibility of peak-level data is methodologically inappropriate. PCA can demonstrate sample-level variability or clustering but is not a valid proxy for evaluating the reproducibility of specific genomic regions like peaks or accessible chromatin sites.

To illustrate some of the concerns above, I have included IGV snapshots of several genomic regions featured in the manuscript. These include ATAC-seq and CUT&Run signal tracks, the *Sus scrofa* blacklist, the ATAC-seq nc, and the CUT&Run IgG control. Regardless of overlap with the blacklist, the enrichment observed at several of these loci appears inconsistent or potentially confounded by background signal, and may benefit from additional validation or clarification.

Reviewer #2

(Remarks to the Author)

I would like to thank the authors for providing an extensive point-by-point response to my concerns and for their efforts to improve their manuscript. And the authors have addressed the concerns I have raised. So, I suggest to accept it.

Version 2:

Reviewer comments:

Reviewer #1

(Remarks to the Author)

I would like to thank the authors for addressing my concerns and for their efforts to improve the manuscript. Therefore, I suggest accepting the manuscript in its current form.

Point-by-point answers

Reviewers' comments:

Reviewer #1 (Remarks to the Author):

Summary:

Zhang *et al.* investigated the role of Fosl2 in porcine and mouse models, with a primary focus on the antral and ovulatory phases. Their study generated valuable transcriptomic and epigenetic data to assess the involvement of Fosl2. They identified accessible regions activated in granulosa cells, linking these regions to genes associated with granulosa cell development and the regulation of Fosl2. While the results appear promising, the study has significant flaws concerning the models used for the "ovulation phase" and lacks transparency in providing adequate supplementary information and standardized analytical approaches for both transcriptomic and epigenetic data. The following concerns require clarification from the authors.

Thank you so much for taking the time to carefully review our MS. The constructive comments are very helpful in improving the quality of our work. Your comments are shown in bold font below, and specific concerns have been numbered. Our point-by-point answers are provided in standard blue font, with all modifications and additions to the MS highlighted in red text. Once again, we sincerely appreciate your efforts to provide such valuable suggestions!

Q1.1: Maturation of porcine granulosa cells and oocytes to recapitulate the ovulatory phase. In lines 92-95, the authors mention: 'We derived porcine cumulus-oocyte complexes (pCOCs) from porcine ovaries and matured them *in vitro* to generate porcine granulosa cells (pGCs) and oocytes at different phases

(Fig. 1a).' One of the questions that needs to be addressed is how the authors optimized the conditions to achieve the highest similarity with the follicle maturation process in porcine ovaries (i.e., achieving optimal conditions for IVM). Even considering that they have applied the best possible conditions, there is always the challenge of achieving a transcriptome and epigenetic regulation that closely resembles natural conditions in various biological contexts. Therefore, it would be beneficial for the authors to demonstrate the extent to which variability in IVM impacts the transcriptomic and epigenetic methods used in their study.

R1.1: We sincerely appreciate your constructive comments on this important issue. As noted, the *in vitro* maturation (IVM) rate in porcine models is indeed not 100%, which aligns with the widely accepted consensus in the field^{1, 2, 3}. The current international average for porcine IVM efficiency ranges between 60-80%, influenced by factors such as species-specific traits, rearing conditions, and culture systems. In order to reduce the variability caused by IVM in subsequent analyses, we strictly standardized the collection protocol for porcine cumulus-oocyte complexes (pCOCs): specifically, we aspirated COCs from medium-sized follicles (3-6 mm) by using an 18-gauge needle connected to a 10-mL syringe. To further optimize the selection process and enhance IVM efficiency, oocytes enclosed by multiple layers of tightly packed granulosa cells (GCs) and exhibiting homogeneously dark cytoplasm were selected for maturation. By strictly adhering to internationally validated protocols, we successfully achieved a stable IVM efficiency of approximately 65%. By utilizing this platform, we have successfully generated transgenic and chimeric piglets and published robust findings, further underscoring the reliability of the IVM protocol^{4, 5}.

In regard to transcriptional and epigenomic analyses, we acknowledge that incomplete IVM introduces immature antral-phase pGCs (AP-pGCs) into ovulatory phase populations (OP-pGCs); however, these AP-pGCs inherently serve as the experimental control group in our design, and by comparing differentially expressed genes (DEGs) and differentially accessible regions (DARs) between OP-pGCs (non-100% purity) and AP-pGCs, we effectively exclude genes with minimal differential signals due to heterogeneity of OP-pGCs, thereby enhancing the significance of identified markers. Alternatively, the increase in the baseline further enhances the existence of DEGs or DARs. This approach has been further validated in murine models, where the rate of maturation in murine COCs (mCOCs) was 100% by superovulation.

Despite this purity, murine DAR analyses still highlight *Fosl2* as a top-ranked mediator, consistent with findings in porcine models (Revised Fig. 6). This cross-species consistency confirms that incomplete IVM efficiency in pigs may not compromise the fundamental pattern during follicular maturation. As the reviewer pointed out, we have recognized the necessity for methodological clarity, and in response, we have provided additional details of IVM protocols in the Methods section as follows: “ *To generate pGCs and oocytes at the ovulatory phase, the pCOCs were washed three times in the in vitro maturation (IVM) medium (Gibco, 3110-035) supplemented with 0.1% polyvinylalcohol (Sigma, P8136), 0.026 M sodium bicarbonate (Sigma, S5761), 3.05 mM D-glucose (Sigma, G7021), 0.91 mM sodium pyruvate (Sigma, P4562), 1 mg/mL gentamicin (Sigma, G1264), 10% porcine follicular fluid, 0.57 mM cysteine (Sigma, C7352), 0.5 mg/mL luteinizing hormone (Sigma, L5269), 0.5 mg/mL follicle-stimulating hormone (Sigma, F2293), and 10 ng/mL epidermal growth factor (Sigma, S4127). Approximately 70-80 pCOCs were then placed into individual wells (NEST, 702001), each well containing 500 μ L of IVM medium overlaid with 400 μ L of mineral oil. The plates were cultured for 42-44h at 39°C and 5% CO₂ to facilitate maturation. Ovulatory-phase pGCs were removed from the pCOCs by a brief 5-min exposure to 0.1% hyaluronidase solution. The resulting MII oocytes, characterized by the presence of an extruded first polar body, were individually inspected under stereoscopic microscopy (Nikon, Japan) for further use.*” (Revised MS, Lines 531-545). We sincerely hope these explanations can address your concerns.

Q1.2: Mouse superovulation model used in Figure 5. The authors used mature, cycling mice (8 weeks old) for superovulation and ovulatory phase studies, but this model presents several challenges compared to prepubertal models. One of the primary issues is follicular asynchrony, as sexually mature mice naturally undergo estrous cycles, resulting in a heterogeneous population of follicles at various developmental stages. This variability impacts the response to PMSG and hCG stimulation, as some follicles may already be at the antral stage while others remain preantral, leading to less uniform follicular growth compared to prepubertal mice, where follicles are more synchronized. Consequently, the efficiency of superovulation in mature mice tends to be lower, as studies have reported a reduced number of ovulated oocytes due to follicular heterogeneity.

R1.2: Thank you for your valuable comments on the selection of 8-week-old sexually mature female mice as *in vivo* research models (Revised Extended Data Fig. 6a). We fully acknowledge your observation that 8-week-old mature mice can exhibit higher follicular heterogeneity compared to 4-week-old prepubertal mice, which lead to differences in the numbers of mCOCs (Response Fig. 1.1a). Following your suggestion, we then compared the superovulated oocyte number between *Fosl2*-deleted mice and controls in 4-week-old prepubertal mice, and found a substantial reduction of mCOCs in *Fosl2*-deleted mice, which indicated the poor sensitivity to gonadotropins (Response Fig. 1.1b). This reduction was consistent with the conclusions in 8-week-old mature mice. It seems that the absence of *Fosl2* had a profound effect on the reproductive capabilities of female mice, regardless of whether they were 4-week-old or 8-week-old.

Response Fig. 1.1 | The quantification of ovulated oocytes by superovulation. **a**, Quantification of oocyte number per female between 4-week-old and 8-week-old mice. **b**, Quantification of oocyte number per female between 4-week-old *Fosl2*^{F/F} and *Fosl2*^{cKO} mice. Error bars indicate the mean \pm S.E.M. of three independently performed experiments. * $p < 0.05$, *** $p < 0.001$, as determined using Student's *t*-test.

Notably, variations in superovulation efficiency resulted in quantitative differences in mCOCs; however, these differences did not affect the synchronized developmental stages observed after PMSG/hCG stimulation. Nevertheless, our study was not intended to compare the absolute numbers of follicular populations at the beginning, as the number of mGCs surrounding the ovulatory-phase follicle was sufficient to meet our experimental requirements in both 4-week-old and 8-week-old mice for downstream analyses, such as RNA-seq, ATAC-seq. This ensures the acquisition of synchronous ovulatory-phase mGCs under PMSG/hCG stimulation and provides adequate GC samples for downstream studies, regardless of superovulation efficiency. Moreover, the selection of sexually mature mice was primarily driven by their capacity to closely mimic adult physiological conditions, as their ovarian

characteristics more closely approximate those of reproductive-age human females. Although prepubertal mice offer enhanced follicular synchronization, they exhibit substantial differences from sexually mature mice in the mechanisms of hormonal responsiveness and oocyte maturation^{6,7}. Importantly, previous studies have demonstrated that 8-week-old mice have just reached sexual maturity and are commonly used for fertility testing for their fully developed reproductive systems and stable hormone levels^{8,9,10}. Given our research goals of assessing reproductive capacity and ovarian aging, the mature mouse model was more suitable despite the challenge of follicular asynchrony (Revised Fig. 7b-e, j and k). The details for the choice of 8-week-old sexually mature female mice have also been added in the Methods section (Revised MS, Lines 547-548). Thank you again for your insightful comments.

Q1.3: scRNA-seq Data from *Fosl2^{F/F}* and *Fosl2^{CKO}*. For this experiment, the authors used two 8-week-old female mice for each genotype (*Fosl2^{F/F}* and *Fosl2^{CKO}*) from the same litter. Were these mice monitored to confirm that they were at the same stage of the estrous cycle? When mature cycling mice (8 weeks old) are used as a model for transcriptomic and epigenetic studies aimed at investigating changes during the ovulatory phase, several critical factors must be considered to ensure data reliability and biological relevance. One significant challenge is the variability in baseline gene expression, as follicular development is not synchronized. This lack of synchronization leads to transcriptomic profiles that reflect a mixture of follicular stages rather than a distinct ovulatory phase signature. Such high biological variability complicates the isolation of ovulation-specific transcriptional changes. Additionally, heterogeneous chromatin states across follicles at different developmental stages may confound epigenetic analyses, as changes in chromatin accessibility will not be uniform, potentially obscuring ovulation-driven chromatin remodeling events. Furthermore, batch effects and sample normalization are crucial considerations, as the high variability among animals necessitates larger sample sizes and stringent normalization strategies. These factors suggest that the immature mouse model may provide a more controlled and predictable system for studying the ovulatory phase. The absence of this information, combined with the poor distinction of cell population marker expression in Figure S9, raises several questions regarding the comparative analyses used to demonstrate the effects of *Fosl2* knockout (KO).

R1.3: Thank you for your careful reading regarding our experimental design. We fully agree with your valuable insights into the synchronization of the estrous cycle and its potential impact on data interpretation. We sincerely apologize for omitting details regarding tissue collection for scRNA-seq analysis in the original MS. Indeed, all mice (*Fosl2^{F/F}* controls and *Fosl2^{CKO}* mutants, n = 2 per group) underwent estrous cycle synchronization before sequencing. Vaginal cytology smear analyses revealed predominantly keratinized, anucleated epithelial cells in both groups, which exhibit characteristic estrus-phase morphological features: flattened polygonal structure, irregular borders, and complete nuclear dissolution (Revised Extended Data Fig. 7c). The cytological results confirm that all sampled mice were in the same estrus phase during ovary collection, thereby minimizing follicular developmental heterogeneity. Moreover, synchronizing mice to the estrus phase enabled us to specifically assess the effects of *Fosl2* knockout on the “pre-ovulatory ovarian microenvironment”, avoiding confounding signals from mixed cycle stages. We have added the estrous cycle monitoring result in the revised MS, and the related protocol for vaginal cytology smears has also been added in the Methods section (Revised MS, Lines 601-609).

Revised Extended Data Fig. 7c, Vaginal cytology smear analyses showing the characteristic estrus-phase morphological features in *Fosl2^{F/F}* and *Fosl2^{CKO}* mice that were subjected to scRNA-seq. Scale bar, 50 μ m.

With respect to the concern regarding the use of immature models, we have detailed the advantages of employing a mature mouse model, as mentioned in R1.2, which focuses on assessing reproductive capacity and ovarian aging. Additionally, *Fosl2* expression significantly increases during the developmental transition from antral follicles to ovulatory follicles, suggesting that the function of *Fosl2* is likely to be exerted after sexual maturity. However, the ovaries of immature mice have not yet undergone a complete estrous cycle due to the immaturity of the hypothalamic-

pituitary-ovarian (HPO) axis¹¹; hence, they may not accurately reflect the true function of *Fosl2* in the pre-ovulatory microenvironment. Through the aforementioned estrous cycle synchronization design, we also effectively mitigated the follicular heterogeneity in the mature mouse model.

Moreover, we fully understand your concerns about batch effects and sample normalization in our scRNA-seq data which we have been elaborated in the R1.14. We also appreciate your concern regarding the initial marker gene heatmap in Extended Data Fig. 9a, which has now been moved to Extended Data Fig. 8a. In the original analysis, marker genes were identified using thresholds of $\log_2FC > 0.25$, $\text{min.pct} = 0.1$, and p value < 0.01 . While this approach maximized sensitivity, it risked retaining genes with subtle or sporadic expression differences. Therefore, we increased min.pct from 0.1 to 0.5, and this adjustment prefers markers with stronger fold-changes and consistent cell-type specificity, effectively filtering out low-prevalence signals that may contribute to ambiguous patterns. As a result, the updated heatmap demonstrated distinct expression patterns for key lineage markers, such as *Inhba* in mural GCs and *Ghr* in atretic GCs, which was highly consistent with the previous scRNA-seq data in ovaries¹². We sincerely value your suggestions, which has strengthened the transparency of our study.

Revised Extended Data Fig. 8a, Heatmap of the top five markers of each GC subcluster by fold change.

Other Comments:

Q1.4: In Figure 1c, the authors present a figure comparing gene expression across different replicates and samples based on FPKM (Fragments Per Kilobase of transcript per Million mapped reads). It is recommended, however, to use Transcript Per Million (TPM) for comparing gene expression levels. While the overall expression pattern may not change, the expression values for each gene would differ and become more directly comparable.

R1.4: Thank you for your insightful guidance. Following your suggestion, we have revised our analytical approach and visualization methodology regarding the heatmap you pointed out. The revised Fig. 1c has been regenerated using TPM values derived through conversion from FPKM measurements, as detailed below. We have also updated the Methods section with detailed descriptions (Revised MS, Lines 683-685). Thank you again for your constructive suggestion!

Revised Fig. 1c, Heatmaps showing the expression (TPM) of RNA-seq in pGCs and oocytes across different maturation phases (left), with example genes listed (right).

Q1.5: In lines 102–106 (“In contrast...”), the authors state that the expression of *Lhx8*, *Sycp3*, and *Sox30*, identified as oocyte-specific markers, is not detectable in porcine granulosa cells (pGCs). This raises two questions: First, why is it necessary to demonstrate that oocyte markers are not expressed in granulosa cells? Secondly, “detectable” is not an appropriate term in this context, as the heat map appears to be based on the z-scores of FPKM values. It is expected that a lack of visible color in the heat map does not necessarily indicate the absence of expression. To support their claim, it is recommended to include a separate graph (e.g., a bar plot) showing the actual FPKM values for these genes.

R1.5: Thank you for your valuable suggestions. We fully agree with your comment and apologize for any confusion caused by the inappropriate description in the original MS. Indeed, to ensure the reliability of our oocyte-GCs separation procedures, we deliberately highlighted several representative oocyte-specific marker genes within the pGCs transcriptomes. This analysis further validates the robustness of our pGCs RNA-seq data, which has also been employed in other studies (see Figure 1e in Wu *et al.*'s research)¹³; however, as you suggested, additional studies are warranted. To enhance clarity within space limitations, we have now presented the TPM values of selected marker genes in the Results section (Revised Extended Data Fig. 1c). Corresponding statement has also been rephased in the revised MS (Revised MS, Lines 104-107). Thank you once again for your careful attention to this detail!

Revised Extended Data Fig. 1c, Bar plots showing the representative gene expression (TPM) of RNA-seq under different maturation phases. Two biological replicates are independently performed for each sequencing. One-way ANOVA followed by Dunnett's multiple comparisons test is used for statistical analysis. ** $p < 0.01$, *** $p < 0.001$.

Q1.6: Extended Data Fig. 3: It is uncommon to use IGV snapshots to demonstrate universal overlap. Moreover, the conclusion that the accessible regions are generally shared between ATAC-seq and CUT&Tag data cannot be made solely

based on such snapshots. Both ATAC-seq and CUT&Tag data are typically processed to identify "peaks," which represent regions of enrichment. To compare the peak lists from these methods and determine the degree of overlap, tools like Intervene (<https://bmcbioinformatics.biomedcentral.com/articles/10.1186/s12859-017-1708-7>), specifically the "intersect" feature, could be utilized. This approach would provide a more accurate assessment of how many regions are identified by both techniques. Additionally, using tools like GREAT to annotate open chromatin regions from the CUT&Tag and ATAC-seq data and overlapping their associated genes would be more meaningful than relying on an IGV snapshot.

R1.6: Thank you for pointing out the limitations of relying solely on IGV snapshots for assessing chromatin accessibility overlap. We fully agree that quantitative peak comparison is essential for robust conclusions. According to your suggestion, we employed Intervene-based quantification to assess the peak overlap between our ATAC-seq data and H3K4me3, H3K27ac-driven CUT&Tag data. As shown in the revised Extended Data Fig. 2g and h, we discovered that a significant proportion of the identified chromatin accessibility sites overlapped with promoter and enhancer signals. Furthermore, when comparing peak overlap during both the antral and ovulatory phases, a significantly higher percentage of our ATAC-seq peaks overlapped with promoter signals compared to enhancer signals, which was consistent with our observations in the IGV snapshots (Revised Extended Data Fig. 2g, h).

Revised Extended Data Fig. 2g and h, Bar plots displaying the distribution of ATAC-seq signals at the antral and ovulatory phases in pGCs with H3K4me3 and H3K27ac CUT&Tag signals using the Intervene tool. Intersection numbers are labeled.

To assess the biological relevance of overlapping and non-overlapping regions, we examined functional annotation using GREAT, as you suggested. Our analysis

revealed that ATAC-seq and H3K4me3 gene annotations exhibit greater similarity at both the antral and ovulatory phases (Response Fig. 1.2a). Furthermore, genes adjacent to H3K4me3 regions show significantly higher overlap with those near ATAC-seq regions compared to H3K27ac-associated genes (Response Fig. 1.2b). These findings are also consistent with our IGV snapshot observations. To enhance clarity within space limitations, we have replaced the original IGV snapshots with Intervene diagrams in the revised MS (Revised Extended Data Fig. 2g, h). Thank you again for your valuable suggestions!

Response Fig. 1.2 | Gene annotation in ATAC-seq and CUT&Tag data across the antral and ovulatory phases. a, Bar plots displaying the distribution of ATAC-seq, H3K4me3 and H3K27ac CUT&Tag signals as analyzed by GREAT. **b,** Venn diagram showing the overlap between ATAC-proximal genes and genes proximal to H3K4me3 and H3K27ac, respectively.

Q1.7: Lines 164-167: The authors mention 'In agreement with GO...', but this claim is not well documented. It is necessary to provide the GDGs list and include a few insertion tracks of well-recognized GC biomarkers to support the claim. Additionally, in Extended Figure 4d, it is unclear what data were used to show the expression or accessibility of the GDGs list.

R1.7: Thank you for your insightful guidance. Our GO enrichment analysis of genes neighboring GAAs revealed associations with multiple developmental regulation pathways, prompting us to investigate their potential involvement in follicular maturation. Notably, 37.5% (1,021/2,722) of GAA-proximal genes overlap with the GDG list, including well-recognized GC biomarkers highlighted in Revised Extended

Data Fig. 3d. To further visualize this overlap, we selected representative overlapping genes from these two gene sets and used their FPKM values to generate Extended Data Fig. 3e. Meanwhile, GDGs were rigorously defined based on our transcriptomic analysis of GCs during follicular maturation, with the full list provided in Supplementary Table 3 (Revised MS, Lines 697-703). The detailed definition of GDG and corresponding control groups has also been addressed in the subsequent R1.12. For transparency and reproducibility, the locations and descriptions of GAAs are also included in the Supplementary Information (Supplementary Table 5).

Revised Extended Data Fig. 3d, Venn diagram showing the overlap between GAA-proximal genes and identified GDGs list.

Q1.8: Extended Data Figure 5b and its relation to Fos11 expression: First, in Figure 5b, it is not clear which type of expression value (e.g., RPKM, TPM, etc.) was used to show the change in the expression of the top genes (Atf3, Fos12, Fos11, Junb, Jun, Fos). Secondly, the Fos11 expression in the heatmap of Figure 5b, from the antral to ovulatory phase, demonstrates an increased expression level. However, in Figure 5d, the relative mRNA expression of Fos11 is decreased during the same phase transition as shown in Figure 5b. Does the author have any explanation for this discrepancy?

R1.8: We sincerely appreciate your attention to this detail. As the reviewer pointed out, our multi-platform analysis reveals seemingly contradictory regulation patterns: while RNA-seq data using normalized RPKM values showed a marginal Fos11 mRNA increase ($\log_2 \text{FC} = 0.2$, $p = 0.08$, $p_{adj} = 0.21$) in Revised Extended Data Fig. 4b that did not reach statistical significance, qPCR quantification paradoxically indicated decreased expression during the antral to ovulatory transition. To resolve this discrepancy, we conducted immunoblotting analysis, which demonstrated progressive accumulation of Fos11 protein during follicular maturation (Response Fig. 1.3). However, this mRNA-protein discordance aligns with established biological principles, where gene expression regulation operates at multiple levels including mRNA stability, translational efficiency, and protein degradation¹⁴.

Moreover, the compensatory potential of Fos11 may explain this apparent discrepancy, particularly given our observation that Fos12 deficiency in GCs fails to induce severe infertility. This suggests that the AP-1 transcription factor family may compensate through functional redundancy within Fos12. Nevertheless, the critical role of Fos12 in follicular development should not be overlooked, as its unique expression kinetics, chromatin occupancy patterns, and distinct reproductive phenotypes are all well-illustrated in our study. We have explained the potential functions of Fos11 and other members of the AP-1 family in the Discussion section as follows: “Furthermore, the potential compensatory roles of *c-Fos* family members that share structural homology with Fos12 (e.g., Fos11) during follicular maturation merit consideration. For example, the absence of severe fertility defects in GC-specific Fos12 knockout mice may reflect functional redundancy through dosage compensation by Fos11, given their conserved DNA-binding domains. Nevertheless, deletion of Fos12 in mice results in lethality and affects the development of the heart, bones, and intestines, highlighting its critical and non-redundant regulatory role in maintaining overall embryonic and postnatal health¹⁵. This suggests that while compensatory mechanisms may mitigate acute fertility loss, Fos12 exerts unique temporal or context-dependent control over pathways critical for ovarian homeostasis, underscoring its indispensable pathophysiological relevance. From another perspective, these TFs participate in chromatin remodeling, suggesting the potential functional interplay and combinatorial effects across different developmental contexts, aligning with the enrichment of AP-1 TFs at GAAs in our study (Fig. 3a)^{16, 17, 18, 19}.” (Revised MS, Lines 464-477). Thank you for your attention to this important issue, as it provides valuable insights for future research on the roles of other AP-1 family transcription factors during folliculogenesis.

Response Fig. 1.3, Immunoblotting analysis (left) and quantification (right) of Fos11 expression in pGCs in different maturation phases. Error bars indicate the mean \pm S.E.M. of three independently performed experiments. ** $p < 0.01$, as determined using Student's *t*-test.

Q1.9: Extended Figure 6c: This figure is introduced as “GO enrichments of the 1137 regions that close after Fosl2 knockdown using GREAT analysis. Bar length represents the enriched p-value for biological processes.” However, it appears that the figure is the result of motif enrichment analysis, not GO enrichment. Additionally, the meaning of the colors on the plot is not explained.

R1.9: Thank you for the careful examination of the MS, and we regret the mistake in the original description of Extended Data Fig. 6c, which has now been moved to Fig. 4b. Indeed, the bar length represented the motif enrichment of the regions that close after Fosl2 knockdown using GREAT analysis, rather than GO enrichment scores as previously misstated. Meanwhile, the highly ranked motifs for bZIP factors were highlighted in red. To address this issue, we have corrected the mistake in revised Figure legends and conducted a comprehensive review of all figure legends to prevent similar errors. Thank you again for your valuable comments!

Q1.10: Line 231-236, Figure 4c: the author shows Spearman’s correlation of differential accessibility peaks between follicular maturation phases and accessibility regions that changed after Fosl2 knockdown, with $r=-0.202$. The correlation coefficient is relatively close to zero, suggesting a weak association between the two variables. The weak correlation might not be biologically meaningful. The same weak correlation has been shown in Figure 3e ($r = 0.406$), which both suggests that Fosl2 knockdown has a limited impact on differential accessibility peaks; other factors besides Fosl2 knockdown might influence differential accessibility peaks during follicular maturation.

R1.10: We sincerely thank you for raising this important issue regarding the interpretation of correlation coefficients. For the analysis in the original Fig. 4c, which has now been moved to Extended Data Fig. 5c, we acknowledge that the inverse correlation ($r = -0.202$) suggests that Fosl2 knockdown alone may not fully account for chromatin accessibility changes during follicular maturation. This observation aligns with your insight that other regulatory factors (e.g., Fosl1, as explained in R1.8) might compensate for or cooperate with Fosl2 during follicular maturation. Nevertheless, the directional consistency of this negative correlation supports the non-redundant role of Fosl2 in specific chromatin remodeling events to a certain degree. To prevent potential

overinterpretation, we have moved this analysis to the Extended Data section with a modified statement in revised MS (Revised MS, Lines 239-242). Regarding Fig. 3e, while the positive correlation ($r = 0.463$) between Fosl2-bound regions and maturation-associated accessibility changes is indeed moderate, we emphasize that such effect sizes are well-documented in studies of transcription factor-mediated chromatin dynamics. For example, similar moderate correlation coefficients have been reported in previous research, such as $r = 0.4297$ in Dinh *et al.*, $r = 0.322$ in Ling *et al.*, and $r^2 = 0.18$ in Chen *et al.*^{20, 21, 22}. Importantly, this correlation remains statistically significant ($p < 0.001$), reinforcing its biological relevance. As the reviewer pointed out, we have revised the main MS to more explicitly state that Fosl2 functions as part of a broader regulatory network rather than a sole driver of chromatin remodeling in the Discussion section (Revised MS, Lines 464-477).

Q1.11: Lines 247–250: The sentence starting with “To further investigate...” lacks clarity, and the explanation of the data and techniques used is not well articulated. It appears that the clustering of Fosl2 categories is based on CUT&Tag for Fosl2, which was then mapped to the ATAC-seq data of pGCs in both normal and KD models. If this assumption is correct, it remains unclear and undocumented why the authors stated that “only the ovulatory-specific peaks underwent extensive decreases in accessibility upon Fosl2 silencing (Fig. 4e).” There seems to be a considerable difference in AP-specific peaks compared to Fosl2-KD1 and Fosl2-KD2. Additionally, the figures on the right side of Fig. 4e are not explained, either in the text or in the figure legends, making it difficult to understand their relevance.

R1.11: Thank you for your careful reading and valuable comments! We fully agree with this comment and apologize for any confusion caused by the inappropriate description in the original MS. To investigate whether Fosl2 participated in regulating chromatin accessibility, we first classified the Fosl2 peaks using Fosl2 CUT&Tag data and assigned these clusters to our ATAC-seq data during follicular maturation and following Fosl2 knockdown, respectively (Revised Fig. 4e, left panel). Notably, Fosl2 peaks that increased at the ovulatory stage (marked as ovulatory-specific, or OP-specific) correlated with regions of enhanced chromatin accessibility from the antral to ovulatory phase (Revised Fig. 4e, middle panel). However, these peaks exhibited a marked reduction following Fosl2 knockdown, in contrast to the unchanged

accessibility observed in antral-specific peaks (Revised Fig. 4e, right panel). The results indicated that ovulatory-specific peaks, not antral-specific peaks, which were classified as GAAs, were dynamically regulated by *Fosl2* expression. To better visualize this difference, we have also included a graph showing the enrichment of normalized ATAC-seq peaks after *Fosl2* knockdown. As the reviewer pointed out, we realized that the conclusion was confusing; therefore, we have rephrased the sentence to make the MS more readable: *“To further investigate whether Fosl2 participated in regulating chromatin accessibility at putative promoter-anchored GAA regions, the aforementioned differential Fosl2 occupancy peaks identified with Fosl2 CUT&Tag were assigned to accessibility profiles during follicular maturation and following Fosl2 knockdown. Of note, the strength of ovulatory- and antral-specific peaks in Fosl2 occupancy consistently agreed with chromatin accessibility dynamics in the process of follicular maturation; yet, only the ovulatory-specific peaks exhibited extensive decreases in accessibility upon Fosl2 silencing, contrasting with the stable accessibility of antral-specific peaks (Fig. 4e).”* (Revised MS, Lines 249-257). In addition, the relevant figure legend has also been rephrased for greater clarity (Revised MS, Lines 1085-1089). Thank you again for your careful reading!

Q1.12: The idea of focusing on GDGs (GC-involved development genes) related to GAAs is intriguing. However, certain aspects of the GDGs lack transparency. The article does not provide a clear definition of GDGs. Which groups of genes and genomic regions are classified as GDGs? Were these genes selected from a specific database, or are they derived from analyses conducted by the authors themselves?

R1.12: We thank you for highlighting this important issue. The definition of GDG and its joint analysis were inspired by Wang *et al.*'s research on their self-defined ZGA transcriptome dataset, and the GDGs were systematically identified through transcriptomic profiling of GCs during follicular maturation²³. Specifically, GDGs were defined as those not expressed or lowly expressed at the antral phase, but become upregulated (at least 2-fold upregulation) at the ovulatory phase, ultimately yielding 2,068 qualifying genes. For comparative analysis, an equivalent set of control genes was randomly selected from protein-coding genes. The complete GDG list and detailed methodology are available in the Methods section and Supplementary Table 3 (Revised MS, Lines 697-703). We sincerely appreciate your attention to this detail!

Q1.13: Considering the second section of this part of the results, the title is misleading considering the results provided. Only the first part briefly covers what the authors concluded from comparing pGCs to mice. In the second section, the authors shift back to discussing pGCs without effectively connecting their results to the findings in mice. Thirdly, the authors attempt to demonstrate that Fosl2 replacement from the oocyte to pGCs occurs through gap junctions. However, based on the experiments and explanations provided, such a conclusion is not substantiated. If this replacement indeed occurs through gap junctions, the authors should have shown that disrupting or destroying the gap junctions would interrupt Fosl2 replacement. Instead, in their pGC co-culture strategies, they show elevated Fosl2 expression in both antral and ovulatory follicles, which does not conclusively support their claim.

R1.13: We sincerely appreciate your valuable comments on this important issue! We fully agree with this comment and regret that these unclear descriptions caused misunderstandings. What we really want to convey in the original Fig. 6 is that the mechanisms of Fosl2 in orchestrating follicular maturation are conserved across mammalian species, both *in vitro* and *in vivo*.

In the initial part of the data presented in Fig. 6, we confirmed that the role of Fosl2 in the porcine model (*in vitro*) was conserved in the murine *in vivo* model (Revised Fig. 6a-d). Interestingly, we subsequently observed opposing expression patterns of Fosl2 in GCs and oocytes in mice (Revised Fig. 6e,f). Since previous studies have shown that oocytes-GCs interactions play a critical role in follicle development by exchanging some materials between oocytes and GCs, such as cAMP, which control the meiotic maturation, we wondered if Fosl2 shuttles from oocytes to GCs during the transition from the antral to ovulatory phase^{24, 25}. To test this hypothesis, we needed to isolate GCs from COCs at the antral stage to disrupt physical communication between oocytes and GCs, following the previously reported system^{2, 25}. Considering that this experiment needs to be performed *in vitro* and the relative ease of obtaining COCs in pigs, we then employed a porcine model to validate this assumption. Our results indeed showed that Fosl2 functions independently of direct intercellular connections with oocytes (Revised Fig. 6h).

However, as the reviewer pointed out, we realized that the term “gap-junction” was inappropriately used in our study, since the experimental design did not account

for the microvilli function that may exist between oocytes and GCs²⁵. Even so, our *in vitro* porcine model directly demonstrated that Fosl2 does not function in follicle maturation by shuttling in channel between oocytes and GCs. To avoid misunderstandings, we have rephrased the MS regarding the description related to gap junctions as follows: “*Throughout most of the course of folliculogenesis, oocyte-GC physical interactions such as gap junctions and microvilli facilitate transfer of molecules between the oocyte and GCs for signaling communication*^{24, 25}.” (Revised MS, Lines 322-324). Our findings highlight the potential for alternative modes of oocyte-GC communication, suggesting a promising direction for future mechanistic studies.

Q1.14: (scRNA-seq) In the final section of the results, the authors focus primarily on the single-cell RNA-seq data of *Fosl2^{F/F}* and *Fosl2^{cKO}*. However, the analysis lacks transparency and documentation, and the analysis pipeline is not well standardized. The authors have not provided any gene signatures for each cluster. Regarding data processing, they mentioned filtering cells based on two criteria (mitochondrial content higher than 30% and fewer than 500 genes per cell). These criteria are not commonly used in single-cell RNA-seq analyses, and the authors did not provide a clear rationale for selecting such stringent cutoffs. These choices could significantly impact downstream analyses, such as the number of clusters or cell types identified in *Fosl2^{F/F}* and *Fosl2^{cKO}*. Thirdly, there is no explanation of whether batch effect correction was considered or applied to account for potential technical variability. For instance, it is unclear whether the *Fosl2^{F/F}* and *Fosl2^{cKO}* libraries were prepared under the same conditions or sequenced on different days or runs. Finally, the authors did not specify how they optimized the number of clusters in their analysis or which statistical tests they used to evaluate the stability and separation of the clusters. Additionally, it remains unclear how the initial clusters were identified and subsequently annotated as sub-clusters of granulosa cells. The poor distinction between clusters in the heatmap (Extended Data Fig. 9a) of the top five genes suggests a non-standardized scRNA-seq analysis pipeline or poor sequencing quality.

R1.14: Thank you for your informative guidance. We apologize for any inappropriate description regarding our single-cell RNA-seq analysis in the original MS; these comments have significantly improved the transparency of our study.

First of all, we acknowledge and regret the initial omission of cluster-specific marker gene lists in our single-cell analysis. While the ovarian scRNA-seq on *Fosl2^{F/F}* and *Fosl2^{CKO}* constituted a focused yet non-central component of this study, we recognize that transparent reporting of cellular identities is essential for reproducibility. Following dimensionality reduction and clustering by the Seurat algorithm, we identified multiple clusters from the previous ovary study, which were combined to represent major ovarian cell categories²⁶. We have now provided complete marker gene lists for all major clusters (e.g., granulosa cells, theca cells) and subclusters (e.g., antral vs. mural granulosa cells) in Supplementary Tables 6 and 7, ranked by adjusted *p* values and log2 fold change, respectively. Our clustering patterns in our scRNA-seq datasets, regardless of sample size, aligned closely with published ovarian references^{12, 27}.

In regard to the data filtering standard, we sincerely apologize for our inadvertent oversight and the lack of precision in methods proofreading; indeed, cells were filtered out if they did not meet the following criteria: 1) a minimum of 500 detected genes, 2) more than 1,000 detected UMIs, 3) mitochondrial UMI counts constituting less than 10% of the total and 4) hemoglobin gene expression levels below 0.1%. We use the *Fosl2^{F/F}* sample as an example here (Response Fig. 1.4a, b). Following QC filtering, each sample contained approximately 8,000 valid cells, meeting the requirements for scRNA-seq analysis²⁸. The data criteria have been rephrased and clarified in the Methods section (Revised MS, Lines 741-746).

Response Fig. 1.4 | Quality control (QC) in scRNA-seq analysis of *Fosl2*^{F/F}. a, Violin plots and scatter diagram showing the ovarian cells in *Fosl2*^{F/F} before QC a, and after QC b.

Meanwhile, we fully understand your concerns about batch effects in our scRNA-seq data. In fact, we have employed rigorous experimental and computational measures to minimize batch effects in our scRNA-seq analysis. During the experimental design, all samples were processed in parallel by a single operator within a 24-hour period to eliminate technical variability. Libraries were then sequenced on the same NovaSeq 6000 flow cell to ensure uniform sequencing depth and quality. Data integration via canonical correlation analysis (CCA) was conducted following established scRNA-seq integration practices²⁹. The detail for the correction of batch effect has also been added in the Methods section (Revised MS, Lines 636-638).

In regard to the identification of initial clusters, we are sorry for the omission of the clustering workflow in the original MS. Indeed, using Seurat's graph-based clustering algorithm, we systematically tested resolution parameters from 0.06 to 0.5. The clustree package (v0.5.0) was employed to visualize hierarchical relationships between clusters across different resolutions (Response Fig. 1.5a). At a resolution of 0.3, we observed optimal separation of biologically distinct populations while maintaining cluster stability (Response Fig. 1.5b). This parameter yielded distinct clusters within the GC population, consistent with established ovarian follicle differentiation trajectories^{12, 26, 27, 30}. Moreover, the UMAP visualization also confirmed spatial separation with min_dist set at 0.3, which optimized the preservation of local structure. This multi-modal approach ensures that our sub-clusters reflect biologically meaningful GC subtypes rather than technical artifacts. The details for cluster identification have now been added in the Methods section (Revised MS, Lines 747-750).

Response Fig. 1.5 | The identification of clusters in scRNA-seq analysis for murine ovaries. a, Cluster tree showing the relationship between resolution and the number of clusters. **b,** UMAP plot featuring different cell clusters of ovaries in 2-month-old female mice. Clustering analysis revealing distinct ovarian cell populations.

As for the heatmap resolution in Extended Data Fig. 9a, which has now been moved to Extended Data Fig. 8a, we have increased min.pct from 0.1 to 0.5 as explained in detail in R1.3. This adjustment has increased the marker stringency, demonstrating distinct expression patterns for key lineage markers. Once again, we apologize for any inconvenience caused by the previous descriptions of our scRNA-seq analysis and appreciate your kind suggestions for strengthening our study.

Q1.15: ATAC-seq and CUT&Tag Data: In the analysis of these epigenetic datasets, the authors did not specify whether blacklist regions or unwanted reads, such as singletons and duplicates, were removed prior to analysis. Notably, high-signal regions, particularly those in blacklist regions, can introduce significant bias and directly impact the accuracy of peak calling when using tools such as MACS2. Furthermore, with respect to the ATAC-seq data, the authors reported that “the ATAC-seq peaks of each sample were merged to generate a consensus set of unique peaks” (Methods, line 671). This raises concerns regarding the reproducibility of the reported peaks, as no evidence was provided to confirm their consistency across samples. Moreover, the authors did not justify their choice of a p-value threshold of 0.05 for peak calling, nor did they evaluate stricter thresholds, such as 0.01 or 0.03, which would result in varying numbers of identified signals. The lack of optimization or validation to establish the most appropriate threshold leaves ambiguity about the reliability of the selected approach. Given that this analysis directly affects downstream results, including GAAs and GIAs, it is strongly recommended to identify reproducible peaks across biological replicates using the Irreproducibility Discovery Rate (IDR) method. This is a widely recognized and reliable method for objectively evaluating the reproducibility of high-throughput assays.

R1.15: Thank you for your careful reading and the professional comments on the details of the epigenetic data analysis. We fully understand your concerns about these issues, and they are highly significant for improving our research.

Regarding the removal of unwanted reads, we apologize for not explicitly mentioning these details in our initial submission due to the lack of a consistent blacklist: in our study, we have followed the standard analysis procedures for ATAC-seq and CUT&Tag data, where blacklist filtering was the default step. For the mouse data, step for removing the blacklist regions was applied using the ENCODE blacklist regions (mm10)³¹. However, for the pig data, there is currently no widely accepted and well-established blacklist for the pig genome, despite a preliminary and unverified version being recently published in May of this year³². This made it impossible to process the pig data the same way as the mouse data. To prevent confusion, we did not therefore emphasize blacklist-based read removal in the original MS. Even so, we have subjected strict quality control on raw data and carefully checked read distributions to make up for the lack of a pig genome blacklist. We alternatively excluded regions homologous to the mm10 blacklist and removed peaks with abnormally high signals ($>10\times$ median coverage) in IgG controls for pig data. As a result, the ATAC-seq data displayed a bimodal fragment-size distribution, with peaks in the nucleosome-free region and mononucleosome-protected region, which indicated the high efficiency Tn5 enzyme without the disturbance from unwanted regions (Revised Extended Data Fig. 2a, d). In response to your question, we have rephrased the Methods section to include the removal of blacklist regions in the revised MS (Revised MS, Lines 709-711). We apologize again for any misunderstanding caused by our negligence.

In regard to the misunderstanding for the choice of threshold for peak calling, we recognize that the original MS's juxtaposition of statistical thresholds for peak calling (q value) and differential analysis (p value) in adjacent sentences may have accidentally obscured the distinction between these two distinct analytical steps. Indeed, we clarify that the step via MACS2 actually used a q value threshold of < 0.05 , not a raw p value threshold. This choice explicitly controls the false discovery rate (FDR) at 5%, which is a widely adopted pipeline in ChIP-seq and ATAC-seq analysis that aligns with ENCODE standards. This structural proximity, combined with our insufficient emphasis on their methodological differences, could have led to potential misunderstandings. We have now rigorously separated these descriptions in the revised Methods section, and sincerely thank you for your attention to this issue.

As for the concern regarding consensus peaks we merged in our analysis, we fully agree with this comment and recognize the advantages of IDR method in evaluating repeatability. Indeed, in preliminary analyses, we once performed IDR-based peak merging (v2.0.4) using ENCODE-recommended parameters. Venn diagram comparisons revealed that IDR-derived GAAs and GIAs formed subsets of the BEDTools-merged consensus peaks, indicating higher stringency, but potential biological signal loss (Response Fig. 1.6). Alternatively, the combined peak using BEDTools merge can capture regions significantly open in at least one sample, thereby minimizing the risk of overlooking low-frequency yet functionally important sites, and this method has been adopted in the vast majority epigenomic studies, including the recently published ones ^{33, 34, 35, 36}. Moreover, our datasets have exhibited high reproducibility across biological replicates at robust sequencing depth. The mapping ratio to the porcine or murine genome exceeded 85% for all samples, with a NRF > 80%, indicating minimal PCR amplification bias. Most of FRiP ranged from 25% to 32%, consistent with ENCODE standards of $\geq 20\%$ in open chromatin analysis, further validating the signal-to-noise ratio of our datasets. Additionally, the potential batch effects were assessed using PCA also emphasized the reproducibility of our data (Revised Extended Data Fig. 2c). To improve the transparency of our study, we have summarized the quality control in the uploaded Supplementary information (Supplementary Table 4). The corresponding quality control process has also been rephrased in the Methods section (Revised MS, Lines 705-709).

Response Fig. 1.6, Venn diagram showing the overlap between peaks identified with IDR methods and BEDtools in GAAs and GIAs.

We fully acknowledge your suggestion regarding the use of IDR for peak calling, as it provides a rigorous statistical framework to minimize false positives. While IDR-based peak correction would not alter the fundamental conclusions, however, implementing IDR at this stage would require reprocessing the raw data from the initial

analysis, which is computationally intensive and time-consuming. Given that our study is in its final stages and all downstream analyses are based on the current peak set, reanalysis could significantly delay the key findings of *Fosl2* in this field. Meanwhile, we are confident our alternative method is well-supported, based on the above-mentioned data reproducibility. We have also provided raw alignment files and processing scripts for peers to reproduce the analysis with IDR, uploaded in NCBI's Gene Expression Omnibus (GSE267974). Once again, we sincerely appreciate your insightful comments and constructive suggestions, which have significantly strengthened the rigor and clarity of our study!

Q1.16: RNA-seq data: The authors have not provided documentation to demonstrate the quality control (QC) of the RNA-seq data. Additionally, there are no plots, such as principal component analysis (PCA), to show the separation of samples from different follicular states (germinal vesicle and MII in oocytes, antral and ovulatory phases in porcine granulosa cells), which would indicate transcriptomic differences among these groups. It appears that the authors have applied a standard RNA-seq pipeline, but there are some concerns with this approach. Conventional normalization methods generally address sequencing depth but fail to correct for other technical biases, such as those introduced during library preparation. This issue can be more pronounced in tissues like the ovary, where hormonal levels fluctuate, and small hormonal changes can directly contribute to variation in gene expression. Therefore, I strongly recommend performing an additional normalization step (methods such as remove unwanted variation, RUVseq) after transcript quantification to remove unwanted biases and produce normalized counts, followed by differential expression analysis to identify DEGs in each comparison.

R1.16: Thank you for your informative suggestion and comment to strengthen the RNA-seq analysis. We fully agree with this comment and sincerely regret the omission of the RNA-seq data quality control (QC) in the original MS. In our study, we have conducted a rigorous QC procedure for the RNA-seq data. During data generation, FastQC was used to assess the quality of the raw sequencing data. We examined several indicators, including base quality distribution, read length distribution, and GC content. Reads with low quality (reads where bases with a quality score of less than Q20

accounted for more than 50%) and those contaminated with adapters were removed. The summary of the sequencing data including quality control values including Q30 value, mapping rate, GC content has been provided in the uploaded Supplementary information (Supplementary Table 4). The corresponding quality control process has also been rephrased in the Methods section (Revised MS, Lines 680-683).

We are also sorry for not including the PCA plot of RNA-seq in the original MS. Indeed, the PCA plot has displayed distinct clustering of samples across different follicular maturation phases, indicating significant transcriptomic differences thereby further validating the quality and reliability of our research. As a result, this plot has been added in the revised MS to better illustrate the transcriptomic variations among samples (Revised Extended Data Fig. 1a). In addition, your concerns about the technical biases for RNA-seq data analysis are highly valuable. As you pointed out, we fully agree that conventional normalization methods have limitations, especially in tissues like the ovary where hormone levels fluctuate. As per your suggestion, we have applied the RUVseq method to eliminate unwanted variation and conduct further normalization. This includes the heatmap modification you highlighted in Q1.4. Corresponding analyses of DEGs and associated assessments, along with detailed RNA-seq analysis methodologies, have been rephrased in the revised MS (Revised MS, Lines 685-693). The updated list of DEGs during follicular maturation has also been provided in the uploaded Supplementary information (Supplementary Table 2). Again, thank you for the constructive comments to advance the understanding of our work!

Minor issues:

Q1.17: Line 37 missing reference: “In terms of reproduction, GCs...”. It would be better to include the criteria used for determining the DEGs either directly on the figure or in the figure legend (Extended Data Figure 1a).

R1.17: Thank you for your helpful suggestions. We have added the relevant references in the revised Introduction section (Revised MS, Line 39): *Guo J, et al. Oocyte stage-specific effects of MTOR determine granulosa cell fate and oocyte quality in mice. Proc Natl Acad Sci U S A 115, e5326-5333 (2018)*. Following your suggestion, the determination of DEGs has been added in the revised Figure legends (Revised MS, Lines 1176-1177). Meanwhile, the full list of DEGs is accessible in the Supplementary Information (Supplementary Table 2).

Q1.18: Lines 106-109 (related to Figure 1d): Some references need to be added. Based on which references are the genes referred to as 'oocyte support genes'? Also, what do the dashed lines in the figure represent?

R1.18: Thank you for highlighting this; we sincerely appreciate your attention to this detail. Studies have illustrated that oocytes lacking GCs exhibit extensive transcriptional changes compared to normal oocytes, involving many key genes during oocyte maturation, consistent with our findings in Revised Fig. 1d. These genes, expressed in oocytes, are essential for maintaining oocyte growth, metabolic support, and meiotic competence. In accordance with your suggestion, we have added the relevant references to better support our conclusion (Revised MS, Line 111): (1) *Gilchrist RB, Lane M, Thompson JG. Oocyte-secreted factors: regulators of cumulus cell function and oocyte quality. Hum Reprod Update 14, 159-177 (2008).* (2) *Zhang M, Su YQ, Sugiura K, Xia G, Eppig JJ. Granulosa cell ligand NPPC and its receptor NPR2 maintain meiotic arrest in mouse oocytes. Science 330, 366-369 (2010).* (3) *Zhang Y, et al. Transcriptome Landscape of Human Folliculogenesis Reveals Oocyte and Granulosa Cell Interactions. Mol Cell 72, 1021-1034.e1024 (2018).* In Fig. 1d, the vertical dashed line indicates the threshold for identifying upregulated (left side) or downregulated (right side) gene expression, while the horizontal dashed line shows the statistical significance threshold. This pattern is consistent with standard practices in RNA-seq volcano plot analysis, and we have added a detailed description in the figure legend to enhance readability (Revised MS, Lines 1140-1142).

Q1.19: Line 112: What does 'overexpressed DEGs' mean in the context of a typical transcriptomic comparison between two given conditions? What are those genes? Would it be possible to provide a table?

R1.19: Thank you for your patient reading. The term “overexpressed DEGs” used in Extended Data Fig. 1f essentially denotes a functionally gene set of GC-involved developmental genes (GDGs) that are overexpressed at the ovulatory phase in pGCs. In the revised Extended Data Fig. 1d and e, we focus on the signaling processes associated with differentially expressed genes (DEGs) in GCs, where we observed that those overexpressed genes in GCs are consistently implicated in folliculogenesis pathways (Revised Extended Data Fig. 1f). Subsequent analyses revealed that these

GDGs exhibit unique regulatory mechanisms. We fully understand your concern regarding this terminology; however, to preserve the logicity and readability, we deliberately deferred the introduction of the GDG concept to a later section (Revised Fig. 2). We also recognize that our initial description might have caused confusion, and we have rephrased sentences in the revised MS as follows: “*In particular, those overexpressed genes at the ovulatory phase were predominantly engaged in folliculogenesis pathways, reinforcing the notion that the rewiring of a developmental gene subset occurred during follicular maturation (Extended Data Fig. 1f, g).*” (Revised MS, Lines 113-116). Additionally, a catalog of GDGs has been included in Supplementary information (Supplementary Table 3).

Q1.20: Figure 3h: The terms 'shared' (upper panel), 'OP/AP up only,' and '7829' (lower panel) are unclear. Additionally, earlier in line 142, the number of GAAs regions is introduced as 8646, but in Figure 3h (upper panel), the GAAs-specific regions are listed as 26,577. It would be helpful if the author could clarify why there is such a significant discrepancy here. Thirdly, while the text (lines 217-220: 'To further...') clearly explains the overlap of GAAs with Fosl2-gained peaks, the figure is somewhat misleading, as Fosl2 is neither shown in the figure nor mentioned in the legend.

R1.20: Many thanks for your careful reading and valuable comments! We sincerely apologize for the inaccuracies in the original labeling of Fig. 3h. The goal of this panel was to compare the dynamics of GAA peaks with Fosl2 occupancy by analyzing their co-occurrence. Specifically, we performed overlap analysis to evaluate whether GAA-containing regions represent precise targets of Fosl2-gained peaks (Revised Fig. 3h, upper panel). We acknowledge the confusion caused by the term “OP/AP up only”, and this label was intended to denote Fosl2 CUT&Tag peaks that were upregulated. Meanwhile, we also regret the error in reporting the number of GAA regions, which arose from a labeling inversion during data annotation. Indeed, the correct value for GAA-specific regions is 2,088 (originally mislabeled as “OP/AP down only”), not 26,557 as stated in the original MS. Therefore, as the reviewer pointed out, we have used more intuitive terminology to label the data to avoid similar misunderstandings (Revised Fig. 3h). In addition, the labels on the other graphs have also been comprehensively checked. Thank you again for your invaluable reminder.

Revised Fig. 3h, Bar plots displaying the distribution of differential FosL2 binding signals with GAAs, Intersection numbers are labeled.

Q1.21: The idea of randomly selected genes is not clear. In addition, how did the authors decide to select the random genes?

R1.21: Thank you for your valuable comments. Indeed, we randomly selected an equal number of protein-coding genes ($n = 2068$) that are expressed in GCs during follicular maturation, with an FPKM > 1 in the RNA-seq data, to serve as GDG controls. To ensure the comparability of these gene sets, we performed Kolmogorov-Smirnov tests, which confirmed no significant differences ($p > 0.05$) between the antral and ovulatory phases. As requested, we have detailed the randomization protocol in the revised Methods section to ensure transparency (Revised MS, Lines 701-703).

Q1.22: Line 177: It should be 'Fshr' instead of 'frsh'.

R1.22: Thank you for your careful reading! We apologize for the typo and have corrected it in the revised MS. In addition, the spelling throughout the MS has also been comprehensively checked to ensure accuracy.

Q1.23: For the motif analysis, there is a discrepancy: line 189 mentions the HOMER algorithm, whereas the legend for Figure 3a refers to the HOMER2 algorithm.

R1.23: Many thanks for your patient reading! The motif analysis among GAAs and GIAs was determined using HOMER algorithms, not HOMER2 algorithms. We are sorry for the mistake in the figure legend and have corrected it in the revised MS.

Q1.24: Lines 253 and 245 require references for the mention of TOBIAS tools and “unidentified pluripotency TF motifs...”.

R1.24: Thank you for your helpful suggestions. We sincerely apologize for the omission of citations that may have failed to fully acknowledge the contributions of other researchers, and we have added the relevant references to the revised MS (Revised MS, Line 249): (1) *Bentsen M, et al. ATAC-seq footprinting unravels kinetics of transcription factor binding during zygotic genome activation. Nat Commun 11, 4267 (2020).* (2) *Zhu M, et al. Developmental clock and mechanism of de novo polarization of the mouse embryo. Science 370, eabd2703 (2020).* (3) *Jang J, Wang Y, Kim HS, Lalli MA, Kosik KS. Nrf2, a regulator of the proteasome, controls self-renewal and pluripotency in human embryonic stem cells. Stem Cells 32, 2616-2625 (2014).*

Q1.25: Extended figure 6a. It has not been mentioned what data have been used to create the heatmap.

R1.25: Thank you for your insightful question regarding the gene selection in Extended Data Fig. 6e. The heatmap aims to visually support our finding that “a plethora of GDGs were significantly downregulated after *Fosl2* knockdown compared with randomly selected genes” (Revised Fig. 5c). We therefore selected several representative and significant GDGs ($p < 0.05$) that were in proximity to GAA motifs (± 50 kb), including those well-described in Fig. 2e, to show their expression after *Fosl2* knockdown. These partially selected examples were shown to align with our focus on *Fosl2*-mediated transcriptional regulation. In addition, the catalog of GDGs mentioned in Q1.7 and Q1.12, is accessible in Supplementary information for transparency and reproducibility (Supplementary Table 3).

Q1.26: Figure 6d: It is not clearly indicated whether the IF figures represent antral or ovulatory follicles.

R1.26: Thank you for your informative guidance, and we apologize for any confusion caused by the inappropriate descriptions in the original Figure labels. For Fig. 6d, our data clearly demonstrated that Fosl2 exhibits intense fluorescence during the ovulatory phase, in contrast to the low signal observed in mGCs at the antral phase (Revised Fig. 6d). As for Fig. 6e, Fosl2 fluorescence in oocytes within pre-antral phase follicles (originally labeled as secondary follicles) shows strong signals, which then diminish in later pre-ovulatory phase follicles. To ensure consistency and improve readability, we have rewritten the corresponding sentences and figure legend in the revised MS (Revised MS, Lines 324-326, 1125-1126).

Q1.27: Line 309: “a representative folliculogenesis biomarker, Mapk1” need reference.

R1.27: Thank you for your valuable comments. We have added the relevant reference in the revised MS (Revised MS, Line 317): *Choi J, Jo M, Lee E, Choi D. ERK1/2 is involved in luteal cell autophagy regulation during corpus luteum regression via an mTOR-independent pathway. Mol Hum Reprod 20, 972-980 (2014).*

Q1.28: Line 360: Missing reference for the statement: “using the Seurat algorithm...”

R1.28: Thank you for your friendly reminder. The main processing steps using Seurat (v5.0.2) have been rephrased in the Methods section and the relevant reference regarding the Seurat algorithm has been added in the revised MS (Revised MS, Line 742).

Q1.29: Line 421: Missing reference for the statement: “strongly supported previously identified regulatory elements...”

R1.29: Thank you for highlighting this! We apologize for the oversight and have added the relevant reference in the revised MS (Revised MS, Line 429): *Fan X, et al. Integrated single-cell multiomics analysis reveals novel candidate markers for prognosis in human pancreatic ductal adenocarcinoma. Cell Discov 8, 13 (2022).*

Q1.30: Line 431: Missing reference for the statement: “however, most research has concentrated on highly expressed TFs...”

R1.30: Many thanks for your careful reading. The relevant references have been added in the revised MS (Revised MS, Line 437): (1) *Eferl R, Wagner EF. AP-1: a double-edged sword in tumorigenesis. Nat Rev Cancer 3, 859-868 (2003).* (2) *Shaulian E, Karin M. AP-1 as a regulator of cell life and death. Nat Cell Biol 4, E131-136 (2002).* (3) *van Dam H, Castellazzi M. Distinct roles of Jun: Fos and Jun: ATF dimers in oncogenesis. Oncogene 20, 2453-2464 (2001).*

Q1.31: Line 450: Missing reference for the statement: “Of note, the expression of c-Fos TFs in the ovary is required for ovulation and folliculogenesis.”

R1.31: Thank you for carefully reading the MS, and we have added the citation in the revised MS (Revised MS, Line 456): *Choi Y, Rosewell KL, Brännström M, Akin JW, Curry TE, Jr., Jo M. FOS, a Critical Downstream Mediator of PGR and EGF Signaling Necessary for Ovulatory Prostaglandins in the Human Ovary. J Clin Endocrinol Metab 103, 4241-4252 (2018).*

Q1.32: Line 459: Missing reference for the statement: “Although Fosl2 deletion in mice is lethal, impacting the development of the heart, bone...”

R1.32: Thanks for your helpful comment! We have added the relevant reference in the revised MS (Revised MS, Line 471): *Eferl R, Zenz R, Theussl HC, Wagner EF. Simultaneous generation of fra-2 conditional and fra-2 knock-out mice. Genesis 45, 447-451 (2007).*

Q1.33: "Lines 653 lacks a reference: 'package RSEM...'"

R1.33: Many thanks for your patient reading! The reference has been added in the revised MS (Revised MS, Line 685). Once again, thank you for the constructive comments to advance the understanding of our work!

Reviewer #2 (Remarks to the Author):

Summary:

In this article, the authors systematically characterized the reorganization of chromatin accessibility that underpins extensive transcriptomic alterations in granulosa cells (GCs) across various stages of follicular maturation in murine and porcine models, and identified the transcription factor *Fosl2* as a key regulator, orchestrating the dynamics of chromatin accessibility to regulate GCs function during follicular maturation. Moreover, I appreciate the authors for their thorough efforts in elucidating the role of the transcription factor *Fosl2* in granulosa cells (GCs) during follicular maturation, particularly through the use of highly reliable 'wet lab' data. Especially, their work is valuable in the context of the mouse models with GC-specific *Fosl2* deletion, which adds significant depth to our understanding of the molecular mechanisms involved. The quality of the presentation is excellent, and the structure of the presentation is clear. Overall, the conclusions appear sound and reliable. However, there are several issues that should be addressed before publication in *Nature Communications*. Especially, the detailed description of the quality of relevant omics data is lacking in current manuscript, which largely reduce the confidence in the related conclusions.

We are truly grateful for the considerable time and great efforts you have invested in reviewing our research, which has been invaluable in enhancing the quality of our work! Based on your insightful suggestions, we have carefully revised and supplemented our previous MS, and the specific revisions are outlined below.

Q2.1: For Result-1 and Fig. 1. The authors showed GCs undergo globally distinct chromatin landscape remodeling during follicular maturation using porcine model. But there was no the description of the quality of relevant omics data, which can increase the credibility of the results, such as mapping ratio, non-redundant read fraction (NRF), fraction of reads in peaks (FRiP) and others. Given that, the authors should check the whole manuscript and revised it.

R2.1: We sincerely appreciate your insightful comments. Your suggestions regarding the additional data quality metrics are critical for enhancing the credibility of our results. As you pointed out, we fully recognize that data quality is fundamental to omics analysis. Indeed, the ATAC-seq datasets in revised Fig. 1 exhibit high reproducibility

across biological replicates at a robust sequencing depth. The mapping ratio to the porcine or murine genome exceeded 85% for all samples, with a NRF > 80%, indicating minimal PCR amplification bias. Most FRiP values ranged from 25% to 32%, consistent with ENCODE standards of $\geq 20\%$ in open chromatin analysis, further validating the signal-to-noise ratio of our datasets. Potential batch effects were assessed using PCA, which also emphasized the reproducibility of our data (Revised Extended Data Fig. 1a, 2c). The summary of the sequencing data, including quality control values mentioned above, has been provided in the uploaded Supplementary information (Supplementary Table 4). The corresponding quality control process has also been rephrased in the Methods section (Revised MS, Lines 711-714). We are committed to ensuring data transparency and reproducibility, which are crucial to scientific research. Thank you again for highlighting this important issue!

Q2.2: For Lines 170-171. The authors described “Conversely, the pathways enriched in genes adjacent to GIAs were significantly associated with dysregulated development and exhausted growth capacity (Fig. 2d)”. But Fig. 2d exhibited some pathways (e.g., ‘Signal transduction’, ‘Intracellular signal transduction’, ‘Regulation of neuron migration’, ‘Regulation of postsynaptic density assembly’, and others). How to understand their roles in dysregulated development and exhausted growth capacity?

R2.2: Thank you for raising this valuable question. We selected the example pathways from Fig. 2d to illustrate the link between the enriched pathways of genes near GC-inactivated accessibility regions (GIAs) and those related to developmental dysregulation and growth capacity depletion. Signal transduction and intracellular signal transduction are essential mechanisms enabling GCs to sense their microenvironment and regulate proliferation and differentiation during folliculogenesis^{37, 38}. For example, abnormal activation or inhibition of pathways like WNT and MAPK can disrupt the spatiotemporal coordination of cell fate during follicular development, leading to premature differentiation of GCs and hindering follicular maturation³⁹. As for the pathway regulating neuronal migration, studies show that in non-neural tissues, migration-related genes may participate in processes such as epithelial-mesenchymal transition (EMT). Furthermore, dysregulation of these genes could result in structural abnormalities in the granulosa cell layer surrounding the

oocyte, impairing nutrient exchange⁴⁰. Additionally, synaptic dysfunction can weaken intercellular communication through postsynaptic density proteins, potentially causing similar developmental arrest of oocytes and accelerating follicular atresia^{41, 42}. In conclusion, the pathways in Revised Fig. 2d, though seemingly diverse, are all key nodes in the network of cellular responses to external signals and internal homeostasis regulation. The aforementioned signaling pathways are crucial throughout folliculogenesis. Notably, during the ovulatory phase, GCs cease proliferation and undergo expansion, and the enrichment of these pathways near GIAs mirrors the GCs state at this phase^{38, 43}. We sincerely appreciate your pointing out this issue and have supplemented part of the citation discussed above in the revised MS to enhance readability (Revised MS, Lines 171-174).

Q2.3: For Lines 222-224. The authors described “Distal Fosl2 binding peaks were also predominantly associated with upregulated genes, revealing a principal role for Fosl2 as a transcriptional activator modulating GAA remodeling (Fig. 3i)”. As we know, the three-dimensional genome architecture (i.e., topologically associating domains [TADs]) can promote/inhibit interaction between promoters and distal enhancers. Given that, I recommended the authors can investigate the relationship among TADs, Distal Fosl2 binding peaks and the promoters of its nearest genes, based on potential public Hi-C data.

R2.3: Thank you for your very constructive and insightful suggestions! The reviewer has raised a fundamental question regarding the validation of chromatin spatial interactions in our study. We fully acknowledge the importance of high-resolution Hi-C data for investigating the distal regulatory role of Fosl2. In fact, we initially attempted to integrate publicly available Hi-C datasets from mammalian GCs to support our findings; however, our systematic investigation revealed that existing Hi-C studies in ovarian cells predominantly focus on oocytes^{33, 44, 45}. In addition, current oocyte Hi-C analyses typically adopt resolutions of 10-20 kb, which may be insufficient to observe the interaction between TF and promoter⁴⁶. Therefore, this limitation directly motivated our alternative approach by employing 3C-qPCR to test the hypothesis that Fosl2 binds distal regulatory elements to physically interact with GDG promoters (e.g., *Cyp11a1*, *Inhba*). Following the adapted 3C protocols, six primer pairs were designed to flank predicted Fosl2 binding sites identified by CUT&Tag in pGCs, focusing on regions >

80-150 bp from the nearest restriction enzyme digestion site (Response Fig. 2.1a) 47. Quantitative analysis of 3C-qPCR libraries revealed that interaction frequencies between *Fosl2* binding sites and GDG promoters were significantly enriched compared to negative genomic DNA. Moreover, gel electrophoresis of ligation products further confirmed that interactions spanned linear distances >2 kb, consistent with chromatin loop formation (Response Fig. 2.1b). Our 3C-qPCR approach, to some degree, provides insight into how distal *Fosl2* modulates GDG expression during follicular maturation. We agree that these findings would be strengthened by high-resolution Hi-C approaches, such as HiChIP and promoter-capture Hi-C, and we sincerely suggest that future studies provide TF perturbation experimental and analytical evidence to fully resolve loop dynamics in GCs. We sincerely hope that these explanations address your concerns.

Response Fig. 2.1 | *Fosl2* binds distal regulatory elements to physically interact with GDG promoters. **a**, Location diagram of 3C-qPCR primers within *Cyp11a1* and *Inhba* loci of pGCs at the ovulatory phases. **b**, Gel electrophoresis displayed products from genomic DNA control and experimental 3C libraries. Key DNA fragment sizes of the markers were indicated on the left.

Q2.4: For Lines 366-368. The authors described “Although the total captured cell numbers were consistent at the beginning, GC subclusters significantly decreased after *Fosl2* loss (Extended Data Fig. 9c)”. But Extended Data Fig. 9c just exhibited the number of GCs. So, the authors should add the comparative analysis of the cell proportions of major cell types between *Fosl2*^{F/F} and *Fosl2*^{cKO}.

R2.4: Thank you for your insightful guidance. We sincerely apologize for the lack of clarity in our original description of the conclusion drawn in the MS. In response to your suggestion, we have added statistical analyses comparing the proportions of major cell types in *Fosl2*^{F/F} and *Fosl2*^{cKO} mice (Revised Extended Data Fig. 8c). This result demonstrates that the proportion of GCs decreases from 49.21% in controls to 33.82%,

which further supports our conclusion that GC subclusters are significantly reduced following *Fosl2* ablation. This result has now been synchronized and updated in the revised MS. Thank you again for your kind comments!

Extended Data Fig. 8c, Numbers and percentages of major cell clusters in *Fosl2*^{F/F} and *Fosl2*^{cKO} mice.

Q2.5: For Bulk RNA-seq data processing in Methods section. The authors performed R package DESeq2 to obtain DEGs with log₂ (fold-change) ≥ 1 and p value < 0.05. However, the reported studies generally used adjusted p value as cutoff to minimize false positive results. So, the authors should clarify why using p value, and not adjusted p value.

R2.5: Thank you very much for your careful reading and constructive suggestion! We fully agree with your comment and sincerely apologize for our inadvertent oversight and the lack of precision in our description of the DEG analysis, which may have led to misunderstandings. Indeed, in our research, the DEGs for bulk RNA-seq were identified using DESeq2 (v1.24.0) with adjusted p values, consistent with the analysis applied to the DARs identified in ATAC-seq data⁴⁸. We fully acknowledge, as you pointed out, that adjusted p values are essential for controlling the false positive rate in multiple testing, particularly in high-throughput data analysis. We have carefully corrected the typographical errors in the Methods section, and thoroughly reviewed the entire MS to avoid similar inaccuracies. In addition, the DEGs data, including adjust p values, have also been provided in the uploaded Supplementary information (Supplementary Table 2). We deeply regret this error and appreciate your attention to this important detail. Once again, thank you for your valuable efforts in advancing our research!

References

1. Yuan Y, *et al.* Quadrupling efficiency in production of genetically modified pigs through improved oocyte maturation. *Proceedings of the National Academy of Sciences of the United States of America* **114**, E5796-E5804 (2017).
2. He H, *et al.* Selective autophagic degradation of ACLY (ATP citrate lyase) maintains citrate homeostasis and promotes oocyte maturation. *Autophagy* **19**, 163-179 (2023).
3. Kang JT, *et al.* Effects of melatonin on in vitro maturation of porcine oocyte and expression of melatonin receptor RNA in cumulus and granulosa cells. *Journal of pineal research* **46**, 22-28 (2009).
4. Zheng Q, *et al.* Reconstitution of UCP1 using CRISPR/Cas9 in the white adipose tissue of pigs decreases fat deposition and improves thermogenic capacity. *Proceedings of the National Academy of Sciences of the United States of America* **114**, E9474-E9482 (2017).
5. Zhang H, *et al.* Rescuing ocular development in an anophthalmic pig by blastocyst complementation. *EMBO molecular medicine* **10**, e8861 (2018).
6. Pei Z, Deng K, Xu C, Zhang S. The molecular regulatory mechanisms of meiotic arrest and resumption in Oocyte development and maturation. *Reprod Biol Endocrinol* **21**, 90 (2023).
7. Alfradique VAP, *et al.* The effect of age and FSH stimulation on the ovarian follicular response, nuclear maturation, and gene expression of cumulus-oocyte complexes in prepubertal gilts. *Theriogenology* **199**, 57-68 (2023).
8. Umehara T, *et al.* The acceleration of reproductive aging in Nrg1(flox/flox);Cyp19-Cre female mice. *Aging cell* **16**, 1288-1299 (2017).
9. Habara O, Logan CY, Kanai-Azuma M, Nusse R, Takase HM. WNT signaling in pre-granulosa cells is required for ovarian folliculogenesis and female fertility. *Development* **148**, dev198846 (2021).
10. Zhao ZH, Gu LJ, Zhang XG, Wang ZB, Ou XH, Sun QY. Single-cell and spatial transcriptomes reveal the impact of maternal low protein diet on follicular cell composition and ovarian micro-environment in the offspring. *J Nutr Biochem* **136**, 109789 (2025).
11. Buttram VC, Jr. The immature HPO axis. *J Reprod Med* **14**, 21-25 (1975).
12. Isola JVV, *et al.* A single-cell atlas of the aging mouse ovary. *Nat Aging* **4**, 145-162 (2024).
13. Wu J, *et al.* The landscape of accessible chromatin in mammalian preimplantation embryos. *Nature* **534**, 652-657 (2016).
14. Buccitelli C, Selbach M. mRNAs, proteins and the emerging principles of gene expression control. *Nat Rev Genet* **21**, 630-644 (2020).
15. Eferl R, Zenz R, Theussl HC, Wagner EF. Simultaneous generation of fra-2 conditional and fra-2 knock-out mice. *Genesis* **45**, 447-451 (2007).
16. Martínez-Zamudio RI, *et al.* AP-1 imprints a reversible transcriptional programme of senescent cells. *Nat Cell Biol* **22**, 842-855 (2020).
17. Wolf BK, *et al.* Cooperation of chromatin remodeling SWI/SNF complex and pioneer factor AP-1 shapes 3D enhancer landscapes. *Nat Struct Mol Biol* **30**, 10-21 (2023).
18. Roychoudhuri R, *et al.* BACH2 regulates CD8(+) T cell differentiation by controlling access of AP-1 factors to enhancers. *Nat Immunol* **17**, 851-860 (2016).
19. Biddie SC, *et al.* Transcription factor AP1 potentiates chromatin accessibility and glucocorticoid receptor binding. *Mol Cell* **43**, 145-155 (2011).

20. Dinh DT, *et al.* Progesterone receptor mediates ovulatory transcription through RUNX transcription factor interactions and chromatin remodelling. *Nucleic Acids Res* **51**, 5981-5996 (2023).
21. Ling ITC, Sauka-Spengler T. Early chromatin shaping predetermines multipotent vagal neural crest into neural, neuronal and mesenchymal lineages. *Nat Cell Biol* **21**, 1504-1517 (2019).
22. Chen Z, *et al.* In vivo CD8(+) T cell CRISPR screening reveals control by Fli1 in infection and cancer. *Cell* **184**, 1262-1280.e1222 (2021).
23. Lai F, *et al.* NR5A2 connects zygotic genome activation to the first lineage segregation in totipotent embryos. *Cell Res* **33**, 952-966 (2023).
24. Herlands RL, Schultz RM. Regulation of mouse oocyte growth: probable nutritional role for intercellular communication between follicle cells and oocytes in oocyte growth. *J Exp Zool* **229**, 317-325 (1984).
25. Zhang Y, *et al.* Oocyte-derived microvilli control female fertility by optimizing ovarian follicle selection in mice. *Nat Commun* **12**, 2523 (2021).
26. Morris ME, *et al.* A single-cell atlas of the cycling murine ovary. *Elife* **11**, e77239 (2022).
27. Hao Y, *et al.* Dictionary learning for integrative, multimodal and scalable single-cell analysis. *Nat Biotechnol* **42**, 293-304 (2024).
28. AlJanahi AA, Danielsen M, Dunbar CE. An Introduction to the Analysis of Single-Cell RNA-Sequencing Data. *Mol Ther Methods Clin Dev* **10**, 189-196 (2018).
29. Butler A, Hoffman P, Smibert P, Papalexi E, Satija R. Integrating single-cell transcriptomic data across different conditions, technologies, and species. *Nat Biotechnol* **36**, 411-420 (2018).
30. Zhao ZH, *et al.* Single-cell RNA sequencing reveals the landscape of early female germ cell development. *Faseb j* **34**, 12634-12645 (2020).
31. Amemiya HM, Kundaje A, Boyle AP. The ENCODE Blacklist: Identification of Problematic Regions of the Genome. *Sci Rep* **9**, 9354 (2019).
32. Kong M, *et al.* Identification of blacklist regions in cattle and pig genomes. *Genomics* **117**, 111027 (2025).
33. Li D, *et al.* Dynamic transcriptome and chromatin architecture in granulosa cells during chicken folliculogenesis. *Nat Commun* **13**, 131 (2022).
34. Hoelzl S, Hasenbein TP, Engelhardt S, Andergassen D. Aging promotes reactivation of the Barr body at distal chromosome regions. *Nat Aging*, ahead of print (2025).
35. Van Nerum K, *et al.* α -Ketoglutarate promotes trophoctoderm induction and maturation from naive human embryonic stem cells. *Nat Cell Biol*, **27**, 749-761 (2025).
36. Tian Y, *et al.* Histone H1 deamidation facilitates chromatin relaxation for DNA repair. *Nature*, **641**, 779-787 (2025).
37. Gilchrist RB, Lane M, Thompson JG. Oocyte-secreted factors: regulators of cumulus cell function and oocyte quality. *Hum Reprod Update* **14**, 159-177 (2008).
38. Turathum B, Gao EM, Chian RC. The Function of Cumulus Cells in Oocyte Growth and Maturation and in Subsequent Ovulation and Fertilization. *Cells* **10**, 2292 (2021).
39. Wang X, *et al.* Granulosa Cell-Layer Stiffening Prevents Escape of Mural Granulosa Cells from the Post-Ovulatory Follicle. *Adv Sci (Weinh)* **11**, e2403640 (2024).
40. Liu C, *et al.* Granulosa cell mevalonate pathway abnormalities contribute to oocyte meiotic defects and aneuploidy. *Nat Aging* **3**, 670-687 (2023).

41. Baldini GM, *et al.* Genetic Abnormalities of Oocyte Maturation: Mechanisms and Clinical Implications. *Int J Mol Sci* **25**, 13002 (2024).
42. Alder J, Lu B, Valtorta F, Greengard P, Poo MM. Calcium-dependent transmitter secretion reconstituted in *Xenopus* oocytes: requirement for synaptophysin. *Science* **257**, 657-661 (1992).
43. Coticchio G, *et al.* Oocyte maturation: gamete-somatic cells interactions, meiotic resumption, cytoskeletal dynamics and cytoplasmic reorganization. *Hum Reprod Update* **21**, 427-454 (2015).
44. Ke Y, *et al.* 3D Chromatin Structures of Mature Gametes and Structural Reprogramming during Mammalian Embryogenesis. *Cell* **170**, 367-381.e320 (2017).
45. Hug CB, Grimaldi AG, Kruse K, Vaquerizas JM. Chromatin Architecture Emerges during Zygotic Genome Activation Independent of Transcription. *Cell* **169**, 216-228.e219 (2017).
46. Liu C, *et al.* Granulosa cell mevalonate pathway abnormalities contribute to oocyte meiotic defects and aneuploidy. *Nat Aging* **3**, 670-687 (2023).
47. Hagège H, *et al.* Quantitative analysis of chromosome conformation capture assays (3C-qPCR). *Nature Protocols* **2**, 1722-1733 (2007).
48. Love MI, Huber W, Anders S. Moderated estimation of fold change and dispersion for RNA-seq data with DESeq2. *Genome Biol* **15**, 550 (2014).

Point-by-point answers

Reviewers' comments:

Reviewer #1 (Remarks to the Author):

Summary:

The revised manuscript addresses several of the technical concerns raised in my initial review, and I would like to acknowledge the authors' thoughtful responses and the additional data provided to improve the clarity and transparency of the genomic analyses. These efforts are appreciated. That said, several key issues related to the robustness and reproducibility of the epigenomic analyses remain insufficiently addressed, and I believe they still warrant further consideration before the conclusions can be fully supported.

We sincerely appreciate the time and effort you dedicated to reviewing our MS. Your insightful feedback has significantly improved our work. In response, we have carefully revised the MS based on your suggestions, and the specific changes are outlined below.

Q1.1: In response to Comment 1.10 and Figure 4, the revised manuscript presents heatmaps and profile plots suggesting a role for Fosl2 in chromatin accessibility. While these visualizations are informative, the central conclusion would benefit from additional quantitative analyses supported by appropriate statistical tests. This would help reinforce the claim and provide a stronger evidentiary basis for Fosl2's proposed function. The inclusion of summary statistics (such as average signal intensities with boxplots) would enhance interpretability and address potential concerns regarding signal variability. The Spearman's correlation analyses presented offer some insight into global signal trends, but the strength of the correlation still appears modest. As such, the conclusion that Fosl2 is a key

determinant of open chromatin might currently appear somewhat overstated. Presenting this as a trend, rather than a definitive conclusion, or complementing the analysis with additional quantitative comparisons, would improve the balance between interpretation and evidence.

R1.1: We sincerely appreciate your constructive comments on this important issue. With respect to the concern regarding the inquiry for the visualization of Fosl2's impact in revised Fig. 4, we would like to clarify that we have already provided macro-scale quantification of global and GAA-proximal signal intensities after Fosl2 knockdown (Revised Fig. 4a and c). In-depth analyses quantify how the loss of Fosl2 reshapes the binding landscapes of downstream transcription factors (Revised Fig. 4b and d). Furthermore, our multi-omics integration quantitatively demonstrates the dynamic influence of Fosl2 on chromatin accessibility during follicular maturation and following Fosl2 knockdown by measuring the signal intensities (Revised Fig. 4e). At the locus-specific level, representative GDG examples (e.g., *Cyp11a1*) corroborate these findings, while we employ additional quantitative analyses including box plots and immunofluorescence assays to further substantiate the role of Fosl2 in subsequent Fig.5 (Revised Fig. 4f and Fig. 5). Our analytical approach has aligned with prevailing practices in multi-omics studies, focusing on core aspects commonly emphasized and adhering to the principle of storytelling progression as well. We sincerely appreciate your emphasis on rigorous evidence and have incorporated this perspective into our interpretations in revised MS (Revised MS, Lines 236-239).

In regard to the correlation analyses, we fully agree that Fosl2 is not the sole determinant of chromatin accessibility during folliculogenesis, and we sincerely apologize if our original phrasing overstated its exclusivity. For example, we have discussed how other regulatory factors (e.g., Fosl1, as addressed in Response 1.8) might compensate for or cooperate with Fosl2 during follicular maturation. Building on these findings, we are now launching exploratory studies using *Fosl1^{CKO}* models to ultimately unravel the broader regulatory dynamics of the AP-1 family during follicular maturation in the future. Corresponding discussion has been added to the Discussion section (Revised MS, Lines 464-478). To prevent potential overinterpretation, we have moved the Spearman's correlation analyses to the Extended Data with modified commentary (Revised MS, Lines 239-242). Moreover, we also systematically revised the MS to replace definitive terms (e.g., exclusive) with nuanced language emphasizing

the contributory role of *Fosl2*. Once again, we thank you for your insightful suggestions, which have significantly strengthened the evidentiary balance throughout our work, and we sincerely hope that these explanations address your concerns.

Q1.2: The authors have addressed the lack of a standardized pig genome blacklist by applying various filtering strategies, including the use of mm10 blacklist homologs. While this effort is appreciated, I would respectfully suggest that a direct comparison with the most recent *Sus scrofa* blacklist could further clarify the extent to which artifact-prone regions were addressed. Even a partial or preliminary overlap analysis would enhance transparency and help contextualize the limitations and strengths of the dataset.

R1.2: Thank you for your careful reading and the professional comments on the details. We fully understand your concerns about the lack of application of the latest *Sus Scrofa* blacklist at present. Indeed, we attempted to analyze this preliminary and unverified blacklist, published recently in our prior response. However, our reanalysis revealed noteworthy yet contradictory observations. While certain regions near GDG genes overlap with the proposed blacklist zones (as shown in the reviewer-provided Supplementary materials), the distinct peak profiles in these areas—characterized by well-defined shapes and robust signal intensity in our research—stand in marked contrast to the low, rectangular peaks exemplified in Figure 1b of the cited study¹. That is to say, the characteristic peak profiles demonstrated in our data are morphologically inconsistent with the signature artifacts of blacklist-designated regions. This observed divergence raises critical questions regarding the precision of the current blacklist annotations. Such uncertainties necessitate further experimental verification before the widespread implementation of the cited research; yet, we observed limited experimental substantiation for the blacklist in the cited work. Strikingly, we employed strict experimental signal verification at key GDG loci, as shown in your provided Supplementary materials, through extensive ChIP-qPCR and immunofluorescence validations in our MS (Revised Fig. 2g and 5h, Extended Data Fig. 3h). To further support the reliability of our data, we also conducted H3K4me3-decorated ChIP-qPCR on *Mapk1* and *Star*, as the reviewer provided in the Supplementary materials, and found significant elevations in chromatin accessibility during follicular maturation, reinforcing these findings from the original conclusion (Response Fig. 1.1a, b).

Collectively, we sincerely appreciate the value of the new blacklist resource but have noted some unresolved questions regarding its accuracy. To ensure the highest standards, we have decided to wait before including it in our analysis at present. This allows time for more community testing and experimental checks to confirm its reliability, and we will gladly adopt improved versions as they become available and widely accepted.

Response Fig. 1.1 | The quantification of the chromatin accessibility in GDGs under different follicular maturation phases. **a** and **b**, ChIP-qPCR is used to measure the relative H3K4me3 levels for GAAs within the corresponding **a** *Mapk1* and **b** *Star* gene at the antral and ovulatory phases. The enrichment is normalized to a 1:10 dilution of the input. Error bars indicate the mean \pm S.E.M. of three independently performed experiments. One-way ANOVA followed by Dunnett's multiple comparisons test is used for statistical analysis. *** $p < 0.001$.

Q1.3: The methodology surrounding the IDR analysis of GAAs and GIAs remains unclear. By standard definition, IDR must be performed on raw peak calls from biological replicates, following proper filtering and quality control. It is not clear from the authors' response whether these steps were followed consistently. Moreover, while the authors compare IDR-derived peak sets to BEDTools-merged consensus peaks, they do not provide key IDR metrics such as reproducibility scores or thresholds. Without these, it is difficult to assess the reliability of the peak reproducibility claims. The argument that BEDTools merging captures rare but meaningful signals may be valid; however, the lack of overlap with IDR-defined peaks raises legitimate concerns about biological reproducibility. Given that the GAAs and GIAs are central to the study's conclusions, the lack of rigorous reproducibility assessment limits confidence in their functional relevance.

R1.3: We are grateful for your expert guidance on the methodology for GAAs and GIAs and apologize for omitting details regarding the IDR analysis. Indeed, your emphasis on proper replicate peak processing perfectly aligns with our practices, and we confirm our IDR analysis adhered to the 0.05 significance cutoff with full metrics now provided (Response Fig. 1.2a, b). Regarding the peak selection strategy, we sincerely appreciate your valid observation about the reproducibility advantages of IDR. As noted in our previous point-by-point response to Q1.15, IDR-derived peaks represented a high-stringency subset of our BEDTools-merged consensus peaks. While this ensures technical rigor, we respectfully note that stringent thresholds may exclude biologically relevant low-frequency sites, a concern mitigated by the BEDTools approach, which retains regions significant in ≥ 1 replicate, as commonly implemented in recent epigenomic studies^{2, 3, 4, 5}. Moreover, while IDR-based peak correction would not alter the fundamental conclusions, restructuring the foundational peaks, including the initial data processing steps, would demand substantial computational resources and time. Given that this study is nearing completion after several years of development, reanalysis would trigger a cascade of downstream re-validation efforts. Moreover, we have prioritized transparency by raw data deposition to empower community verification (GEO accession: GSE267974), which enables others to replicate the IDR-based analysis. We deeply value your methodological rigor and welcome further guidance on balancing these considerations in future work. Again, thank you for your understanding and support.

Response Fig. 1.2 | IDR metrics for pCGs during the antral and ovulatory phases. **a** and **b**, The typical peak ranks, scores and IDR values at the **a** antral phase and **b** ovulatory phase in IDR analysis. The corresponding filtering and quality control metrics are listed below.

Q1.4: Finally, the use of PCA as a justification for reproducibility of peak-level data is methodologically inappropriate. PCA can demonstrate sample-level variability or clustering but is not a valid proxy for evaluating the reproducibility of specific genomic regions like peaks or accessible chromatin sites.

R1.4: Thank you very much for your careful reading! We fully agree with your comment that PCA primarily captures sample-level variance rather than region-specific consistency. We sincerely apologize for the inappropriate analytical approach in our previous point-by-point response to Q1.15. We hope that the other details shown in R1.15, including the mapping ratio, NRF, and FRiP, can help address the reproducibility of our data across biological replicates. Again, thank you for your constructive comments, which have significantly strengthened the clarity of our study.

Supplementary materials: To illustrate some of the concerns above, I have included IGV snapshots of several genomic regions featured in the manuscript. These include ATAC-seq and CUT&Run signal tracks, the Sus scrofa blacklist, the ATAC-seq nc, and the CUT&Run IgG control. Regardless of overlap with the blacklist, the enrichment observed at several of these loci appears inconsistent or potentially confounded by background signal, and may benefit from additional validation or clarification.

Response: We sincerely appreciate your follow-up inquiry regarding the blacklist considerations raised in Q1.3. Regarding the points of doubt concerning our epigenomic data and the blacklist, we have provided a detailed explanation in R1.3, and we sincerely hope these explanations address your concerns. While we stand by our analytical approach, we recognize that the proposed blacklist holds value pending further validation through experimental verification. Once again, thank you for the constructive comments, which help advance the understanding of our work!

Reviewer #2 (Remarks to the Author):

Summary:

I would like to thank the authors for providing an extensive point-by-point response to my concerns and for their efforts to improve their manuscript. And the authors have addressed the concerns I have raised. So, I suggest to accept it.

We are deeply grateful for your kind recommendation to accept our MS. Your thoughtful feedback not only significantly enhanced the quality of our work but also confirmed that our point-by-point revisions met your expectations. Moreover, your insights will continue to inform and guide our future research. Thank you once again for your invaluable contributions!

References

1. Kong M, *et al.* Identification of blacklist regions in cattle and pig genomes. *Genomics* **117**, 111027 (2025).
2. Li D, *et al.* Dynamic transcriptome and chromatin architecture in granulosa cells during chicken folliculogenesis. *Nat Commun* **13**, 131 (2022).
3. Hoelzl S, Hasenbein TP, Engelhardt S, Andergassen D. Aging promotes reactivation of the Barr body at distal chromosome regions. *Nature Aging*, (2025).
4. Van Nerum K, *et al.* α -Ketoglutarate promotes trophoctoderm induction and maturation from naive human embryonic stem cells. *Nature Cell Biology*, (2025).
5. Tian Y, *et al.* Histone H1 deamidation facilitates chromatin relaxation for DNA repair. *Nature*, (2025).

The revised manuscript addresses several of the technical concerns raised in my initial review, and I would like to acknowledge the authors' thoughtful responses and the additional data provided to improve the clarity and transparency of the genomic analyses. These efforts are appreciated. That said, several key issues related to the robustness and reproducibility of the epigenomic analyses remain insufficiently addressed, and I believe they still warrant further consideration before the conclusions can be fully supported.

In response to Comment 1.10 and Figure 4, the revised manuscript presents heatmaps and profile plots suggesting a role for *Fosl2* in chromatin accessibility. While these visualizations are informative, the central conclusion would benefit from additional quantitative analyses supported by appropriate statistical tests. This would help reinforce the claim and provide a stronger evidentiary basis for *Fosl2*'s proposed function. The inclusion of summary statistics (such as average signal intensities with boxplots) would enhance interpretability and address potential concerns regarding signal variability.

The Spearman's correlation analyses presented offer some insight into global signal trends, but the strength of the correlation still appears modest. As such, the conclusion that *Fosl2* is a key determinant of open chromatin might currently appear somewhat overstated. Presenting this as a trend, rather than a definitive conclusion, or complementing the analysis with additional quantitative comparisons, would improve the balance between interpretation and evidence.

The authors have addressed the lack of a standardized pig genome blacklist by applying various filtering strategies, including the use of mm10 blacklist homologs. While this effort is appreciated, I would respectfully suggest that a direct comparison with the most recent *Sus scrofa* blacklist could further clarify the extent to which artifact-prone regions were addressed. Even a partial or preliminary overlap analysis would enhance transparency and help contextualize the limitations and strengths of the dataset.

The methodology surrounding the IDR analysis of GAAs and GIAs remains unclear. By standard definition, IDR must be performed on raw peak calls from biological replicates, following proper filtering and quality control. It is not clear from the authors' response whether these steps were followed consistently.

Moreover, while the authors compare IDR-derived peak sets to BEDTools-merged consensus peaks, they do not provide key IDR metrics such as reproducibility scores or thresholds. Without these, it is difficult to assess the reliability of the peak reproducibility claims. The argument that BEDTools merging captures rare but meaningful signals may be valid; however, the lack of overlap with IDR-defined peaks raises legitimate concerns about biological reproducibility. Given that the GAAs and GIAs are central to the study's conclusions, the lack of rigorous reproducibility assessment limits confidence in their functional relevance.

Finally, the use of PCA as a justification for reproducibility of peak-level data is methodologically inappropriate. PCA can demonstrate sample-level variability or clustering but is not a valid proxy for evaluating the reproducibility of specific genomic regions like peaks or accessible chromatin sites.

To illustrate some of the concerns above, I have included IGV snapshots of several genomic regions featured in the manuscript. These include ATAC-seq and CUT&Run signal tracks, the *Sus scrofa* blacklist, the ATAC-seq nc, and the CUT&Run IgG control. Regardless of overlap with the blacklist, the

enrichment observed at several of these loci appears inconsistent or potentially confounded by background signal, and may benefit from additional validation or clarification.

Refseq Genes

ATG9A

Refseq Genes

FCGR3A

M=D
HSPA6